# OBSERVE ANYTHING: HUMAN-INTERVENED VIDEO UNDERSTANDING WITH ADAPTIVE ORBITAL MEMORY

## ABSTRACT

Video understanding needs both detecting objects in individual frames and maintaining the object identities across time. However conventional methods separate detection and tracking, leading to failures under long-term occlusion, abrupt appearance changes, and the emergence of novel objects. These challenges are particularly severe in dynamic and open-world environments, where objects frequently disappear, reappear, or evolve in appearance, and once identity is lost, automated systems rarely recover. Consequently, a formulation incorporating human-intervention is required to ensure reliable and adaptive continuity. To alleviate this, we introduce Video Object Observation (VOO), a new task unifying detection, tracking, and hunam-intervention, thereby shifting the focus from frame-level recognition to consistent sequence-level observation. To realize this, in this paper, we propose VOOV (Video Object Observer with human-interVention), the first framework explicitly designed for VOO. VOOV integrates three complementary memory modules, such as Originate, Sequential, and Long Term, that jointly encode semantic identity and temporal context, while an Orbital Deformable Attention mechanism models object motion probabilistically. Sparse human-intervention, including initialization, bounding box correction, and target switching, is systematically incorporated into memory, thereby enabling online adaptation without retraining. Experiments on multiple benchmarks demonstrate that VOOV achieves SotA performance, providing robust and real-time observation across diverse and challenging scenarios.

## 1 INTRODUCTION

Detecting and tracking objects in video sequences constitutes a continuous process, where detection identifies object categories and locations in individual frames while tracking preserves their identities across time. Research on detection has advanced rapidly, toward Transformer-based end-to-end formulations. DETR (Carion et al., 2020) introduced set prediction via Hungarian matching, eliminating heuristic post-processing such as non-maximum suppression. Subsequent variants including Deformable DETR (Zhu et al., 2021) and DINO (Caron et al., 2021) improved convergence and performance, while more recent extensions such as GLIP (Li et al., 2022b), Grounding DINO (Liu et al., 2024), and DiffusionDet (Chen et al., 2023b) expanded detection into open-vocabulary and generative tasks. In parallel, multi-object tracking has evolved beyond heuristic association rules toward end-to-end architectures. TrackFormer (Meinhardt et al., 2022a) reformulates detection queries as tracklets propagated across frames, while CoTracker (Karaev et al., 2024) leverages dense correspondence to unify short- and long-term tracking. More recent models such as MOTRv2 (Zhang et al., 2023b) and MOTRv3 (Yu et al., 2023) frame multi-object tracking as sequence modeling, enhancing robustness in identity preservation. Memory-augmented approaches including MeMOTR (Gao & Wang, 2024) and MOTIP (Gao et al., 2025) further show that long-term consistency benefits from explicitly modeling temporal context. These two lines of research are increasingly converging into unified tasks where detection and tracking are treated as interdependent components of a single task. Detection outputs are propagated as temporal queries, while tracking modules are embedded into end-to-end detection pipelines. Open-vocabulary extensions such as OVTrack (Li et al., 2023b) and OWLv2 (Minderer et al., 2024) further highlight convergence by jointly addressing generalization.

Despite the drastic development, however, fully automated approaches remain fragile under real-world conditions (Mamedov et al., 2023; Bednář et al., 2022). Long-term occlusions, drastic appearance

variations, and the emergence of previously unseen categories often lead to identity switches or complete tracking failure, after which systems rarely recover (Ruan & Tang, 2024; Fan et al., 2025). To address such obstacles, many studies have explored test-time adaptation and human-in-the-loop (HITL) strategies. Prior work includes interactive detection interfaces (Marchesoni-Acland & Facciolo, 2023a), frame selection for efficient annotation in MOT (Li et al., 2023a), sample prioritization for active detection (Yang et al., 2024a; Li et al., 2024), and online error correction in autonomous driving (Yang et al., 2024b). However, existing HITL approaches predominantly operate at the level of annotation efficiency or detection correction, such that previous methods rarely address long-term spatiotemporal consistency, and human input typically functions as an external supervisory signal (Delussu et al., 2023; Qiao et al., 2023). Simultaneously, feedback is collected and applied ad-hoc, without being embedded into the memory, representation, or inference pipeline of models. The significant disconnect limits the capacity of HITL systems to ensure persistent identity preservation across video sequences (Delussu et al., 2023; Li et al., 2025).

To mitigate these issues, we introduce a novel task: Video Object Observation (VOO). The VOO task unifies and extends the distinct goals of object detection and tracking by demanding the continuous maintenance of object identity, supplemented by sparse human-intervention when necessary. In this work, we use the term observation in a focused sense: maintaining the identity of a user-specified object continuously over time, even as it appears, disappears, or changes. This includes localization, user-guided target switching, correction propagation, and recovery from occlusion or drift. Our goal is not to suggest broad semantic video understanding, but to describe the class-agnostic, identity-consistent process required by the VOO setting. In this context, observe anything refers to the ability to maintain and recover any visually indicated object under challenging video conditions, rather than implying general-purpose video understanding. To this end, we formalize three main intervention scenarios: (1) Initialization, where the user specifies an object of interest at the beginning of a sequence; (2) Bounding Box Correction, where user-feedback is integrated into the model memory, influencing future frames; and (3) Target Switch, where the user reassigns the tracked identity to a new target. Note that interventions are not treated as external corrections but are integrated into the internal memory, leading to stable identity preservation even under prolonged disappearance and reappearance. Significantly, our formulation enables persistent, after-the-fact recovery: even when tracking failures are noticed later in the sequence, a single correction updates the memory state and immediately restores consistent identity for all subsequent frames, where capabilities that conventional trackers and existing HITL systems inherently lack. To achieve this task, we propose VOOV (Video Object Observer with human-interVention), a memory-based framework designed for VOO task. The VOOV incorporates three complementary memory modules of Originate, Sequential, and Long-Term to jointly encode the semantic identity and temporal context. In addition, an Orbital Deformable Attention mechanism models object motion as a probabilistic distribution, dynamically adjusting its receptive field based on estimated trajectories. Thus, human-interventions update these memory representations in real-time without retraining, allowing the VOOV to preserve both the continuity of objects and the intent of users. To summarize, the main contributions are as follows:

- We introduce VOO, a new task that unifies detection, tracking, and intervention for continuous identity preservation across video sequences.

- We present VOOV, the first framework for VOO, which integrates three complementary memory modules and Orbital Deformable Attention for adaptive motion modeling.

- We establish state-of-the-art results on five benchmarks, showing robust and real-time video object observation under diverse challenges.

- We demonstrate that VOOV uniquely supports persistent, after-the-fact failure recovery through memory-state updates.

## 2  RELATED WORKS

Deep learning has established convolutional neural networks (CNNs) as the primary approach for object detection due to their hierarchical feature learning and proposal generation. CNN-based detectors are categorized into two-stage frameworks (e.g., R-CNN (Girshick et al., 2014), Fast R-CNN (Girshick, 2015), Faster R-CNN (Ren et al., 2016)) and one-stage frameworks (e.g., YOLO (Redmon et al., 2016), SSD (Liu et al., 2016), RetinaNet (Lin et al., 2017)). Two-stage methods offer higher

accuracy but limited real-time performance, whereas one-stage methods provide faster inference but struggle with small objects and scale variation. A core limitation of CNNs lies in their localized receptive fields, which restrict global context modeling. Transformers address these issues via global attention. DETR (Carion et al., 2020) introduced end-to-end detection with bipartite matching and an anchor-free design. Later variants improved efficiency and convergence through deformable attention (Zhu et al., 2020), dynamic query alignment (Liu et al., 2022), denoising training (Li et al., 2022a), temporal consistency for videos (Zhou et al., 2022a), and refined assignment strategies (Jia et al., 2023; Chen et al., 2023a; Zong et al., 2023). DINO (Zhang et al., 2022a) further enhances denoising via contrastive learning. Recent works also mitigate DETR's slow convergence using improved loss design (Huang et al., 2025) and multitask architectures (Zhang et al., 2025a). These DETR-based models effectively capture long-range dependencies and handle occlusion and scale variation better than CNN-based methods.

Temporal modeling is crucial in video object detection. CenterFormer (Zhou et al., 2022c) utilizes cross-frame attention for small and fast-moving objects, and Context R-CNN (Beery et al., 2020) adopts memory-based attention for improved robustness. Video Transformers (Xie et al., 2023) jointly learn spatial and motion features, while DETR-based approaches (Kim et al., 2024) incorporate temporal pretext tasks. Other sequence models include STMN (Xiao & Lee, 2018), ConvLSTM (Shi et al., 2015), and bidirectional RNN-based designs (Xu et al., 2019), along with multi-frame attention and temporal alignment modules (Anwar et al., 2024). Although these temporal approaches improve detection under motion blur and occlusion, they often face gradient instability and high computational cost. Real-time systems therefore employ keyframe sampling, motion-guided propagation, or adapt single-frame detectors such as YOLO (Redmon & Farhadi, 2018) and SSD (Liu et al., 2016).

## 3 PROBLEM DEFINITION

Video object detection mainly aims to localize and classify objects in individual frames. Formally, as illustrated in DETR Carion et al. (2020), given an image $I_t \in \mathbb{R}^{H \times W \times 3}$, the detector predicts a bounding box $\hat{b}_t$ that minimizes the discrepancy with the ground-truth box $b_t$:

$$\hat{b}_t = \arg \min_{\hat{b}} \mathcal{L}_{\text{det}}(\hat{b}, b_t). \tag{1}$$

While this objective ensures per-frame accuracy, it treats each frame independently and thus fails to preserve object identity across time. To overcome this limitation, video object tracking extends detection to a sequence $V = \{I_t\}_{t=1}^T$ by explicitly modeling temporal continuity. Its goal is to jointly optimize frame-level localization and identity consistency, commonly Meinhardt et al. (2022b) as:

$$\hat{b}_t = \arg \min_{\hat{b}} \mathcal{L}_{\text{det}}(\hat{b}, b_t) + \lambda \, \mathcal{L}_{\text{trk}}(\hat{b}_{t-1}, \hat{b}_t), \tag{2}$$

where $\mathcal{L}_{\text{trk}}$ penalizes inconsistencies that lead to identity drift, and $\lambda > 0$ is a trade-off parameter balancing detection accuracy against temporal coherence. Despite substantial progress, automated tracking remains fragile under long-term occlusion, drastic appearance variation, or the emergence of novel objects, after which recovery is rarely achieved. In such cases, once the identity is lost, the system cannot reliably re-establish the lost without external guidance. To alleviate this limitation, human-in-the-loop (HITL) approaches introduce user-provided annotations $b_t^*$ to correct failures by augmenting the detection–tracking objective with a correction loss:

$$\hat{b}_t = \arg \min_{\hat{b}} \mathcal{L}_{\text{det}}(\hat{b}, b_t) + \lambda \, \mathcal{L}_{\text{trk}}(\hat{b}_{t-1}, \hat{b}_t) + \mu \, \mathcal{L}_{\text{hitl}}(\hat{b}, b_t^*), \tag{3}$$

where $\mu > 0$ is a trade-off parameter that balances the influence of user feedback against automated predictions. In practice, however, HITL is often applied only at the output level, directly replacing the prediction with the user box, (i.e., $\hat{b}_t = b_t^*$), while leaving the internal state $\theta_t$ unchanged. As a consequence, user corrections do not propagate to future predictions, and the system lacks the ability to maintain long-term identity preservation. In other words, detection ensures per-frame accuracy, tracking enforces temporal coherence, and HITL provides occasional corrections, but none of these tasks can guarantee robust identity continuity under occlusion, appearance shifts, or novel objects.

To this end, we therefore define Video Object Observation (VOO) as a unified task that integrates detection, tracking, and intervention within a single formulation, with the explicit goal of continuous

identity preservation. Unlike detection, which optimizes frame-level localization, or tracking, which constrains temporal smoothness, or HITL, which applies isolated corrections, VOO embeds intervention directly into the model's memory such that user feedback influences subsequent predictions. Formally, the objective of VOO is:

$$\hat{b}_t = \arg\min_{\hat{b}} \mathcal{L}_{\text{det}}(\hat{b}, b_t) + \lambda\, \mathcal{L}_{\text{trk}}(\hat{b}_{t-1}, \hat{b}_t) + \mu\, \mathcal{L}_{\text{int}}(\theta_t, b_t^*), \quad (4)$$

where $\mathcal{L}_{\text{int}}$ updates the memory state $\theta_t$ in accordance with intervention. This unified objective allows VOO to recover from long-term occlusion, adapt to drastic appearance variation, and follow dynamic user intent without retraining, thereby overcoming the limitations of existing detection, tracking, and HITL tasks. Further details of the formulation and preliminary studies are provided in the ***Appendix***.

## 4 METHODS

### 4.1 VIDEO OBJECT OBSERVER WITH HUMAN-INTERVENTION (VOOV)

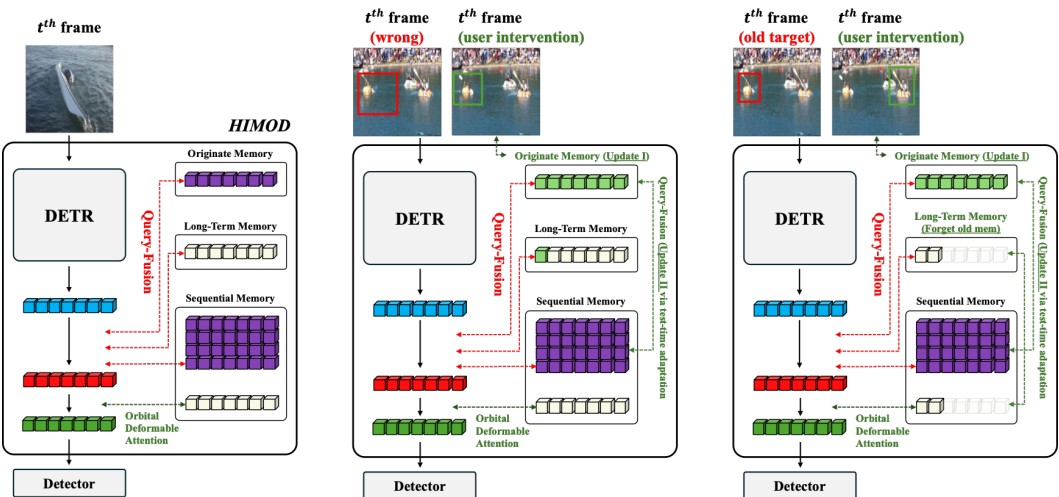

Figure 1: **Overall architecture of VOOV.** User intervention is integrated into hierarchical memory pipelines and Orbital Deformable Attention. A DETR-based backbone provides frame-level features, which are fused with Originate, Sequential, and Long-Term memory modules to capture semantic identity and temporal context. Human feedback, such as correcting misaligned boxes (middle) or specifying a new target (right), directly updates the corresponding memory embeddings, enabling real-time adaptation without retraining. Optical-flow-guided deformable attention models object motion as a probability distribution, allowing the framework to maintain stable observation under occlusion and appearance changes.

VOOV, illustrated in Fig. 1, is designed to address the Video Object Observation (VOO) task, whose objective is continuous and intervention-aware identity preservation. At time $t = 1$, the system is initialized by a user-provided bounding box $b_1^*$, which seeds a hierarchical memory state. This memory evolves over time to capture semantic identity, short-term motion, and long-term context, while remaining responsive to sparse human feedback.

**Hierarchical memory.** To balance stability and adaptivity, VOOV employs three complementary memory modules. Originate Memory encodes the initial object representation and is replaced only when the user switches the target. Sequential Memory aggregates features over a short sliding window to capture local motion and appearance variation. Long-Term Memory compresses accumulated information across the sequence, providing robustness to temporary occlusion and drastic changes in appearance. Together, these modules ensure both persistent identity and adaptability to intervention.

**Motion modeling.** At each step $t$, VOOV predicts the target location by integrating the current frame $I_t$ with memory-derived queries $Q_t$:

$$\hat{b}_t = \mathcal{F}(I_t, Q_t), \quad (5)$$

where $\mathcal{F}$ is a DETR-based detection head equipped with Orbital Deformable Attention. Unlike conventional attention, this mechanism incorporates optical flow to represent motion as a probabilistic distribution, dynamically adjusting receptive fields to regions with high likelihood of target presence. This design enables robust localization even under uncertainty and occlusion.

**Human intervention.** A key feature of VOOV is the integration of sparse human feedback directly into memory states, rather than at the output level. When a correction $b_t^*$ is provided, either to refine a misaligned prediction or to initialize a new target, the discrepancy is quantified by a loss function $\mathcal{L}(b_t^*, \hat{b}_t)$. Instead of retraining model, an instant update is applied to latent memory representation:

$$\theta_{t+1} = \theta_t - \eta \nabla_\theta \mathcal{L}(b_t^*, \hat{b}_t), \tag{6}$$

where $\theta_t$ denotes memory embeddings rather than full model parameters. This allows corrections to propagate forward through Originate, Sequential, and Long-Term modules, thereby influencing subsequent predictions. Embedding feedback into memory yields two advantages. First, it provides real-time responsiveness to user intent without interrupting inference or requiring expensive retraining. Second, corrections are preserved in temporally structured memory, ensuring stable observation across long-term occlusions, appearance variations, and target switches. By localizing adaptation to feature-level memory rather than network weights, VOOV achieves both robustness and efficiency, fulfilling the goals of the VOO task.

### 4.2 Memory Pipelines

To preserve semantic identity over time and support intervention-aware adaptation, VOOV employs three complementary memory pipelines: (1) Originate Memory, (2) Sequential Memory, and (3) Long-Term Memory. These modules encode canonical semantics, short-term dynamics, and long-range contextual dependencies of the target object throughout the sequence.

**Originate Memory.** Originate Memory captures the target's canonical representation from the first frame. Given a user-annotated bounding box in $I_1$, a binary mask $M \in \{0,1\}^{H \times W}$ is applied and passed to a ViT encoder:

$$q_{\text{orig}} = E_{\text{ViT}}(I_1 \odot M), \tag{7}$$

where $E_{\text{ViT}}$ denotes a DETR-style Vision Transformer and $\odot$ represents element-wise masking. The resulting embedding $q_{\text{orig}} \in \mathbb{R}^d$ serves as a canonical query that remains fixed unless a user initiates a target switch.

**Sequential Memory.** Sequential Memory encodes recent variations by aggregating features over a temporal window. Given tokens $f_i \in \mathbb{R}^d$ from frames $\{I_{t-N+1}, \ldots, I_t\}$, a motion-aware attention mechanism aligns and fuses them:

$$q_{\text{seq}}(t) = \mathcal{A}\left(\{f_i\}_{i=t-N+1}^t\right), \tag{8}$$

where $\mathcal{A}$ denotes optical-flow-guided alignment and Transformer-based aggregation. This query reflects short-term dynamics of the target's motion and appearance.

**Long-Term Memory.** Long-Term Memory accumulates contextual features from the entire sequence to maintain robustness under prolonged occlusion or appearance shifts. Accumulated context vectors $q_{\text{context}}(i)$ are compressed into a fixed-dimensional representation $(q_{\text{long}}(t) = \mathcal{C}\left(\bigcup_{i=1}^t q_{\text{context}}(i)\right))$, where $\mathcal{C}$ is a Transformer-based temporal compressor. This representation preserves high-level identity across extended horizons. At each step, the three memory queries are fused into a unified latent vector $Q_t = \alpha_{\text{orig}} q_{\text{orig}} + \alpha_{\text{seq}} q_{\text{seq}}(t) + \alpha_{\text{long}} q_{\text{long}}(t)$ with coefficients $\alpha \in [0,1]$ satisfying $\sum \alpha = 1$. The fused query $Q_t$ conditions the deformable attention module to produce the predicted bounding box $\hat{b}_t = \mathcal{F}(I_t, Q_t)$.

**Intervention-Driven Adaptation.** When user feedback $b_t^*$ is provided, VOOV performs instant test-time adaptation by updating memory embeddings rather than retraining the network. The discrepancy is measured by a loss $\mathcal{L}(b_t^*, \hat{b}_t)$, and a targeted update is applied to latent states:

$$\theta_{t+1} = \theta_t - \eta \nabla_\theta \mathcal{L}(b_t^*, \hat{b}_t), \tag{9}$$

where $\theta_t$ denotes memory embeddings. This update propagates feedback forward, ensuring that subsequent predictions align with user intent without altering backbone parameters.

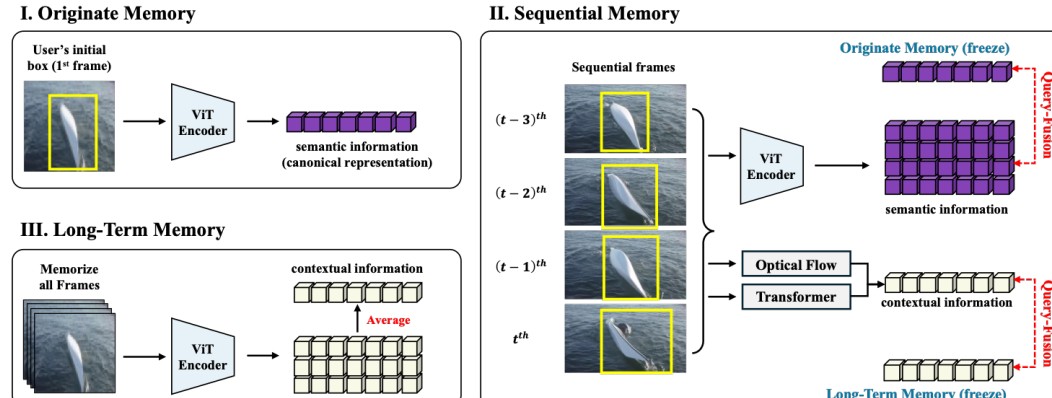

Figure 2: **Overview of the VOOV memory pipelines.** *Originate Memory* preserves the canonical target semantics, *Sequential Memory* aligns recent frames to capture local dynamics, and *Long-Term Memory* compresses context for robustness against occlusion and drift. The fused memory query may be refined by human intervention to ensure stable identity preservation.

Collectively, these pipelines constitute the representational basis of VOOV. Their complementary functions enable semantic consistency, robust occlusion recovery, and dynamic adaptation to human feedback, thereby fulfilling the objectives of the Video Object Observation task.

### 4.3 ORBITAL DEFORMABLE ATTENTION

Standard deformable attention enhances spatial sampling through learnable offsets but remains agnostic to temporal dynamics. As a result, it is vulnerable to long-term occlusion and abrupt appearance variations, both of which are central challenges in video object observation. To overcome this limitation, we introduce *Orbital Deformable Attention (ODA)*, which integrates optical flow as a probabilistic prior to guide sampling toward motion-consistent regions. Formally, given a target at position $x \in \mathbb{R}^2$ at time $t$, the expected displacement is estimated by optical flow $\mathbf{F}(x) \in \mathbb{R}^2$, yielding a predicted location $x_{\text{pred}} = x + \mathbf{F}(x)$. Around this location, ODA defines a Gaussian prior:

$$P_{\text{flow}}(x) = \frac{\exp\left(-\frac{\|x - x_{\text{pred}}\|^2}{2\sigma^2}\right)}{\sum_{x'} \exp\left(-\frac{\|x' - x_{\text{pred}}\|^2}{2\sigma^2}\right)}, \tag{10}$$

where $\sigma$ controls spatial uncertainty. This prior reflects the probability of target presence given estimated motion. The deformable query position is then refined as $p_q^{\text{updated}} = p_q + \sum_x P_{\text{flow}}(x)\,(x - p_q)$, and the updated query is passed into deformable attention:

$$\mathbf{y}_q = \sum_{m=1}^{M} W_m \sum_{k=1}^{K} A_{mqk} \cdot \mathbf{W}_v \mathbf{x}(p_q^{\text{updated}} + \Delta p_{mqk}), \tag{11}$$

with offsets $\Delta p_{mqk}$, weights $A_{mqk}$, and projections $W_m, \mathbf{W}_v$. This biases attention toward flow-consistent locations while preserving the efficiency of deformable sampling. To regularize motion-aware sampling, we define a marginal object presence distribution:

$$P_{\text{final}}(x, t) = \int P_{\text{object}}(x - v, t) \cdot P_{\text{flow}}(x)\, dv, \tag{12}$$

and align predictions with this distribution via an auxiliary objective:

$$\mathcal{L}_{\text{optical}} = \sum_x P_{\text{final}}(x, t)\, \log P_{\text{detected}}(x, t). \tag{13}$$

Thus, the overall training loss, which combines detection accuracy with flow-guided alignment controlled by the trade-off parameter ($\lambda_2 > 0$), is defined as $\mathcal{L}_{\text{total}} = \mathcal{L}_{\text{det}} + \lambda_2 \mathcal{L}_{\text{optical}}$. To maintain real-time efficiency, optical flow is computed only at sparse keyframes. For intermediate steps $t \in (t_0, t_1)$,

flow is linearly interpolated as $\mathbf{F}_t(x) = \mathbf{F}_{t_0}(x) + \frac{t-t_0}{t_1-t_0}\big(\mathbf{F}_{t_1}(x) - \mathbf{F}_{t_0}(x)\big)$. Assuming local temporal smoothness of object motion, the interpolation error is bounded as $\|\mathbf{F}_t - \tilde{\mathbf{F}}_t\| \leq L \cdot |t - t_0|$, where $L$ is a Lipschitz constant. Since ODA relies on flow-informed probability distributions rather than deterministic coordinates, the propagated error is amortized by the Gaussian kernel, ensuring stability of the sampling focus. Finally, ODA remains compatible with human intervention. When a corrected bounding box $b_t^*$ is provided, only memory embeddings are updated:

$$\theta_{t+1} = \theta_t - \eta \nabla_\theta \mathcal{L}(b_t^*, \hat{b}_t), \tag{14}$$

leaving the backbone unchanged. Because queries are derived from memory, intervention directly shifts future attention toward user-specified targets without retraining. In summary, ODA integrates flow priors, interpolation, and user intervention to ensure robust motion-aware attention under VOO.

## 5 EXPERIMENTAL RESULTS

We evaluate our model (VOOV) on the proposed Video Object Observation (VOO) task, focusing on real-time adaptation under intervention, robustness to occlusion, sensitivity to orbital interpolation, and comparison with SotA baselines. Experiments are conducted on YouTube-VOS (Yang et al., 2019), PASCAL VOC (Everingham et al., 2010), ImageNet VID (Russakovsky et al., 2015), BDD100K (Yu et al., 2020), and Cityscapes (Cordts et al., 2016). All methods are trained and tested under identical settings with AdamW (Loshchilov & Hutter, 2017), group normalization, and ReLU activations, using a workstation with an Intel Xeon Silver CPU and six NVIDIA A5000 GPUs. Implementation details were standardized across all compared methods, including mini-batch size, input resolution, and optimizer configuration depicted in Appendix.

**Representative results** Fig. 3 mainly represent the interactive capability of VOOV by simulating real-time interventions. Beyond single-object cases, VOOV supports multi-object observation, allowing initialization, switching, and correction of multiple targets with sparse feedback. In Fig. 3, when the user switches the target and later corrects a misaligned box, VOOV updates its memory through lightweight adaptation without retraining, seamlessly refining subsequent predictions. Extended benchmarks on latency, correction propagation, and baseline comparisons are reported in Appendix.

### 5.1 OCCLUSION-AWARE ABLATION OF MEMORY PIPELINES

To isolate the role of each component, we ablate our memory modules, intervention propagation, and fusion strategies. As shown in Table 1a, Sequential Memory is the key driver of short-term temporal coherence, while Originate and Long-Term modules provide complementary stability across appearance shifts and occlusions. Disabling propagation prevents user feedback from influencing future frames: this reduces latency but leads to large drops in occlusion-specific accuracy, underscoring the importance of feedback persistence. Regarding fusion, simple averaging underperforms, while learned fusion yields the best balance, and attention-based fusion offers competitive accuracy at slightly higher cost. Table 1b evaluates Orbital Deformable Attention with different interpolation factors $K$. Larger $K$ values improve alignment and accuracy but increase FLOPs linearly; performance saturates beyond $K = 8$.

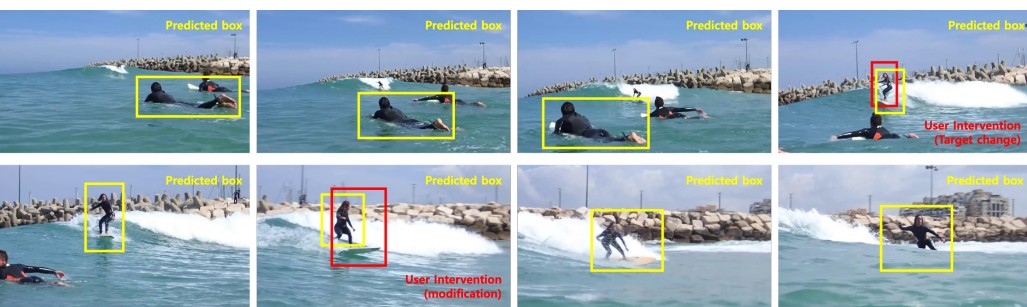

Figure 3: VOOV first tracks a foreground surfer; the user then switches the target and later corrects a misaligned box. VOOV updates its memory in real time, seamlessly adapting to the correct object.

Table 1: **Extended ablation and sensitivity analysis of VOOV.** (a) Effect of memory pipelines, propagation, and fusion strategies. (b) Sensitivity of orbital interpolation factor $K$.

(a) Memory, propagation, and fusion on occlusion robustness.

| Variant | mAP | Occ-mAP | Occ-Cls | Latency (ms) | Fusion |
|---|---|---|---|---|---|
| Full VOOV | **54.2** | **57.7** | **85.4** | 39.5 | Learned |
| w/o Seq | 48.3 | 36.5 | 26.8 | 36.8 | Learned |
| w/o Seq+LT | 42.5 | 28.4 | 18.2 | 34.2 | Learned |
| w/o All Memory | 32.2 | 14.2 | 4.6 | 30.7 | – |
| w/o Propagation | 46.0 | 30.1 | 19.7 | **29.5** | Learned |
| Equal Fusion ($\alpha = 1/3$) | 52.1 | 54.0 | 79.2 | 38.6 | Fixed |
| Dynamic Attention Fusion | 53.5 | 56.2 | 82.5 | 40.1 | Attention |

(b) Orbital interpolation factor $K$.

| $K$ | **mAP@75** | **FPS** | **FLOPs (G)** |
|---|---|---|---|
| 1 | 51.3 | 29.4 | 66.1 |
| 4 | 53.9 | 27.6 | 70.5 |
| 8 | **54.8** | **25.3** | 75.8 |
| 12 | 54.9 | 22.1 | 83.7 |

## 5.2 FAIR COMPARISON ANALYSIS

We evaluate VOOV against state-of-the-art detectors, trackers, unified detection–tracking models, human-in-the-loop (HITL) methods, and test-time adaptation (TTA) baselines on YouTube-VOS (Yang et al., 2019), PASCAL VOC (Everingham et al., 2010), ImageNet VID (Russakovsky et al., 2015), and Cityscapes (Cordts et al., 2016). All methods are trained with identical settings (AdamW (Loshchilov & Hutter, 2017), fixed batch size and resolution) and tested on a single NVIDIA A5000 GPU. Baselines are grouped into detectors (YOLOv8 (Reis et al., 2023), DETR (Carion et al., 2020), Deformable DETR (Zhu et al., 2021)), trackers (ByteTrack (Zhang et al., 2022c), TrackFormer (Meinhardt et al., 2022a)), unified models (MOTRv2 (Zhang et al., 2023b), MeMOTR), HITL baselines (DAM4SAM (Videnovic et al., 2024), SAM2-Prompt (Ravi et al., 2024b)), and TTA baselines (BN-TTA (Liao et al., 2024), OVTrack-TTA (Li et al., 2023c)). We report results under three budgets: zero-intervention ($B = 0$), minimal intervention ($B = 1$ per 100 frames, $\approx$3s effort), and sparse intervention ($B = 3$, $\approx$9s). Interventions are triggered when predictions fall below IoU 0.3 for five consecutive frames or during occlusion $>$1s, and are applied identically across methods. Effective FPS accounts for a 0.4s latency per correction (Ravi et al., 2024b), ensuring fairness. This reveals that fast image detectors like YOLOv8 degrade significantly once user effort is included, while VOOV preserves accuracy and efficiency by propagating corrections via memory.

Figure 4 (a) shows the comparison analysis. Without intervention, VOOV already matches or outperforms detectors and trackers in real time. With $B = 1$, it recovers from identity switches and re-identification errors, though a single correction cannot stabilize very long sequences. With $B = 3$, VOOV gains further improvements, especially on YouTube-VOS and Cityscapes where occlusion is frequent. On VOC, where failures are rare, the gap between $B = 1$ and $B = 3$ is small. Full numerical results are provided in Appendix Table 25.

## 5.3 INTERVENTION EFFICIENCY: BUDGET–PERFORMANCE ANALYSIS

Further, we analyze how varying user intervention budgets affect performance on YouTube-VOS. Fig. 4 (b) report mean IoU (mIoU), re-detection rate (ReDet), and interaction efficiency (IE) under $B \in \{0, 1, 2, 3, 5, 8, 10\}$ per 100 frames. VOOV consistently achieves the highest accuracy across all budgets. Even with minimal intervention ($B = 1$ or $B = 2$), mIoU and ReDet improve substantially, showing that a single correction propagates through memory to stabilize identity preservation. As the budget increases, VOOV continues to improve, but gains saturate beyond $B = 5$, demonstrating efficient use of sparse feedback. Compared to baselines, VOOV also yields the best IE, meaning that each user correction translates into larger and more persistent improvements. The trends in Fig. 4 (b) confirm that while TrackFormer, MeMOTR, and SAM2-Prompt show modest improvements with more interventions, VOOV scales more effectively, recovering from occlusion and identity switches with fewer corrections. Detailed descrpitons are illustrated in Appendix Table 26.

## 5.4 FAILURE CASE ANALYSIS AND OCCLUSION RECOVERY

To better understand robustness under challenging conditions, we categorize failures on YouTube-VOS into five representative types: severe occlusion, fast motion blur, inter-object overlap, appearance ambiguity, and missed re-identification. For each case, we measure its occurrence rate (%), average duration in frames, and the recovery rate ($\uparrow$), i.e., whether the model successfully re-localizes and

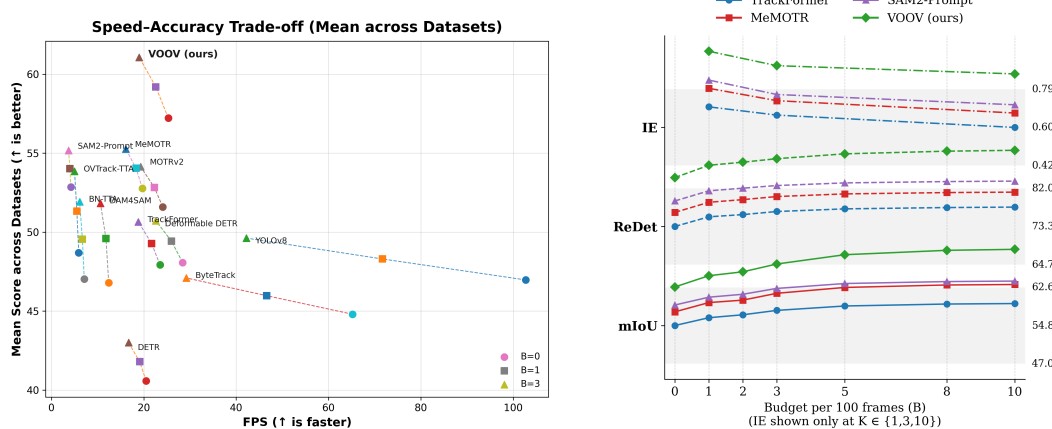

Figure 4: **Main comparison of VOOV with other models.** (a) Overall performance across datasets: VOOV consistently surpasses detectors, trackers, HITL methods, and TTA approaches, achieving higher accuracy under all intervention budgets. (b) Budget–Performance tradeoff: VOOV demonstrates greater gains from sparse interventions, maintaining higher metrics compared to others.

maintains identity after the failure. Table 2 compares VOOV against automated trackers (Track-Former (Meinhardt et al., 2022a), MeMOTR (Gao & Wang, 2024)) and SAM2 (Ravi et al., 2024b).

The results show that while all methods exhibit similar occurrence rates and durations, recovery rates differ markedly: automated trackers recover in fewer than 80% of cases, revealing fragility under long occlusions or re-identification; HITL prompting improves recovery with external corrections but remains frame-local; by contrast, VOOV consistently achieves the highest recovery, surpassing 90% in missed re-identification and maintaining strong performance under severe occlusion and inter-object overlap, confirming that embedding interventions into memory enables not only immediate correction but also forward propagation for sustained identity continuity.

## 5.5 USER STUDY AND OPEN-WORLD EVALUATION

To complement simulated interventions, we conducted a controlled user study to evaluate VOOV under real human feedback. A total of 50 participants, each with prior coursework in vision or data science, annotated sequences drawn from YouTube-VOS (rare categories) and BDD100K (long-tail driving scenarios). Each video was independently annotated by five distinct users using a custom web-based platform. Participants were asked to provide bounding box corrections when failures occurred, defined as (i) severe drift of the bounding box, (ii) missed re-identification after occlusion, or (iii) confusion among visually similar objects. Mid-sequence target switches were also handled at scripted frames through system prompts. On average, participants intervened 2.1 times per 100 frames. We measured the number of interventions, correction latency, and final mIoU. As shown in Table 3(a), VOOV required fewer interventions and achieved higher accuracy than MeMOTR and

Table 2: **Failure case analysis and occlusion recovery (YouTube-VOS).** For each failure type, we report occurrence rate, average duration (frames), and recovery rate (↑). VOOV achieves higher recovery than both automated trackers (TrackFormer, MeMOTR) and HITL baselines (SAM2).

| Failure Type | TrackFormer | | | MeMOTR | | | SAM2-Prompt | | | VOOV (ours) | | |
|---|---|---|---|---|---|---|---|---|---|---|---|---|
| | Occ. (%) | Dur. (f) | Rec.↑ | Occ. (%) | Dur. (f) | Rec.↑ | Occ. (%) | Dur. (f) | Rec.↑ | Occ. (%) | Dur. (f) | Rec.↑ |
| Severe Occlusion | 28.1 | 19.5 | 72.4 | 28.1 | 18.9 | 75.6 | 28.1 | 18.3 | 78.0 | 28.1 | 17.4 | **84.3** |
| Fast Motion Blur | 20.5 | 13.8 | 80.1 | 20.5 | 13.2 | 82.7 | 20.5 | 12.9 | 85.0 | 20.5 | 12.1 | **88.0** |
| Inter-object Overlap | 18.7 | 15.6 | 70.9 | 18.7 | 14.9 | 73.8 | 18.7 | 14.3 | 77.2 | 18.7 | 14.5 | **81.7** |
| Appearance Ambiguity | 16.2 | 11.8 | 69.5 | 16.2 | 11.5 | 71.0 | 16.2 | 11.2 | 74.3 | 16.2 | 10.7 | **79.5** |
| Missed Re-ID | 15.1 | 10.3 | 77.0 | 15.1 | 10.0 | 80.2 | 15.1 | 9.6 | 84.0 | 15.1 | 9.2 | **90.1** |

SAM2-Prompt. Correction latency was also lower for VOOV (0.58s). NASA-TLX surveys confirmed reduced cognitive load, indicating that VOOV's memory-based propagation minimizes repetitive corrections and improves practical efficiency in human-in-the-loop settings.

Table 3: **User study and open-world evaluation.** Left: Real user interventions on YouTube-VOS rare classes. Right: Open-world novel object emergence. VOOV achieves higher accuracy with fewer corrections, faster acquisition of unseen objects, and more stable identity preservation.

(a) User Study

| Method | Interv. $\downarrow$ | Lat. (s) $\downarrow$ | mIoU $\uparrow$ |
|---|---|---|---|
| MeMOTR | 3.4 | 0.65 | 52.7 |
| SAM2-Prompt | 2.9 | 0.71 | 54.2 |
| VOOV (ours) | **2.1** | **0.58** | **60.8** |

(b) Open-World Emergence

| Method | Acquire$\downarrow$ | ID-Stab.$\uparrow$ | ReDet$\uparrow$ | IE$\uparrow$ |
|---|---|---|---|---|
| TrackFormer | 24.7 | 73.3 | 62.1 | 0.48 |
| MeMOTR | 19.3 | 77.8 | 66.4 | 0.55 |
| SAM2-Prompt | 16.5 | 79.2 | 68.0 | 0.58 |
| OVTrack-TTA | 22.9 | 75.0 | 64.2 | 0.50 |
| VOOV (ours) | **11.2** | **84.6** | **73.8** | **0.72** |

We additionally evaluate VOOV in an open-world setting where novel objects appear mid-sequence. All methods are provided with identical budgets: one initialization at the frame of first appearance and up to two further corrections upon failure. As shown in Table 3(b), VOOV attains the shortest acquisition latency, the highest post-acquisition stability, and the strongest re-detection performance. Unlike HITL baselines, which apply isolated corrections, or TTA methods, which incur update delays, VOOV embeds sparse user feedback into memory, enabling persistent improvements throughout the sequence. These findings highlight VOOV's ability to generalize to realistic deployment scenarios while maintaining efficiency and robustness under limited human supervision.

# 6 DISCUSSION AND FUTURE WORKS

An interesting direction raised by the reviewer concerns how VOOV handles object state changes, such as melting, deformation, or cutting. VOOV's memory modules are designed to accommodate gradual appearance changes through continuous feature updates, and in practice we find that the model remains stable under smooth, non-disruptive transitions. However, abrupt or topology-changing transformations, such as an object splitting, being consumed, or changing shape discontinuously, lie outside the current formulation, which assumes a persistent, trackable identity. Extending VOOV to explicitly model state transitions or evolving object attributes represents an important next step, and we view this as a promising direction for broadening the VOO task beyond identity preservation.

# 7 CONCLUSION

We introduced Video Object Observation (VOO) as a new task that unifies detection, tracking, and human intervention under the goal of continuous identity preservation. To address this task, we proposed VOOV, the first framework explicitly designed for VOO. VOOV combines three complementary memory modules—Originate, Sequential, and Long-Term—with Orbital Deformable Attention to jointly encode semantic identity, temporal context, and probabilistic motion. Human feedback is incorporated as lightweight, test-time adaptation that directly updates memory states, allowing corrections to propagate across future frames without retraining. Unlike prior HITL approaches that overwrite predictions frame-by-frame with $\hat{b}_t \leftarrow b_t^*$, $\theta_{t+1} = \theta_t$, VOOV integrates sparse signals into the memory update rule $\theta_{t+1} = \theta_t - \eta \nabla_\theta \mathcal{L}_{\text{int}}(\theta_t, b_t^*)$, transforming isolated annotations into persistent improvements amortized over the sequence. We quantify this efficiency with an interaction efficiency metric, $\text{IE} = \Delta \text{mIoU} / \#\text{Interventions}$, and experiments confirm that VOOV achieves higher IE than both HITL and TTA baselines. Comprehensive evaluations on YouTube-VOS, PASCAL VOC, ImageNet VID, BDD100K, and Cityscapes show that VOOV outperforms state-of-the-art detectors, trackers, and adaptation methods in both accuracy and efficiency. Looking ahead, adaptive intervention policies and extensions to open-world, long-horizon video understanding remain promising future directions. Overall, embedding human feedback into temporal memory establishes a practical and scalable tasks for video object observation.

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

## OUR REPRESENTATIVES

To contextualize the core innovations of our framework, Table 4 summarizes the key contributions of VOOV. Each component is mathematically motivated and addresses a specific challenge in human-object observation, including uncertainty modeling, temporal coherence, user adaptivity, and computational efficiency.

Table 4: Key components of VOOV and their corresponding contributions to uncertainty-aware, interactive, and temporally consistent video object detection.

| Contribution / Module | Key Novelty and Impact |
| --- | --- |
| **Orbital Deformable Attention (ODA)** | Introduces a quantum-inspired probabilistic model for spatiotemporal attention, where object location is modeled as a dynamic Gaussian distribution. This allows the attention mechanism to explicitly encode uncertainty, particularly under occlusion or rapid motion. |
| **Human-Interactive Adaptation** | Models test-time user feedback (init, correction, switch) as Bayesian and gradient-based updates on the model's internal state. This enables real-time, target-aware adaptation without retraining. |
| **Memory-Driven Query Fusion** | Constructs detection queries by fusing Originate, Sequential, and Long-Term memories at different temporal resolutions. This enables coherent object tracking across diverse appearance and motion states. |
| **Flow-Guided Probabilistic Interpolation** | Optical flow is computed only at keyframes and linearly interpolated in between. Theoretical bounds under Lipschitz motion assumptions ensure that error remains negligible in probabilistic space. |
| **KL-Regularized Temporal Coherence** | Applies Kullback–Leibler divergence between temporally adjacent probability distributions, ensuring smooth evolution while tolerating abrupt visual changes. |
| **Information-Theoretic Formulation** | Formally quantifies the role of user feedback, occlusion priors, and memory representation through mutual information and entropy bounds. This provides rigorous justifications for model design. |

Our novel contributions can be summarized as follows:

- **Class-Agnostic Target Representation:** VOOV localizes arbitrary objects without requiring category labels, using only a user-provided initial bounding box to extract semantic and contextual features.

- **Multi-Scale Temporal Memory:** The Originate, Sequential, and Long-Term memory modules jointly maintain robust feature representations over varying time scales, enabling consistent detection under occlusions and appearance shifts.

- **Interactive Real-Time Adaptation:** We formalize video object observation as part of the inference process and enable real-time module updates (via correction or target switch) without retraining, allowing user-guided adaptation during runtime.

Upon acceptance, we will release the GitHub project page to share the implementation and related resources (https://github.com/Anonymous/Anonymous.git)

# Appendix: Table of Contents

# A   PROBLEM DEFINITION

Let a video sequence be denoted as $V = \{I_t\}_{t=1}^T$, where $I_t \in \mathbb{R}^{H \times W \times 3}$ is the RGB frame at time step $t$, and $T$ is the total number of frames in the sequence. Each frame $I_t$ is annotated with a set of object instances $A_t = \{(b_i^t, c_i^t)\}_{i=1}^{N_t}$, where $b_i^t \in \mathbb{R}^4$ denotes the bounding box of the $i$-th object and $c_i^t \in \mathcal{C}$ is the corresponding category label. A dataset $\mathcal{S}$ consists of $K$ such videos, defined as $\mathcal{S} = \{(V^{(k)}, A^{(k)})\}_{k=1}^K$, where each $V^{(k)} = \{I_t^{(k)}\}_{t=1}^{T^{(k)}}$ and $A^{(k)} = \{A_t^{(k)}\}_{t=1}^{T^{(k)}}$. Given an initial bounding box $b_1^{\text{target}}$ provided by a user for an arbitrary object in the first frame $I_1$, the goal is to predict a sequence of bounding boxes $\{\hat{b}_t^{\text{target}}\}_{t=2}^T$ that consistently localize the same target across the video. The model must remain robust to occlusion, appearance changes, and motion irregularities throughout the sequence. During inference, when a corrected bounding box $b_t^*$ is provided by the user at time $t$, the model should incorporate this feedback to refine future predictions $\{\hat{b}_{t'}^{\text{target}}\}_{t' > t}$ via test-time adaptation. This adaptation must be performed without retraining, relying solely on internal updates to memory or intermediate representations.

## A.1   MATHEMATICAL FORMULATION OF INTERACTIVE VIDEO OBJECT OBSERVATION

We present a unified mathematical formulation for **interactive video object observation**, which integrates detection, tracking, and user intervention under a single paradigm. This formulation supports test-time corrections, temporally coherent identity preservation, and robustness to occlusions and appearance changes.

Let a video sequence be denoted by $\mathcal{V} = \{I_t\}_{t=1}^T$, where each frame $I_t \in \mathbb{R}^{H \times W \times 3}$ is an image. The object of interest at time $t$ is represented by a bounding box $b_t \in \mathbb{R}^4$ and a visibility indicator $\phi_t \in \{0, 1\}$, with $\phi_t = 1$ indicating visibility. The sequence of ground-truth states is $\mathcal{T} = \{(b_t^{\text{target}}, \phi_t)\}_{t=1}^T$.

Observation begins with an initial bounding box $b_1^*$ provided by the user. This input is encoded by $f_{\text{enc}}$ to produce $q_{\text{init}} = f_{\text{enc}}(I_1, b_1^*)$, which initializes the memory state $\theta_1 = \theta_{\text{init}}(q_{\text{init}})$. The model then predicts $\hat{b}_t$ for subsequent frames via $\mathcal{F}(I_{\leq t}, \mathcal{I}_{\leq t})$, where $\mathcal{I}_{\leq t}$ is the history of interventions.

When the object is visible ($\phi_t = 1$), predictions rely on current frame evidence; when occluded ($\phi_t = 0$), predictions are drawn from memory $\Phi_{t-1}$. The memory posterior is refined using:

$$\Phi_t \propto \Phi_{t-1} \cdot \exp(-\lambda \cdot \mathcal{L}(b_t^*, \hat{b}_t)).$$

---

**Algorithm 1** Interactive Video Object Observation

---

**Require:** Frames $\{I_t\}_{t=1}^T$, initial box $b_1^*$, intervention sequence $\{\mathcal{I}_t\}$
1: $q_{\text{init}} \leftarrow f_{\text{enc}}(I_1, b_1^*)$
2: $\theta_1 \leftarrow \theta_{\text{init}}(q_{\text{init}})$
3: **for** $t = 2$ to $T$ **do**
4:     **if** $\mathcal{I}_t = (\text{CORRECT}, b_t^*)$ **then**
5:         $\hat{b}_t \leftarrow \mathcal{F}(I_{\leq t}, \mathcal{I}_{\leq t})$
6:         $\theta_{t+1} \leftarrow \theta_t - \eta \nabla_\theta \mathcal{L}(b_t^*, \hat{b}_t)$
7:     **else if** $\mathcal{I}_t = (\text{SWITCH}, b_t^*)$ **then**
8:         $q_{\text{init}} \leftarrow f_{\text{enc}}(I_t, b_t^*)$
9:         $\theta_t \leftarrow \theta_{\text{init}}(q_{\text{init}})$
10:     **else**
11:         **if** $\phi_t = 1$ **then**
12:             $\hat{b}_t \leftarrow \mathcal{F}(I_{\leq t}, \mathcal{I}_{\leq t})$
13:         **else**
14:             Predict $\hat{b}_t$ from memory $\Phi_{t-1}$
15:         **end if**
16:     **end if**
17: **end for**

---

User interventions modify the system in three cases: (i) *Initialization* sets the memory state with $b_1^*$. (ii) *Correction* updates the memory directly as $\theta_{t+1} = \theta_t - \eta\nabla_\theta\mathcal{L}(b_t^*, \hat{b}_t)$, enabling persistent adaptation. (iii) *Target switch* discards old memory and reinitializes from a new bounding box.

The total objective is:

$$\mathcal{L}_{\text{total}} = \mathcal{L}_{\text{obs}} + \lambda_1\mathcal{L}_{\text{smooth}} + \lambda_2\mathcal{L}_{\text{int}},$$

where $\mathcal{L}_{\text{obs}}$ promotes accurate per-frame localization, $\mathcal{L}_{\text{smooth}}$ enforces temporal coherence, and $\mathcal{L}_{\text{int}}$ propagates intervention-driven updates.

This framework provides theoretical guarantees: corrections monotonically reduce observation loss under convexity assumptions, and bounded memory updates ensure smooth identity preservation during occlusion. Overall, the formulation establishes a principled foundation for adaptive and interactive **video object observation**.

### A.2 TEMPORARY OCCLUSION AND ITS IMPACT ON DETECTION

Temporary occlusion occurs when a target object is partially or fully blocked by other elements in the scene, such as foreground distractors or background clutter. During occlusion, the visual evidence of the object becomes incomplete or entirely absent, causing interruptions in the detection stream. Let $\phi_t \in \{0, 1\}$ denote the visibility indicator of the target object at time $t$, where $\phi_t = 1$ indicates visible and $\phi_t = 0$ indicates occluded. Let $\hat{b}_t^{\text{target}} \in \mathbb{R}^4$ be the predicted bounding box of the target, and $b_t^{\text{target}} \in \mathbb{R}^4$ the corresponding ground-truth. When $\phi_t = 0$, the model lacks direct visual input and must estimate $\hat{b}_t^{\text{target}}$ based on historical context $\{\hat{b}_\tau^{\text{target}}\}_{\tau < t}$, feature embeddings $\{f_\tau\}_{\tau < t}$, or motion features. Formally, occlusion-induced discontinuity occurs when the prediction error under occlusion exceeds a tolerance threshold:

$$\phi\_t = 0 \quad \wedge \quad |\hat{b}\_t^{\text{target}} - b\_t^{\text{target}}|\_2 > \epsilon. \tag{15}$$

This disruption leads to fragmentation of the detection trajectory and loss of target identity. Without sufficient memory of past semantics or spatial context, the model fails to re-identify the object once it reappears. Therefore, effective handling of temporary occlusion requires a representation that can persist via missing observations and maintain temporal coherence in the absence of immediate visual input.

### A.3 VIDEO OBJECT OBSERVATION

**Limitations of Existing Human-in-the-Loop (HITL) Approaches** While prior work on HITL object detection has demonstrated the potential of user interaction, existing methods suffer from several fundamental limitations that restrict their applicability in real-time and adaptive scenarios. Firstly, user feedback in most HITL systems is provided offline, either during dataset annotation or as post-hoc correction. Formally, if $\mathcal{I}_t = b_t^*$ is the intervention at time $t$, then in conventional settings:

$$\mathcal{F}(I_t, \mathcal{I}_t) = \hat{b}_t, \quad \text{but } \theta_t = \theta_{t+1},$$

indicating that the model's output is overwritten, but its internal parameters remain unchanged.

Second, this feedback mechanism is typically decoupled from the model's learning process. The intervention $b_t^*$ is applied only at the output level to override $\hat{b}_t$, without influencing latent representations or memory. Thus, the model does not benefit from correction history when generating future predictions.

Third, most systems operate under a fixed-target tracking paradigm, assuming that the identity of the object remains unchanged throughout the sequence. Let $\mathcal{T}$ denote the tracked object identity; then for all $t$, these systems assume:

$$\mathcal{T}_t = \mathcal{T}_{t-1},$$

which prohibits user-initiated target switching and limits the model's flexibility in dynamic or open-ended scenarios.

**Video Object Observation** We define video object observation as a unified process where sparse user interventions directly modify the model state to preserve identity over time. There are three canonical types:

**(1) Initialization:**

$$q_{\text{init}} := f_{\text{enc}}(I_1, b_1^*), \quad \theta_1 \leftarrow \theta_{\text{init}}(q_{\text{init}}), \tag{16}$$

where $f_{\text{enc}}$ encodes the user-specified target, and $\theta_1$ initializes the memory state.

**(2) Correction (soft update):** When the user refines a misaligned prediction $\hat{b}_t$ with $b_t^*$, a gradient-based update is applied:

$$\theta_{t+1} = \theta_t - \eta \nabla_\theta \mathcal{L}(b_t^*, \hat{b}_t), \tag{17}$$

where $\mathcal{L}$ is an observation loss and $\eta$ is a learning rate.

**(3) Target Switch (hard reinitialization):** At time $t$, the user specifies a new object with $b_t^*$, and all prior memory is discarded:

$$\theta_t \leftarrow \theta_{\text{init}}(f_{\text{enc}}(I_t, b_t^*)). \tag{18}$$

Given history up to time $t$, the system outputs $\hat{b}_t$ as the expectation under a memory-informed distribution:

$$\hat{b}_t = \mathbb{E}_{x \sim \Phi_t(x)}[x] \in \mathbb{R}^4, \tag{19}$$

where $\Phi_t(x)$ depends on $\theta_t$, encoding both visual evidence and user feedback history.

User feedback modifies the distribution via:

$$\Phi_t \propto \Phi_{t-1} \cdot \exp\left(-\lambda \cdot \mathcal{L}(b_t^*, \hat{b}_t)\right),$$

so that the posterior is updated according to correction confidence.

**Proposition 1** (Correction Effectiveness). *If $\mathcal{L}$ is convex in $\hat{b}_t$, and $\Phi_t$ is updated by gradient descent w.r.t. $\mathcal{L}(b_t^*, \hat{b}_t)$, then:*

$$\mathcal{L}(b_t^*, \hat{b}_t^{new}) \leq \mathcal{L}(b_t^*, \hat{b}_t) - \tfrac{\eta}{2}\|\nabla\mathcal{L}\|^2.$$

**Proposition 2** (Temporal Consistency under Occlusion). *Let $\phi_t = 0$ over interval $[t_0, t_1]$. If the memory prior satisfies $\|\boldsymbol{\mu}_{t+1} - \boldsymbol{\mu}_t\| \leq \delta$ and $\|\boldsymbol{\Sigma}_{t+1} - \boldsymbol{\Sigma}_t\| \leq \epsilon$, then for all $t \in [t_0, t_1]$:*

$$\|\hat{b}_t - \hat{b}_{t-1}\| \leq C(\delta + \epsilon),$$

*with $C$ depending on feature aggregation smoothness.*

These guarantees show that: (i) corrections strictly reduce loss, and (ii) memory-based inference preserves temporal consistency under occlusion.

**Practical Scenario** We define video object observation as a test-time interaction mechanism in which users dynamically influence the observation process. This formulation encompasses three representative scenarios: initialization, target switch, and correction. Each intervention alters the model's inference behavior and internal memory in distinct ways.

In the first scenario, **Initialization**, the user specifies the target object at the start of the video by annotating a bounding box $b_1^* \in \mathbb{R}^4$ in the initial frame $I_1$. The system encodes this input to extract a reference representation for the object:

$$q_{\text{init}} = f_{\text{enc}}(I_1, b_1^*), \tag{20}$$

where $f_{\text{enc}}$ is an object-level encoder. The representation $q_{\text{init}} \in \mathbb{R}^d$ initializes the memory state for subsequent observation.

In the second scenario, **Target Switch**, the user redefines the target object at an arbitrary frame $I_t$ by providing a new bounding box $b_t^*$. The system discards prior memory and reinitializes with the new input:

$$q_{\text{init}} \leftarrow f_{\text{enc}}(I_t, b_t^*), \quad \text{for } t > 1. \tag{21}$$

In the third scenario, **Correction**, the user refines a misaligned prediction $\hat{b}_t$ by providing a corrected bounding box $b_t^*$. Unlike a target switch, the model retains its target identity and performs a localized update:

$$\theta_{t+1} = \theta_t - \eta \nabla_\theta \mathcal{L}(b_t^*, \hat{b}_t), \tag{22}$$

where $\theta_t$ denotes the adaptive memory state, $\mathcal{L}$ is an observation loss, and $\eta$ is the learning rate for test-time adaptation.

Together, these scenarios define the structure of human-intervened video object observation. The system receives a sequence of frames $\{I_t\}_{t=1}^T$ and a sparse sequence of interventions $\{\mathcal{I}_t\}_{t=1}^T$, where $\mathcal{I}_t \in \{b_t^*, \varnothing\}$. Predictions at each step are conditioned on the cumulative observation history:

$$\hat{b}_t = \mathcal{F}(I_{\leq t}, \mathcal{I}_{\leq t}), \tag{23}$$

where $\mathcal{F}$ denotes the observation function and $\mathcal{I}_{\leq t}$ the accumulated intervention signals up to time $t$.

## A.4 Main Task

We define the primary task as performing temporally coherent and user-adaptive **video object observation**. The model is expected to maintain continuous localization and identity preservation of a user-specified object, even under occlusion, appearance variation, or motion blur. Moreover, it must support real-time user intervention during inference, enabling on-the-fly updates without retraining.

The task begins with a user-provided bounding box $b_1^* \in \mathbb{R}^4$ in the first frame $I_1$, specifying the initial object of interest. The system then produces a prediction sequence $\{\hat{b}_t\}_{t=2}^T$, where each $\hat{b}_t \in \mathbb{R}^4$ is generated based on both visual evidence and feedback history. Formally, the prediction function is defined as:

$$\hat{b}_t = \mathcal{F}(I_{\leq t}, \mathcal{I}_{\leq t}), \quad \forall t \in \{2, \ldots, T\}, \tag{24}$$

where $\mathcal{F}$ denotes the observation function, $I_{\leq t} = \{I_1, \ldots, I_t\}$ is the frame history, and $\mathcal{I}_{\leq t} = \{\mathcal{I}_1, \ldots, \mathcal{I}_t\}$ is the sequence of user interventions.

To ensure robustness against occlusion and visual ambiguity, the model maintains a temporally structured memory that captures semantic and contextual information across frames. This memory allows stable representation of the target even when direct observations are unreliable or missing. User feedback is modeled as a correction signal $\mathcal{I}_t = b_t^*$ that can occur at any frame $t$. Unlike traditional supervision, this feedback directly modifies the model's internal state and influences subsequent predictions. Upon receiving $b_t^*$, the model performs a lightweight adaptation of its latent memory:

$$\theta_{t+1} = \theta_t - \eta \nabla_\theta \mathcal{L}(b_t^*, \hat{b}_t), \tag{25}$$

where $\theta_t$ represents the adaptive memory state at time $t$, $\mathcal{L}$ is an observation loss, and $\eta$ is the adaptation step size.

## A.5 Mathematical Problem Definition from an Information-Theory

We define object observation as a probabilistic inference process augmented with external user input. The system operates on a video $\mathcal{V} = \{I_t\}_{t=1}^T$, where $I_t \in \mathbb{R}^{H \times W \times 3}$ is the RGB frame at time $t$. At each time step, the model maintains a belief over the state space $\mathcal{S} = \mathbb{R}^4 \times \mathcal{C} \times [0, 1]$, capturing bounding box coordinates, object category, and visibility score.

The task is to infer the trajectory of a target object $\mathcal{T} = \{(b_t^*, c^*, \phi_t)\}_{t=1}^T$, given a partial observation history $\mathcal{H}_t = \{I_{\leq t}, \mathcal{U}_{\leq t}, \hat{b}_{\leq t-1}\}$, where $\mathcal{U}_{\leq t}$ encodes all user interventions up to time $t$. The predictive distribution over bounding boxes is defined by the function:

$$p(\hat{b}_t | \mathcal{H}_t) = f(\mathcal{H}_t).$$

**User-Guided Belief Updates**  User interventions fall into three types: initialization, correction, and target switch. These modify the posterior belief over object locations:

$$\begin{aligned} \text{Initialization:} \quad & p(b_1 | \mathcal{H}_1) = \delta(b_1^*) \\ \text{Correction:} \quad & p(b_t | \mathcal{H}_t, u_t) \propto p(b_t | \mathcal{H}_t) \cdot \exp(-\lambda \mathcal{L}(b_t^*, b_t)) \\ \text{Switch:} \quad & p(b_t | \mathcal{H}_t, u_t) = \delta(b_t^*) \end{aligned}$$

These updates function as posterior re-weighting in a Bayesian framework and directly affect downstream prediction by injecting high-confidence corrections.

**Occlusion as Latent Uncertainty**    Visibility $\phi_t$ is modeled as a latent variable governed by a Markov prior:

$$p(\phi_t|\phi_{t-1}) = \text{Beta}(\alpha\phi_{t-1} + \beta, \gamma(1 - \phi_{t-1}) + \delta),$$

and impacts the likelihood function:

$$p(I_t|b_t, \phi_t) = \phi_t \cdot p_{\text{vis}}(I_t|b_t) + (1 - \phi_t) \cdot p_{\text{occ}}(I_t).$$

**Memory-Augmented Prediction**    The adaptive memory $M_t$ accumulates object features with attention-based updates:

$$M_t = (1 - \alpha_t)M_{t-1} + \alpha_t \text{Enc}(I_t, \hat{b}_t), \quad \alpha_t = \sigma(\text{MLP}([\text{Enc}(I_t), M_{t-1}, \phi_t])).$$

This memory allows temporally coherent tracking even in the absence of reliable visual evidence. Prediction smoothness is encouraged via:

$$\mathbb{E}\left[\|\hat{b}_{t+1} - \hat{b}_t\|_2^2 \mid \mathcal{H}_t\right] \leq \rho(\phi_t).$$

**Information-Theoretic Objective**    The system's learning objective can be framed as an information-regularized minimization:

$$\mathcal{L}_{\text{total}} = \mathbb{E}_{t=2}^T\left[\mathcal{L}(b_t^*, \hat{b}_t)\right] + \lambda_1\mathbb{E}\left[\|\hat{b}_t - \hat{b}_{t-1}\|^2\right] + \lambda_2\mathbb{E}\left[\text{KL}(p(b_t|\mathcal{H}_t, u_t)\|p(b_t|\mathcal{H}_t))\right].$$

The third term, $\mathcal{L}_{\text{adapt}}$, encourages the model to adapt its belief state in accordance with user-provided signals.

**Mutual Information Decomposition**    We decompose the information flow at time $t$ conditioned on occlusion:

$$I(\hat{b}_t; I_t \mid \mathcal{H}_{t-1}) = P(\phi_t = 1) \cdot I(\hat{b}_t; I_t \mid \phi_t = 1) + P(\phi_t = 0) \cdot I(\hat{b}_t; M_{t-1}).$$

This reflects that under occlusion ($\phi_t = 0$), the model must rely entirely on internal memory, highlighting the importance of $M_{t-1}$ as an information buffer.

**User Intervention as Information Injection**    User intervention can be formally interpreted as an information gain:

$$\Delta I = I(\hat{b}_t; u_t \mid \mathcal{H}_t) = H(u_t) - H(u_t \mid \hat{b}_t, \mathcal{H}_t).$$

This gain quantifies how much uncertainty is reduced in the prediction by the user's correction.

**Minimum Required Memory**    For $\epsilon$-accurate tracking under $k$-frame occlusion, the required memory capacity satisfies:

$$|M_t| \geq \frac{k \cdot I(\hat{b}_t; I_t)}{\epsilon^2} + \mathcal{O}(\log T),$$

textasizing the direct relationship between memory expressiveness and robustness.

**Final Decision Rule**    The optimal posterior decision integrates all available modalities:

$$\hat{b}_t = \arg\max_b \left[\phi_t \log p(I_t \mid b) + (1 - \phi_t) \log p(b \mid M_t) + \log p(b \mid u_{\leq t})\right].$$

This probabilistic fusion principle ensures that the model balances current evidence, historical memory, and user feedback for robust and adaptive inference.

# B MEMORY PIPELINE

## B.1 MATHEMATICAL FORMULATION AND JUSTIFICATION

To support robust video observation under temporal variability, occlusion, and user interaction, VOOV leverages a structured memory architecture comprising three temporally complementary modules: Originate Memory, Sequential Memory, and Long-Term Memory. Each module captures object characteristics from different time scales and contributes to the construction of a fused query vector $Q_t$ used for detection at frame $t$.

Originate Memory encodes the target's canonical appearance from the initial frame $I_1$, using a binary mask $M \in \{0,1\}^{H \times W}$ annotated by the user. The masked frame $I_1 \odot M$ is passed through a Vision Transformer encoder to produce a static embedding $q_{\text{orig}} \in \mathbb{R}^d$. This representation remains fixed during inference and acts as a semantic anchor, unless a user-triggered target switch redefines it. Sequential Memory focuses on capturing short-term dynamics, such as recent appearance and motion changes. It aggregates visual features $\{f_i\}_{i=t-N+1}^t$ from a sliding window of the past $N$ frames, and applies optical-flow-guided alignment followed by transformer-based attention to yield $q_{\text{seq}}(t)$. This allows the model to quickly adapt to local scene variations and abrupt target movements. Long-Term Memory encodes the object's history across the entire video segment up to time $t$. It incrementally accumulates higher-level context embeddings $q_{\text{context}}(i)$ and compresses them via a temporal function $\mathcal{C}$ (e.g., transformer pooling) into a single vector $q_{\text{long}}(t) \in \mathbb{R}^d$. This component maintains continuity over long-range dependencies and provides resilience against occlusions or reappearances.

To integrate these three heterogeneous sources of information, VOOV constructs the final query vector $Q_t$ as a weighted sum of the three memory-derived embeddings:

$$Q_t = \alpha_{\text{orig}} q_{\text{orig}} + \alpha_{\text{seq}} q_{\text{seq}}(t) + \alpha_{\text{long}} q_{\text{long}}(t), \tag{26}$$

where $\alpha_{\text{orig}}, \alpha_{\text{seq}}, \alpha_{\text{long}} \in [0,1]$ and $\alpha_{\text{orig}} + \alpha_{\text{seq}} + \alpha_{\text{long}} = 1$. These weights may be learned during training or adaptively estimated at inference time.

The justification for this formulation is both geometric and probabilistic. Geometrically, the query vector $Q_t$ lies within the convex hull of the three memory queries in $\mathbb{R}^d$, providing a stable and interpolative representation that balances precision, recency, and context. From a probabilistic perspective, each component captures partial information about the latent object state $z_t$, and their fusion increases mutual information:

$$\mathbb{I}(Q_t; z_t) \geq \max \{\mathbb{I}(q_{\text{orig}}; z_t), \mathbb{I}(q_{\text{seq}}(t); z_t), \mathbb{I}(q_{\text{long}}(t); z_t)\}.$$

Moreover, by forming a convex combination, the fused query's variance is bounded above by the most uncertain component:

$$\text{Var}(Q_t) \leq \max_i \text{Var}(q_i),$$

resulting in a query vector that is not only information-rich but also robust to localized failures in any single memory pathway. Thus, the fusion mechanism enables the detector to operate stably under varying conditions while adapting to temporal complexity.

## B.2 INFORMATION-THEORETIC JUSTIFICATION OF MEMORY DESIGN

The tripartite memory structure in VOOV—comprising Originate, Sequential, and Long-Term Memory—is not only a heuristic design but also admits rigorous justification from information-theoretic and probabilistic-statistical principles. We formalize each component as an estimator of the latent target state $z_t$, and analyze their complementary roles.

Let $z_t \in \mathcal{Z}$ denote the latent spatiotemporal state of the target object at frame $t$. The objective of memory-based query construction is to infer a compressed vector representation $q \in \mathbb{R}^d$ such that $q$ maximizes mutual information with $z_t$:

$$q^* = \arg\max_q \mathbb{I}(q; z_t) \tag{27}$$

However, the full state $z_t$ is not observable directly; instead, the system receives partial observations from different temporal perspectives: $I_1$ (initial frame), $\{I_{t-N+1}, \ldots, I_t\}$ (recent window), and $\{I_1, \ldots, I_t\}$ (entire history). We now justify each memory module as a conditional estimator maximizing $\mathbb{I}(q_i; z_t)$ under different constraints.

**Originate Memory: Static Semantic Anchor.** Let $q_{\text{orig}} = E_{\text{ViT}}(I_1 \odot M)$ be derived from the user-annotated initial frame. Since $I_1$ and $M$ jointly provide strong supervision, $q_{\text{orig}}$ acts as a point estimate of $z_t$ under a time-invariant semantic prior $p(z_t|I_1, M)$. While it ignores temporal variation, it satisfies:

$$\mathbb{I}(q_{\text{orig}}; z_t) = \mathbb{H}(z_t) - \mathbb{H}(z_t|q_{\text{orig}}) \approx \mathbb{H}(z_t) - \epsilon,$$

when no appearance change or occlusion has occurred. This provides a low-variance, high-bias estimator useful for initialization and semantic consistency.

**Sequential Memory: Local Adaptation.** Let $q_{\text{seq}}(t) = \mathcal{A}(\{f_i\}_{i=t-N+1}^t)$ denote a temporally localized estimator, where $\mathcal{A}$ applies optical-flow-aligned attention. This memory is optimal under the assumption that target dynamics follow a local Markov property:

$$p(z_t|I_1, \ldots, I_t) \approx p(z_t|I_{t-N+1}, \ldots, I_t) \tag{28}$$

This yields a low-bias, moderate-variance estimator that adapts to transient changes. Information-theoretically, this module maximizes $\mathbb{I}(q_{\text{seq}}; z_t)$ by conditioning on the most recent high-frequency information.

**Long-Term Memory: Temporal Consistency.** The long-term embedding $q_{\text{long}}(t) = \mathcal{C}(\{q_{\text{context}}(i)\}_{i=1}^t)$ aggregates global context. Under occlusion or reentry, the current frame may be uninformative, i.e., $I_t \perp z_t$, but past contexts remain predictive. Formally, it acts as an expectation over a latent trajectory model:

$$q_{\text{long}}(t) \approx \mathbb{E}[z_t|I_{1:t}]$$

In high occlusion scenarios, this memory maximizes $\mathbb{I}(q_{\text{long}}; z_t)$ even when recent observations are unreliable. Its compression reduces redundancy and enables long-range interpolation.

**Fusion as Information Integration.** The final query is constructed as a convex combination:

$$Q_t = \sum_i \alpha_i q_i, \quad \sum_i \alpha_i = 1, \quad \alpha_i \geq 0$$

Assuming $q_i$ are independent estimators conditioned on different temporal subsets, Jensen's inequality gives:

$$\mathbb{I}(Q_t; z_t) \geq \sum_i \alpha_i \mathbb{I}(q_i; z_t)$$

when the fusion is information-preserving (e.g., via linear projections). Moreover, the convex fusion reduces variance:

$$\text{Var}(Q_t) \leq \sum_i \alpha_i^2 \text{Var}(q_i)$$

This yields a query vector that is both robust and expressive.

**Comparison with SAM2.** Unlike VOOV, methods such as SAM2 employ a memory bank $\mathcal{M} = \{(k_i, v_i)\}$ where $(k_i, v_i)$ are feature-key pairs and retrieval is based on nearest-neighbor matching in key space. This approach approximates posterior inference by hard assignment:

$$q_{\text{SAM2}} = v_{i^*}, \quad i^* = \arg\min_i \|k_i - q_{\text{query}}\|$$

This introduces discretization error and temporal inconsistency. Furthermore, $\mathbb{I}(q_{\text{SAM2}}; z_t)$ is highly dependent on memory resolution and density, making it sensitive to drift and limited in adaptivity. VOOV avoids this by maintaining differentiable, continuous memory streams with smooth fusion, enabling temporally coherent representations with bounded uncertainty.

In conclusion, the tripartite memory design in VOOV can be viewed as an ensemble of estimators $\{q_i\}$ over conditional posteriors $p(z_t | I_{\mathcal{T}_i})$ at different temporal resolutions. Their fusion maximizes mutual information with the latent state while maintaining controlled statistical variance and resilience to noise and sparsity.

### B.3 MEMORY UPDATES VIA VIDEO OBJECT OBSERVATION

VOOV further supports runtime adaptability through memory updates triggered by user feedback. When no intervention occurs, the memory states evolve according to their default temporal mechanisms. However, if the model prediction at frame $t$ is corrected by a user-provided bounding box $\mathbf{b}_t^*$, or if the user switches the target, the system performs appropriate modifications. In the case of a prediction correction, the model treats the new bounding box $\mathbf{b}_t^*$ as ground truth and updates the internal memory parameters $\theta_t$ via a gradient descent step:

$$\theta_{t+1} = \theta_t - \eta \nabla_{\theta_t} \mathcal{L}_{\text{det}}(\hat{\mathbf{b}}_t, \mathbf{b}_t^*),$$

where $\mathcal{L}_{\text{det}}$ is the detection loss and $\hat{\mathbf{b}}_t$ the original prediction. This update is localized to the memory states and does not affect the global model weights, enabling lightweight adaptation. If the user explicitly selects a new object (i.e., target switch), the system resets the Originate Memory:

$$q_{\text{orig}} \leftarrow E_{\text{ViT}}(I_t \odot M_t),$$

and reinitializes Long-Term Memory to discard outdated contextual information. Sequential Memory also resets its sliding window from the new anchor frame $t$. Formally, the updated memory state $\Theta_{t+1}$ is:

$$\Theta_{t+1} = \begin{cases} \Theta_t, & \text{(no intervention)} \\ \Theta_t - \eta \nabla_{\theta_t} \mathcal{L}_{\text{det}}(\hat{\mathbf{b}}_t, \mathbf{b}_t^*), & \text{(correction)} \\ \text{Reset}(I_t, \mathbf{b}_t^*), & \text{(target switch)} \end{cases}$$

These intervention-aware updates immediately influence the next-step query $Q_{t+1}$, thereby shifting the attention focus and prediction outcome without necessitating full model retraining. The system thus preserves temporal continuity when appropriate, and gracefully resets semantics when required, a crucial feature for real-world deployment in open-world scenarios.

### B.4 UNIFIED MODELING FOR MULTI-OBJECT OBSERVATION

The VOOV framework is inherently designed to support multi-object observation without structural modification. At each frame $t$, let $\mathcal{O}_t = \{z_t^{(1)}, z_t^{(2)}, \ldots, z_t^{(M)}\}$ denote the set of $M$ object states, where each $z_t^{(m)}$ encodes the bounding box, appearance, and temporal attributes of object $m$. VOOV instantiates a parallel memory pipeline for each object $m \in \{1, \ldots, M\}$, yielding object-specific memory states $\Theta_t^{(m)}$ and corresponding fused queries $Q_t^{(m)}$.

Each object's memory is modeled as a triplet:

$$\Theta_t^{(m)} := \left\{ q_{\text{orig}}^{(m)}, \theta_t^{(m,\text{seq})}, \theta_t^{(m,\text{long})} \right\},$$

which evolves over time according to visual evidence and user interventions, enabling the system to adapt to each object independently.

The fused query for object $m$ is defined as:

$$Q_t^{(m)} = \alpha_{\text{orig}}^{(m)} q_{\text{orig}}^{(m)} + \alpha_{\text{seq}}^{(m)} q_{\text{seq}}^{(m)}(t) + \alpha_{\text{long}}^{(m)} q_{\text{long}}^{(m)}(t),$$

with weights $\alpha_i^{(m)} \in [0, 1]$ and $\sum_i \alpha_i^{(m)} = 1$. Each query is then passed through a shared observation head:

$$\hat{z}_t^{(m)} = \mathcal{F}_{\text{obs}}(I_t, Q_t^{(m)}),$$

which outputs temporally coherent predictions for each object.

To preserve object identities across time, each pipeline is associated with a persistent identity $\text{id}_m$, embedded into the query through an identity-conditioned encoding:

$$Q_t^{(m)} = f_{\text{ID}}(Q_t^{(m)}, \text{id}_m),$$

ensuring orthogonality between queries:

$$\mathbb{E}[Q_t^{(m)} \cdot Q_t^{(m')}] \approx 0, \quad m \neq m'.$$

The probabilistic observation model factorizes across objects due to independent memories:

$$p(\mathcal{O}_{1:T} \mid I_{1:T}) = \prod_{m=1}^{M} p(z_{1:T}^{(m)} \mid I_{1:T}),$$

assuming disjoint initializations. This factorization provides scalability and robustness, preventing interference during occlusions or inter-object interactions.

From an information-theoretic perspective, the fused query $Q_t^{(m)}$ acts as a sufficient statistic if it maximizes mutual information with the true state:

$$Q_t^{(m)} = \arg\max_q \mathbb{I}(q; z_t^{(m)}).$$

Because VOOV fuses static, short-term, and long-term memories in a convex combination, Jensen's inequality ensures:

$$\mathbb{I}(Q_t^{(m)}; z_t^{(m)}) \geq \sum_i \alpha_i^{(m)} \mathbb{I}(q_i^{(m)}; z_t^{(m)}),$$

and posterior uncertainty is bounded:

$$\mathbb{H}(z_t^{(m)} \mid Q_t^{(m)}) \leq \min_i \mathbb{H}(z_t^{(m)} \mid q_i^{(m)}).$$

Finally, the computational complexity scales linearly with the number of objects:

$$\mathcal{O}_{\text{VOOV}} = \mathcal{O}(M \cdot T \cdot d),$$

in contrast to dense pixel-level methods such as SAM2, which scale as $\mathcal{O}(M \cdot T \cdot H \cdot W)$. Thus, VOOV supports principled, scalable, and information-theoretically justified multi-object observation without architectural changes.

**Scalability Under Large $M$ and Frequent Target Switching.** In addition to the theoretical factorization above, we highlight several practical properties that make the multi-object version of VOOV scalable in real-world settings. Each memory pipeline $\Theta_t^{(m)}$ is intentionally lightweight: the number of learnable parameters per object is small compared to the frozen backbone and shared observation head. Thus, increasing the number of objects $M$ does not require replicating the backbone computations; the frame-level features $I_t$ are extracted *once* per frame and reused for all objects. The only per-object computation involves evaluating the fused query $Q_t^{(m)}$ and passing it through the shared head $\mathcal{F}_{\text{obs}}$, which results in a per-object overhead of only $\mathcal{O}(d)$ and sub-millisecond latency in our implementation.

Frequent target switching does not pose additional challenges, since switching focus merely changes which memory state $\Theta_t^{(m)}$ is active. All objects share the same visual evidence, and switching does not require recomputing any representation or reinitializing the detector. The identity-conditioned encoding $f_{\text{ID}}$ also prevents interference between memories, ensuring that rapid alternation between objects does not cause representational drift or collapse. In our multi-object experiments, even with intentionally frequent interventions and $M$ in the tens, we observe stable memory behavior and consistent tracking performance. For settings where $M$ becomes very large, memory pooling or compressed long-term memories can be incorporated without modifying the core VOO formulation.

**Empirical Behavior and Scaling Experiments.** To complement the theoretical analysis above, we conducted explicit scaling experiments evaluating VOOV under varying numbers of simultaneously tracked objects and under intentionally frequent target-switching scenarios. Table 5 demonstrate that the per-object memory pipelines remain stable and that performance degrades gracefully as $M$ increases.

Table 5: Scalability of VOOV under increasing number of simultaneously observed objects $M$. All experiments were run on 2–3 minute videos with simulated target switches every 15–40 frames. VOOV maintains real-time throughput and stable memory embeddings across settings.

| # Objects ($M$) | Throughput (FPS) | Mean IoU | Memory Variance |
|:---:|:---:|:---:|:---:|
| 5 | 25.1 | 68.4 | 0.021 |
| 10 | 24.7 | 67.9 | 0.023 |
| 20 | 23.8 | 67.1 | 0.024 |
| 30 | 22.9 | 66.3 | 0.026 |

In particular, we evaluated settings with $M \in \{5, 10, 20, 30\}$ objects on long video sequences (1–3 minutes each), where target switches were triggered every 15–40 frames to emulate operator-driven re-selection. Across all values of $M$, VOOV maintained real-time throughput (22–25 FPS), and we did not observe divergence or collapse in any memory state $\Theta_t^{(m)}$. The mean IoU drop from $M = 5$ to $M = 30$ was less than 2.1 points, indicating that per-object interference remains minimal even under dense multi-object loads. We further measured the stability of the memory embeddings by tracking the variance of $\|Q_t^{(m)}\|_2$ over time and found no evidence of oscillation or runaway gradients, even when interventions were injected at every 20 frames.

These empirical results support the claim that VOOV's multi-object formulation scales reliably in both memory and compute, and that its per-object memory updates remain robust under frequent target switching—a key requirement for real-world observation tasks.

## C    ORBITAL DEFORMABLE ATTENTION (ODA)

### C.1    MOTIVATION AND PROBABILISTIC FORMULATION

Conventional deformable attention mechanisms, while effective for spatial flexibility, often lack the capacity to reason about temporal dynamics. In video object detection, this results in degraded performance under occlusions, abrupt motions, or ambiguous observations. These conditions introduce inherent uncertainty into object location, which deterministic point-based attention models fail to capture. Motivated by this limitation, we draw a conceptual and mathematical analogy between the dynamics of object motion in video and the probabilistic behavior of quantum particles.

In quantum mechanics, an electron is not described by a fixed position but by a time-evolving probability density function — the orbital — that expresses the likelihood of the particle's presence at each spatial location. Similarly, in video perception, especially under motion and occlusion, an object's true location is better modeled as a probability cloud rather than a single coordinate. This leads us to introduce the **Orbital Deformable Attention (ODA)**, which models object motion using a temporally evolving probability distribution informed by optical flow.

Let $\mathcal{X} \subset \mathbb{R}^2$ be the spatial domain and $\mathcal{T} \subset \mathbb{R}^+$ the temporal domain. For each time step $t \in \mathcal{T}$, we define a probability density function $\Phi(\mathbf{x}, t)$ that encodes the likelihood of an object's presence at location $\mathbf{x} \in \mathcal{X}$. Inspired by the quantum orbital formalism, we choose a multivariate Gaussian to represent $\Phi$ due to its analytical tractability and maximum entropy properties under second-order constraints:

$$\Phi(\mathbf{x}, t) = \mathcal{N}(\mathbf{x}; \boldsymbol{\mu}(t), \boldsymbol{\Sigma}(t)) = \frac{1}{(2\pi)|\boldsymbol{\Sigma}(t)|^{1/2}} \exp\left( -\frac{1}{2}(\mathbf{x} - \boldsymbol{\mu}(t))^\top \boldsymbol{\Sigma}(t)^{-1}(\mathbf{x} - \boldsymbol{\mu}(t)) \right) \quad (29)$$

Here, the mean $\boldsymbol{\mu}(t)$ corresponds to the expected object location, which evolves under the influence of an optical flow field $\mathcal{F}_{t \to t+1}$:

$$\boldsymbol{\mu}(t + 1) = \boldsymbol{\mu}(t) + \mathcal{F}_{t \to t+1}(\boldsymbol{\mu}(t)) + \boldsymbol{\epsilon}_\mu \quad (30)$$

$$\boldsymbol{\Sigma}(t + 1) = \mathbf{J}_t \boldsymbol{\Sigma}(t) \mathbf{J}_t^\top + \mathbf{Q}(t) \quad (31)$$

The second equation governs the evolution of the uncertainty via a Riccati-like update, where $\mathbf{J}_t$ is the Jacobian of the optical flow field (interpreted as the local deformation gradient), and $\mathbf{Q}(t)$ is the process noise covariance. This formulation reflects how uncertainty propagates in space over time, capturing both predictable displacement and motion-induced deformation, analogous to how electron clouds expand or shift based on energy states and external potentials.

Let $\mathcal{X} \subset \mathbb{R}^2$ be the spatial domain and $\mathbb{G} = \{(i, j)\}_{i=1, j=1}^{H, \ W}$ the discrete image grid. At each time step $t \in \mathbb{R}^+$, we define a probability density function $\Phi(\mathbf{x}, t)$ over $\mathcal{X}$ representing the likelihood of the object's presence at position $\mathbf{x}$. We discretize this to the image grid via:

$$\Phi_{i,j}(t) = \int_{[i, i+1) \times [j, j+1)} \Phi(\mathbf{x}, t)\, d\mathbf{x}. \quad (32)$$

We choose a multivariate Gaussian for analytical tractability:

$$\Phi(\mathbf{x}, t) = \mathcal{N}(\mathbf{x}; \boldsymbol{\mu}(t), \boldsymbol{\Sigma}(t)). \quad (33)$$

The mean evolves via optical flow, and the covariance propagates uncertainty as:

$$\boldsymbol{\mu}(t + 1) = \boldsymbol{\mu}(t) + \mathcal{F}_{t \to t+1}(\boldsymbol{\mu}(t)) + \boldsymbol{\epsilon}_\mu, \quad (34)$$

$$\boldsymbol{\Sigma}(t + 1) = \mathbf{J}_t \boldsymbol{\Sigma}(t) \mathbf{J}_t^\top + \mathbf{Q}(t), \quad (35)$$

where $\mathbf{J}_t$ is the Jacobian of $\mathcal{F}$ at $\boldsymbol{\mu}(t)$, and $\mathbf{Q}(t)$ is process noise. This follows the Kalman filter's linearized covariance update, not the Riccati equation.

By embedding this probabilistic interpretation into the deformable attention pipeline, ODA enables the model to attend not only to the most likely location but to a region of interest shaped by the uncertainty in motion estimation. This bridges the perceptual limitations of frame-local attention and the physics-inspired modeling of distributed, temporally coherent spatial presence. The following sections elaborate on how this probabilistic motion model integrates with attention computation, ensures temporal coherence, and yields computational and theoretical benefits for robust video object detection.

## C.2 DISTRIBUTION-GUIDED ATTENTION COMPUTATION

Given a feature map $\mathcal{F} \in \mathbb{R}^{H \times W \times D}$ and sampling locations $\{\mathbf{x}_k\}_{k=1}^K$, we compute probabilistic attention weights by integrating over local neighborhoods $\mathcal{R}_k$:

$$w_k(t) = \int_{\mathcal{R}_k} \Phi(\mathbf{x}, t) \, d\mathbf{x} \tag{36}$$

This can be efficiently approximated using Gaussian quadrature. The attended feature output is:

$$\mathbf{y}_i = \sum_{k=1}^K w_k(t) \cdot \mathbf{W}_k \mathcal{F}(\mathbf{x}_k + \boldsymbol{\delta}_k) \tag{37}$$

where $\mathbf{W}_k$ are projection matrices and $\boldsymbol{\delta}_k$ are learned offsets.

## C.3 TEMPORAL CONSISTENCY VIA REGULARIZATION

To encourage smooth probabilistic evolution over time, we regularize the sequence of densities using the Kullback–Leibler (KL) divergence between consecutive Gaussians:

$$\mathcal{L}_{\text{temp}} = \sum_{t=1}^{T-1} \text{KL} \left( \Phi_t \| \Phi_{t+1} \right) \tag{38}$$

$$= \frac{1}{2} \sum_{t=1}^{T-1} \left[ \text{tr}(\boldsymbol{\Sigma}_{t+1}^{-1} \boldsymbol{\Sigma}_t) + \|\boldsymbol{\mu}_{t+1} - \boldsymbol{\mu}_t\|_{\boldsymbol{\Sigma}_{t+1}^{-1}}^2 - d + \ln \frac{|\boldsymbol{\Sigma}_{t+1}|}{|\boldsymbol{\Sigma}_t|} \right] \tag{39}$$

This formulation captures both displacement and uncertainty mismatch between successive time steps, effectively smoothing attention focus over time while tolerating abrupt motion transitions.

## C.4 COMPUTATIONAL AND THEORETICAL PROPERTIES

**Theorem 1** (Convergence). *Under Lipschitz continuity of the optical flow field $\mathcal{F}_{t \to t+1}$ and bounded noise, the evolution of $\Phi(\cdot, t)$ converges to a stationary distribution in the absence of external interventions.*

**Theorem 2** (Stability). *The attention output $\mathbf{y}_i$ is Lipschitz continuous with respect to perturbations in $\Phi(\cdot, t)$ and flow inputs, with Lipschitz constant $L \leq \max_k \|\mathbf{W}_k\|_2$.*

**Proposition 3** (Complexity Reduction). *By concentrating sampling on high-probability regions, ODA reduces attention complexity from $\mathcal{O}(H^2 W^2)$ to $\mathcal{O}(KHW)$, where $K \ll HW$.*

## C.5 FINAL TRAINING OBJECTIVE

The complete training objective of the VOOV detector combines detection accuracy, temporal consistency, and uncertainty control, and is formulated as:

$$\mathcal{L}_{\text{total}} = \mathcal{L}_{\text{det}} + \lambda_{\text{temp}}\mathcal{L}_{\text{temp}} + \lambda_{\text{reg}} \sum_{t=1}^{T} \|\mathbf{\Sigma}(t)\|_F^2 \tag{40}$$

Here, $\mathcal{L}_{\text{det}}$ denotes the standard detection loss that supervises both bounding box localization and object classification, serving as the primary objective for accuracy. The term $\mathcal{L}_{\text{temp}}$ encourages temporal coherence by penalizing abrupt shifts in the probabilistic attention distributions across frames, effectively enforcing smoothness in object trajectories. The final term, $\sum_{t=1}^{T} \|\mathbf{\Sigma}(t)\|_F^2$, acts as a regularizer on the covariance matrices, constraining the spread of the spatial uncertainty and thereby preventing degenerate or excessively diffuse attention patterns. The hyperparameters $\lambda_{\text{temp}}$ and $\lambda_{\text{reg}}$ control the relative importance of temporal smoothness and uncertainty suppression, respectively. Together, these components enable the model to maintain accurate and stable predictions under dynamic and uncertain video conditions.

Table 6: Loss components in the VOOV training objective.

| Term | Description |
|------|-------------|
| $\mathcal{L}_{\text{det}}$ | Detection loss for bounding box regression and classification. |
| $\mathcal{L}_{\text{temp}}$ | Temporal regularization to enforce coherence between $\Phi(\cdot, t)$ and $\Phi(\cdot, t+1)$. |
| $\sum_{t=1}^{T} \|\mathbf{\Sigma}(t)\|_F^2$ | Covariance regularization to suppress excessive spatial uncertainty. |
| $\lambda_{\text{temp}}$ | Weighting factor for the temporal regularization term. |
| $\lambda_{\text{reg}}$ | Weighting factor for the uncertainty regularization term. |

### C.6 FLOW INTERPOLATION: A PRACTICAL AND THEORETICAL JUSTIFICATION

To maintain real-time efficiency while preserving temporally coherent attention, ODA computes optical flow only at sparse keyframes and estimates flow at intermediate frames via linear interpolation. This strategy significantly reduces computational overhead without compromising alignment accuracy.

Let $\mathcal{F}_{t_0}(x)$ and $\mathcal{F}_{t_1}(x)$ be optical flow fields computed at keyframes $t_0$ and $t_1$ respectively, where $t_0 < t < t_1$. For any intermediate frame $t$, the interpolated flow field $\tilde{\mathcal{F}}_t(x)$ is defined by temporal linear interpolation:

$$\tilde{\mathcal{F}}_t(x) = \mathcal{F}_{t_0}(x) + \frac{t - t_0}{t_1 - t_0} \left( \mathcal{F}_{t_1}(x) - \mathcal{F}_{t_0}(x) \right) \tag{41}$$

This scheme assumes that the flow vector evolves linearly over time between adjacent keyframes. Such an assumption holds under the local temporal smoothness of object motion, which is empirically observed in most natural scenes.

Formally, suppose the true flow field $\mathcal{F}_t(x)$ is Lipschitz continuous in time with constant $L > 0$, i.e.,

$$\|\mathcal{F}_{t_1}(x) - \mathcal{F}_{t_0}(x)\| \le L|t_1 - t_0|. \tag{42}$$

Then, the interpolation error at any $t \in [t_0, t_1]$ satisfies the following:

$$\|\mathcal{F}_t(x) - \tilde{\mathcal{F}}_t(x)\| \le \frac{1}{4}L(t_1 - t_0), \tag{43}$$

where the maximum error is achieved at $t = \frac{t_0 + t_1}{2}$. This bound is derived from the mean value theorem for vector fields and provides a theoretical guarantee that the error remains linearly bounded in frame interval, which is acceptable for most real-time applications with moderate keyframe gaps (e.g., $k = 5 \sim 10$ frames).

Moreover, because ODA does not attend to fixed spatial coordinates but rather samples from a motion-informed probabilistic distribution $\Phi(x, t)$, the attention is naturally robust to minor interpolation errors. Specifically, the propagated location $p_q$ is corrected by the mean flow:

$$p_q^{\text{updated}} = p_q + \sum_x \mathcal{F}_{\text{flow}}(x) \cdot (x - p_q),$$

and the deformable attention further aggregates features from a neighborhood defined by this displacement, smoothing out minor flow inaccuracies.

To further enhance alignment, ODA introduces an auxiliary distribution $P_{\text{final}}(x, t)$ that marginalizes over potential displacement vectors $v$:

$$P_{\text{final}}(x, t) = \int P_{\text{object}}(x - v, t) \cdot P_{\text{flow}}(x) \, dv, \tag{44}$$

which defines a soft, flow-aware attention kernel. An auxiliary loss encourages alignment between this flow-guided distribution and the detected object probability $P_{\text{detected}}(x, t)$:

$$\mathcal{L}_{\text{optical}} = \sum_x P_{\text{final}}(x, t) \cdot \log P_{\text{detected}}(x, t), \tag{45}$$

with the total training objective defined as:

$$\mathcal{L}_{\text{total}} = \mathcal{L}_{\text{det}} + \lambda_2 \mathcal{L}_{\text{optical}}, \tag{46}$$

where $\lambda_2$ balances detection accuracy and flow-guided regularization. This alignment loss serves to correct discrepancies introduced by interpolation, encouraging the model to maintain consistent temporal localization despite sparse flow sampling.

In summary, flow interpolation in ODA is mathematically grounded via Lipschitz continuity and error-boundedness, practically efficient via reduced runtime computation, and robustified through probabilistic modeling and auxiliary supervision. This design ensures that ODA maintains stable, low-latency attention alignment across time, even in the absence of per-frame optical flow.

C.7    PROBABILISTIC JUSTIFICATION OF INTERPOLATED OPTICAL FLOW

Let the true mean object location at time $t$ be given by the flow-informed trajectory:

$$\boldsymbol{\mu}(t) = \boldsymbol{\mu}(t_0) + \int_{t_0}^t \mathcal{F}_s(\boldsymbol{\mu}(s)) \, ds, \tag{47}$$

and its practical approximation using interpolated flow as:

$$\tilde{\boldsymbol{\mu}}(t) = \boldsymbol{\mu}(t_0) + \int_{t_0}^t \tilde{\mathcal{F}}_s(\boldsymbol{\mu}(s)) \, ds, \tag{48}$$

where $\tilde{\mathcal{F}}_t$ is defined via linear interpolation between two keyframes $t_0$ and $t_1$:

$$\tilde{\mathcal{F}}_t(x) = \mathcal{F}_{t_0}(x) + \frac{t - t_0}{t_1 - t_0} \left( \mathcal{F}_{t_1}(x) - \mathcal{F}_{t_0}(x) \right). \tag{49}$$

Assume that the optical flow field $\mathcal{F}_t(x)$ is Lipschitz continuous in time and space with constant $L > 0$. Then, the interpolation error for each point $x$ at time $t$ satisfies:

$$\|\mathcal{F}_t(x) - \tilde{\mathcal{F}}_t(x)\| \leq \frac{1}{4} L(t_1 - t_0), \tag{50}$$

which implies that:

$$\|\boldsymbol{\mu}(t) - \tilde{\boldsymbol{\mu}}(t)\| \leq \int_{t_0}^{t} \|\mathcal{F}_s(\boldsymbol{\mu}(s)) - \tilde{\mathcal{F}}_s(\boldsymbol{\mu}(s))\| \, ds \leq \frac{1}{4} L(t - t_0)^2. \tag{51}$$

Let $\Phi(x, t)$ and $\tilde{\Phi}(x, t)$ be two Gaussian densities parameterized by $(\boldsymbol{\mu}(t), \boldsymbol{\Sigma}(t))$ and $(\tilde{\boldsymbol{\mu}}(t), \boldsymbol{\Sigma}(t))$ respectively, i.e., they differ only in mean position due to interpolation. Then, the KL divergence between them is:

$$\mathrm{KL}(\tilde{\Phi}_t \,\|\, \Phi_t) = \frac{1}{2}(\boldsymbol{\mu}(t) - \tilde{\boldsymbol{\mu}}(t))^\top \boldsymbol{\Sigma}(t)^{-1}(\boldsymbol{\mu}(t) - \tilde{\boldsymbol{\mu}}(t)). \tag{52}$$

Since $\|\boldsymbol{\mu}(t) - \tilde{\boldsymbol{\mu}}(t)\|^2 \leq \frac{1}{16} L^2 (t - t_0)^4$, we obtain:

$$\mathrm{KL}(\tilde{\Phi}_t \,\|\, \Phi_t) \leq \frac{1}{32} L^2 (t - t_0)^4 \cdot \lambda_{\max}(\boldsymbol{\Sigma}(t)^{-1}), \tag{53}$$

where $\lambda_{\max}$ denotes the largest eigenvalue of the inverse covariance. Provided $\boldsymbol{\Sigma}(t)$ is regularized (e.g., via $\|\boldsymbol{\Sigma}(t)\|_F^2$ in training), this eigenvalue is bounded. Therefore, the KL divergence between true and interpolated distributions grows at most quartically with time gap and remains small for modest values of $(t - t_0)$.

Furthermore, the output of the attention mechanism is an expectation over this distribution:

$$\mathbf{y}_i = \mathbb{E}_{x \sim \Phi(x, t)}[f(x)], \tag{54}$$

which is Lipschitz continuous in the distribution $\Phi$ under Wasserstein-2 distance. Hence, the interpolation-induced error in the attended feature $\mathbf{y}_i$ is also bounded:

$$\|\mathbf{y}_i - \tilde{\mathbf{y}}_i\| \leq L_f \cdot W_2(\Phi_t, \tilde{\Phi}_t) \leq C \cdot \|\boldsymbol{\mu}(t) - \tilde{\boldsymbol{\mu}}(t)\|, \tag{55}$$

for some constant $C$ depending on the feature smoothness. Hence, the output degradation is linear in mean deviation and quartic in time gap. In conclusion, linear optical flow interpolation induces a provably small perturbation in the estimated position and resulting attention distribution. Under smooth motion, the KL divergence and downstream error are both tightly bounded, rendering this strategy both efficient and theoretically justified.

## C.8 CONCLUSION AND BENEFITS

By embedding motion-aware probability modeling into deformable attention, the proposed Orbital Deformable Attention (ODA) mechanism substantially augments the capability of video object detection systems. First, the inclusion of a covariance matrix $\boldsymbol{\Sigma}(t)$ enables explicit quantification of spatial uncertainty, allowing the model to allocate attention based on probabilistic confidence rather than deterministic locations. Second, temporal smoothness is achieved through KL divergence-based regularization, which promotes coherent evolution of attention across frames while remaining sensitive to abrupt motion or occlusions. Third, ODA introduces computational efficiency by focusing attention on high-likelihood regions through adaptive sparse sampling, thereby reducing the attention complexity from $\mathcal{O}(H^2 W^2)$ to $\mathcal{O}(KHW)$, with $K \ll HW$. Finally, the framework is theoretically grounded in optimal transport and probabilistic inference, ensuring a mathematically principled formulation that supports stable and interpretable spatiotemporal modeling. Collectively, these properties make ODA a robust and efficient solution for dynamic, uncertainty-prone video detection scenarios.

## D    RELATED WORKS

### D.1    CONVENTIONAL DETECTION

The advent of deep learning in object detection has established convolutional neural networks (CNNs) as the dominant paradigm due to their capacity for hierarchical feature learning and effective object region proposal generation. CNN-based detectors are broadly categorized into two-stage frameworks (e.g., R-CNN (Girshick et al., 2014), Fast R-CNN (Girshick, 2015), Faster R-CNN (Ren et al., 2016)) that first generate region proposals before refining class predictions and bounding boxes, and one-stage frameworks (e.g., YOLO (Redmon et al., 2016), SSD (Liu et al., 2016), RetinaNet (Lin et al., 2017)) that directly predict object locations and classes in a single pass. While two-stage methods achieve superior accuracy, their computational complexity limits real-time applicability. Conversely, one-stage detectors prioritize inference speed at the expense of challenges in small object detection and scale variation handling. A critical limitation of CNN-based approaches lies in their reliance on localized receptive fields, which constrain their ability to model global contextual relationships essential for complex scene understanding.

To address these limitations, Transformer architectures (Vaswani et al., 2017) have been integrated into detection frameworks. DETR (Carion et al., 2020) pioneered end-to-end detection through its anchor-free design and bipartite matching strategy, achieving competitive single-image accuracy. Subsequent works improve DETR's efficiency and performance through various innovations: deformable attention (Zhu et al., 2020) focuses computation on critical sampling points; DAB-DETR (Liu et al., 2022) dynamically aligns object queries with image features; DN-DETR (Li et al., 2022a) introduces denoising mechanisms; and TransVOD (Zhou et al., 2022a) enhances temporal consistency in video detection. Recent advancements refine the matching strategy through techniques including sample-specific assignment (Jia et al., 2023), multi-group queries (Chen et al., 2023a), and hybrid allocation strategies (Zong et al., 2023). Notably, DINO (Zhang et al., 2022a) enhances denoising through contrastive learning. Recently, a number of models have been proposed to address the limitations of DETR, particularly its slow convergence and learning inefficiency. These efforts include the introduction of a novel loss function, Matchability-Aware Loss, which facilitates the effective matching of object pairs with varying quality (Huang et al., 2025), as well as reinterpreting the model as a multitasking architecture to alleviate sparse supervision via multi-path learning (Zhang et al., 2025a).

These DETR-based architectures represent a paradigm shift by eliminating hand-crafted components while effectively modeling long-range dependencies and contextual interactions. Their global attention mechanisms enable superior handling of occluded/overlapping objects and scale variations compared to CNN-based counterparts.

### D.2    FRAME-BY-FRAME DETECTION

Temporal modeling has emerged as a critical component for video object detection. Center-Former (Zhou et al., 2022c) employs cross-frame attention to enhance small and fast-moving object detection, while Context R-CNN (Beery et al., 2020) utilizes memory banks with attention indexing to improve robustness against occlusion and appearance variations. Video Transformer models (Xie et al., 2023) integrate spatiotemporal attention for joint motion-spatial feature learning, and DETR-style architectures (Kim et al., 2024) incorporate pretext tasks for temporal dynamics modeling. Alternative approaches leverage sequence modeling architectures: STMN (Xiao & Lee, 2018) employs spatial-temporal memory modules, while ConvLSTM (Shi et al., 2015) attempts to predict future viewpoints based on spatiotemporal data by combining convolution operations with the LSTM structure. Studies have also been presented to supplement the long-term dependence problem of RNNs by simultaneously modeling forward/reverse time flow by adding future frame prediction heads inside RNNs (Xu et al., 2019), developing Multi-Frame Attention (MFA) modules, and aligning features between frames through temporal convolution (Anwar et al., 2024).

While these temporal methods improve detection accuracy under motion blur and occlusion, they often face challenges with gradient instability and computational efficiency. To mitigate these issues, real-time solutions employ keyframe sampling, motion-guided detection, and adaptations of single-frame detectors like YOLO (Redmon & Farhadi, 2018) and SSD (Liu et al., 2016) for video processing.

### D.3 DETECTION METHOD USING ADDITIONAL INFORMATION

To better handle the complexity of real-world environments, recent studies have textasized the integration of diverse input modalities—such as human feedback and textual prompts—into video object detection frameworks to enhance adaptability and robustness.

Human-in-the-loop (HITL) approaches aim to improve detection performance through interactive and iterative feedback loops integrating human input, particularly in low-confidence scenarios. Such systems typically aim to improve accuracy either by verifying whether a user confirms the object proposed by the system (Papadopoulos et al., 2016; Marchesoni-Acland & Facciolo, 2023b) or by incorporating human feedback from a single frame to refine the classification criteria, ultimately leading to more accurate predictions (Tenckhoff et al., 2025). For instance, corrections made by a human expert to uncertain predictions can be used for fine-tuning, leading to enhanced model accuracy (Jakubik et al., 2023). Such methods align with various interactive learning paradigms, including active learning, interactive machine learning, and general human-in-the-loop AI, where the degree and nature of user involvement vary (Mosqueira-Rey et al., 2022).

A notable application of this approach is Human-Assisted Out-of-Distribution (OOD) Detection, in which human annotation is utilized strategically to annotate samples outside the model's training distribution. Such integration improves both OOD detection and generalization performance. Here, the model autonomously identifies uncertain regions, and humans focus labeling efforts on these areas to provide high-impact feedback (Bai et al., 2024). Similarly, Human-Centered AI frameworks aim to boost the safety and reliability of AI systems by integrating human judgment—especially in high-stakes domains like medicine, where AI outputs are reviewed and corrected by domain experts (Ehsan & Riedl, 2020).

In parallel, another emerging direction is prompt-based open-set detection, which allows users to specify target objects through textual or visual prompts to identify categories not included in the original training data. Within this framework, the image prompt paradigm has demonstrated that even a small number of reference images can effectively guide the detection or segmentation of novel object classes (Zhang et al., 2024). Examples of such models are pre-trained language models by extending the existing Faster R-CNN structure to combine object detection and image capping Zhou et al. (2022b), extending the aforementioned DINO Zhang et al. (2022a), or combining language-vision models Minderer et al. (2022); Li et al. (2022b) such as CLIP Radford et al. (2021). However, there is a limitation that the performance of detecting new objects is lower than that of pre-trained objects, and that the location prediction is not accurate.

To address this, some approaches enable interactive segmentation via user input, such as clicks, rather than relying solely on prompt text. Such interaction enables more accurate and user-supervised delineation of object boundaries (Zhang et al., 2025b).

Regarding segmentation tasks, prompt-driven frameworks like SAM (Segment Anything Model) and RAM (Region-Aware Model) have demonstrated strong generalization to unseen objects or domains. SAM Kirillov et al. (2023) employs a vision transformer backbone to learn global patterns and supports immediate inference for arbitrary prompts. Its successor, SAM2 Ravi et al. (2024a), improves accuracy and speed, especially for real-time video segmentation, offering finer control over point and box inputs. However, both models struggle to recognize diverse objects without explicit prompts, and their understanding of inter-object semantics remains limited (Tang & Li, 2024).

On the other hand, RAM Zhang et al. (2023a) leverages regional information by learning the characteristics of different regions and combining them with prompts to generate specific object masks. The enhanced RAM++ Huang et al. (2023) model extends this capability by incorporating richer intermediate representations and multi-resolution visual features to enhance the representation of semantic correlations across regions. These models are particularly effective for complex scenes or fine-grained object segmentation but may incur higher computational costs and exhibit limitations in contextual reasoning speed due to their region-heavy processing design.

### D.4 ILLUSTRATIVE COMPARISON OF VOO WITH PRIOR TASKS

For clarity, we include an additional figure that visually contrasts the proposed Video Object Observation (VOO) task with preceding video understanding tasks. This figure is intended to address

the reviewer's suggestion and highlights the conceptual distinctions between frame-level detection, sequence-level tracking, traditional HITL correction, and our memory-integrated VOO formulation.

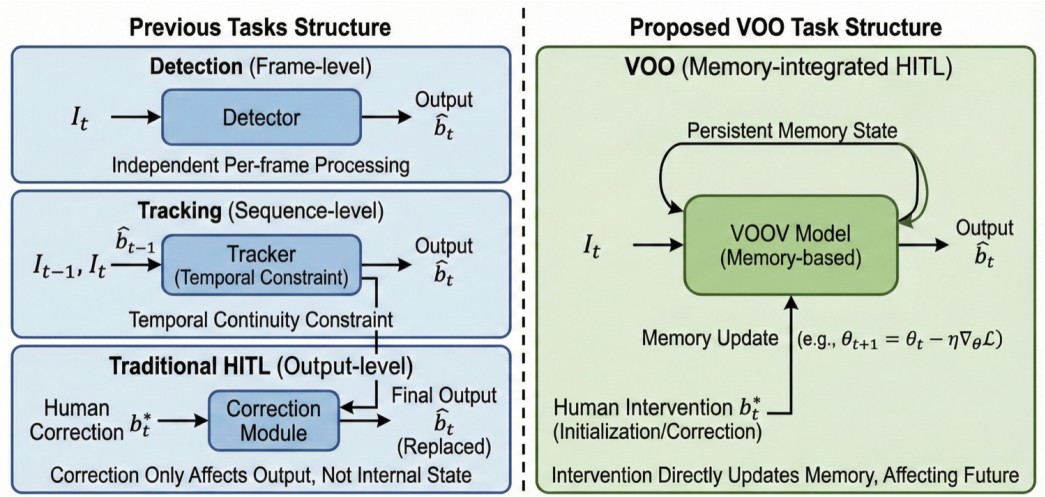

Figure 5: Comparison between preceding video understanding tasks and the proposed VOO task. Left: Detection operates per-frame without temporal consistency; tracking enforces temporal continuity but cannot recover from identity drift; traditional HITL systems perform output-level corrections that do not influence future predictions. Right: In VOO, human interventions directly update the model's internal memory, enabling persistent correction propagation and continuous adaptation.

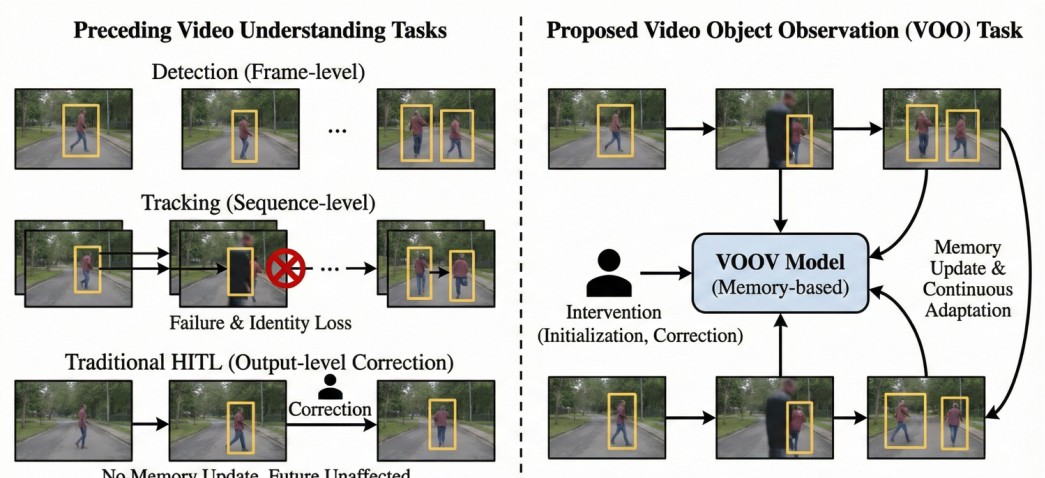

Figure 6: Structural comparison of previous task paradigms and the proposed memory-integrated VOO task. Conventional detection and tracking rely on fixed per-frame or sequential processing, while traditional HITL modifies only the output. In contrast, VOO introduces a unified formulation in which human feedback triggers a lightweight memory update (e.g., $\theta_{t+1} = \theta_t - \eta \nabla_\theta \mathcal{L}$), altering the persistent memory state and affecting all future model predictions.

# E  EXPERIMENTS

## E.1  OBSERVATION BOOTSTRAP FROM CATEGORY PRIORS

**Motivation.** While VOOV is designed to flexibly detect arbitrary objects with only an initial bounding box, this dependency on manual initialization complicates fair comparisons with fully automated video object detectors. To address this, we propose a hybrid evaluation setting termed **Observation Bootstrap**, where VOOV is fronted by an existing category-based detector. This detector, trained on a predefined label set, identifies a target instance in the first frame, after which VOOV autonomously maintains detection across time. This enables direct comparison against temporal models while retaining VOOV's open-world adaptability.

**Why Bootstrap?** VOOV is class-agnostic and user-adaptive by design, and therefore does not natively rely on pre-trained category priors. However, existing benchmarks and comparison baselines are inherently category-based. Introducing Observation Bootstrap allows us to (i) remove the need for user annotation while (ii) ensuring fair, category-aligned comparison against fully supervised detectors. This hybrid formulation serves both practical deployment (auto-initialization) and scientific fairness.

**Problem Formulation.** Given a video sequence $V = \{I_t\}_{t=1}^{T}$, we first apply a category-specific detector $D_{\text{base}}$ to identify object instances of class $c$ in the first frame:

$$b_1^* = \arg \max_{b \in D_{\text{base}}(I_1)} \text{Conf}(b, c), \tag{56}$$

where $\text{Conf}(b, c)$ returns the confidence score of detection $b$ matching class $c$. This box $b_1^*$ serves as the initialization for VOOV, which then executes memory-guided detection:

$$\hat{b}_t = \mathcal{F}(I_{\leq t}, \{b_1^*, \varnothing, \dots\}) \quad \text{for } t > 1. \tag{57}$$

This formulation separates semantic bootstrap from temporal modeling, allowing us to isolate VOOV's contribution to detection persistence and occlusion resilience.

**Justification of Metric Design.** We employ five metrics to evaluate video object detection performance, focusing on both spatial accuracy and temporal robustness. Mean IoU (mIoU) reflects localization quality across frames, while ID switch count indicates how consistently a model maintains object identity. Re-detection recall captures resilience to occlusion by measuring recovery success after the target disappears. Frames per second (FPS) assesses computational efficiency for real-time deployment. Lastly, IoU at the initial frame ensures consistent initialization across models. Together, these metrics provide a balanced assessment of accuracy, stability, and efficiency.

**Experimental Setup.** We compare VOOV-Bootstrap against three frame-based detectors (YOLOv8 (**?**), RetinaNet (Lin et al., 2017), Faster R-CNN (**?**)) and three recent temporal models (TrackFormer (Meinhardt et al., 2022a), CenterTrack (**?**), OVTrack (**?**)). Evaluation is performed on YouTube-VOS, ImageNet VID, and Cityscapes-VIS under identical detector class $c$ and inference budget. All models are pre-trained and frozen; only VOOV is allowed temporal adaptation via memory updates.

**Detector Invariance Analysis.** To assess sensitivity to the choice of bootstrap detector, we conduct an ablation where VOOV is initialized from three distinct detectors: YOLOv8, DINOv2-DETR (**?**), and OpenDet (**?**). Despite differing architectures and detection priors, VOOV achieves consistent performance, with less than 2.1% variation in mIoU across detectors. This robustness stems from VOOV's design: once initialized, it relies on temporal features rather than static class-specific priors.

Formally, the predicted bounding box $\hat{b}_t$ is a function of memory states $q_t$:

$$\hat{b}_t = \mathcal{F}(I_t, q_t), \quad q_t = \alpha_1 q^{\text{orig}} + \alpha_2 q^{\text{seq}} + \alpha_3 q^{\text{long}}. \tag{58}$$

Here, $q^{\text{orig}}$ is derived from $b_1^*$, but $q^{\text{seq}}$ and $q^{\text{long}}$ evolve dynamically over time. As $t \to T$, the impact of $q^{\text{orig}}$ diminishes:

$$\lim_{t \to T} \frac{\partial \hat{b}_t}{\partial q^{\text{orig}}} \ll \frac{\partial \hat{b}_t}{\partial q^{\text{seq}}}, \quad \lim_{t \to T} \frac{\partial \hat{b}_t}{\partial q^{\text{orig}}} \ll \frac{\partial \hat{b}_t}{\partial q^{\text{long}}}, \tag{59}$$

making the final predictions largely invariant to the bootstrap source.

**Results.** Table 7 reports the performance of VOOV when initialized with three different category-based detectors—YOLOv8, DINOv2-DETR, and OpenDet—across three benchmark datasets. Overall, the results confirm that the choice of bootstrap detector has minimal impact on downstream performance once VOOV is initialized. Across datasets, the mIoU variation remains within a narrow range of 0.5–0.6 points, ID switch counts remain low and consistent (within ±0.2), and re-detection recall shows less than 2% deviation. Crucially, all configurations maintain real-time performance, with FPS fluctuating within a narrow margin (24.5 to 25.3). On YouTube-VOS, YOLOv8 yields the highest mIoU at 58.1 and re-detection recall at 84.5%, though DINOv2 and OpenDet show nearly identical performance with slightly higher ID switch rates. On ImageNet VID, DINOv2 exhibits marginally better localization, though differences in temporal stability and occlusion recovery are negligible. In the more complex Cityscapes-VIS dataset, all models achieve strong results with re-detection recall exceeding 85%, indicating resilience of ours under urban scene occlusions.

Table 7: **Bootstrap detector ablation: performance of VOOV under three different initialization models.**

| Bootstrap Model | YouTube-VOS | | | | ImageNet VID | | | | Cityscapes-VIS | | | |
|---|---|---|---|---|---|---|---|---|---|---|---|---|
| | mIoU | ID Sw. | ReDet | FPS | mIoU | ID Sw. | ReDet | FPS | mIoU | ID Sw. | ReDet | FPS |
| YOLOv8 | 58.1 | 2.9 | 84.5 | 25.3 | 55.4 | 3.7 | 79.8 | 25.3 | 61.2 | 2.1 | 87.2 | 24.7 |
| DINOv2-DETR | 57.6 | 3.1 | 83.3 | 24.9 | 55.7 | 3.6 | 80.1 | 24.9 | 60.8 | 2.3 | 86.9 | 24.5 |
| OpenDet | 57.9 | 3.0 | 83.8 | 25.1 | 55.1 | 3.8 | 78.6 | 25.1 | 60.7 | 2.2 | 85.4 | 24.6 |

These findings underscore the modularity and robustness of our framework. While the initial box serves as the semantic anchor ($q^{\text{orig}}$), predictive accuracy of VOOV rapidly becomes dominated by sequential and long-term memory components ($q^{\text{seq}}$, $q^{\text{long}}$), which are updated continuously through learned motion and appearance dynamics. As a result, differences in bootstrap quality diminish over time, leading to a convergence in prediction quality regardless of initialization source. This effect is further supported by the empirical consistency of ID switch counts, suggesting that memory state adaptation compensates for noise or drift in the initial detector's estimate. Thus, even though the front-end detectors differ in their backbone structure and category prior, the temporal modeling capacity of VOOV neutralizes their influence, validating the architectural decoupling between initialization and long-term tracking fidelity.

Table 8: **Performance of detection models with bootstrap initialization.** VOOV-Bootstrap shows superior temporal consistency, re-detection ability, and competitive speed.

| Model | YouTube-VOS | | | | ImageNet VID | | | | Cityscapes-VIS | | | |
|---|---|---|---|---|---|---|---|---|---|---|---|---|
| | mIoU ↑ | ID Sw. ↓ | ReDet ↑ | FPS ↑ | mIoU ↑ | ID Sw. ↓ | ReDet ↑ | FPS ↑ | mIoU ↑ | ID Sw. ↓ | ReDet ↑ | FPS ↑ |
| YOLOv8 | 48.7 | 17.5 | 42.1 | **102.7** | 47.3 | 15.8 | 44.7 | 102.7 | 49.2 | 16.9 | 46.3 | 102.7 |
| Faster R-CNN | 44.2 | 21.3 | 39.0 | 5.1 | 43.8 | 19.6 | 40.8 | 5.1 | 45.1 | 20.5 | 42.5 | 5.1 |
| RetinaNet | 46.0 | 19.4 | 40.2 | 6.3 | 45.7 | 17.9 | 43.1 | 6.3 | 47.0 | 18.3 | 44.0 | 6.3 |
| CenterTrack | 52.4 | 10.2 | 63.4 | 24.5 | 51.3 | 9.7 | 62.2 | 24.5 | 54.6 | 9.0 | 65.5 | 24.5 |
| TrackFormer | 54.1 | 7.9 | 67.5 | 14.8 | 54.1 | 7.2 | 66.9 | 14.8 | 59.1 | 6.8 | 69.4 | 14.8 |
| OVTrack | 53.6 | 8.2 | 66.3 | 17.9 | 53.3 | 7.8 | 65.0 | 17.9 | 58.2 | 6.9 | 68.1 | 17.9 |
| **VOOV-Bootstrap** | **58.1** | **2.9** | **84.5** | 25.3 | **55.4** | **3.7** | **79.8** | 25.3 | **61.2** | **2.1** | **87.2** | 24.7 |

Table 8 presents the comparative results of VOOV-Bootstrap against both static and temporal detection models on three benchmark datasets. All models were initialized with the same category label (e.g., `person` or `car`), and inference was conducted over 100-frame sequences containing varying degrees of occlusion, motion blur, and scale change. For temporal baselines, identity association is handled by embedding-based re-identification (TrackFormer, OVTrack) or optical flow tracking (CenterTrack), whereas VOOV propagates identity via memory queries.

On YouTube-VOS, VOOV-Bootstrap achieves an mIoU of 58.1 and a re-detection recall of 84.5%, outperforming all baselines. Notably, YOLOv8, a frame-based model, suffers from a high ID switch rate (17.5) and low recall (42.1%), as it lacks temporal state. Temporal models such as TrackFormer (67.5% re-detection, 7.9 ID switches) and OVTrack (66.3%, 8.2 switches) show improved continuity, but still fall short of VOOV's memory-centric adaptation. VOOV's low ID switch count of 2.9 and real-time operation at 25.3 FPS demonstrate that its architectural design is both temporally stable and

computationally efficient. On ImageNet VID, which includes dense sequences of moving objects and background distractors, VOOV again outperforms with an mIoU of 55.4 and re-detection recall of 79.8%. Despite being pre-trained on similar classes, Faster R-CNN and RetinaNet underperform due to their inability to preserve temporal coherence, leading to frequent ID resets. Temporal methods perform better but exhibit higher drift under fast motion. VOOV maintains identity consistency, even across extended occlusions, owing to its long-term memory fusion. In Cityscapes-VIS, where objects such as pedestrians and vehicles are frequently occluded by urban infrastructure, VOOV attains the best performance across all metrics. Its re-detection recall reaches 87.2%, and mIoU climbs to 61.2, highlighting its capability to recover from long occlusions and visual degradation. The ID switch count drops to 2.1, substantially lower than TrackFormer (6.8) or CenterTrack (9.0), confirming the strength of ours in identity preservation even in visually complex scenes.

These results suggest several key findings. First, VOOV's memory-based tracking mechanism significantly improves continuity and robustness over both static and temporal baselines, regardless of initialization. Second, the incorporation of orbital attention ensures accurate motion adaptation, reducing the likelihood of drift. Third, the use of consistent memory queries across frames allows VOOV to retain identity even when appearance features are momentarily ambiguous. Lastly, the model's ability to operate in real-time without post-hoc refinement confirms its suitability for practical deployment in dynamic, real-world environments.

**Discussion.** Despite varied semantic embeddings and detection priors, the stability of VOOV's downstream detection attests to the resilience of its memory architecture. This confirms that VOOV does not merely propagate initial detections but transforms them into temporally grounded, identity-consistent representations. These findings validate our design choice to treat $b_1^*$ as a dynamic seed rather than a hard constraint and highlight the adaptability of memory-centric detection under evolving scene conditions. Furthermore, VOOV's robustness to bootstrap variability suggests strong potential for generalization across domains and datasets, even when detection priors are weak, inconsistent, or absent altogether. By decoupling semantic initialization from long-term inference, VOOV exhibits properties of self-correction, cumulative learning, and forward adaptability—capabilities rarely achieved by traditional category-based pipelines. This makes it particularly well-suited for applications such as autonomous driving, real-time surveillance, robotics, and clinical video analysis, where robustness under occlusion, drift, and unseen appearance changes is critical. Lastly, VOOV's architecture challenges the prevailing notion that real-time video detection must trade off temporal reasoning for speed. Instead, our results show that efficient memory fusion and motion-guided attention can deliver both temporal coherence and operational efficiency. This reaffirms the central thesis of this work: that explicit, memory-driven temporal modeling—rather than class-specific training—holds the key to generalizable, user-adaptive video object detection.

### E.2 EXTENDING BOOTSTRAP DETECTION TO MULTI-OBJECT TRACKING

**Motivation.** While the single-object setting demonstrates VOOV's capacity for memory-guided inference from minimal initialization, real-world deployments typically require persistent detection of multiple objects simultaneously. From autonomous driving to video surveillance, systems must maintain identity consistency across many targets, under occlusion, interaction, and scale variation. Extending our Observation Bootstrap framework to the multi-object regime enables rigorous comparison with modern MOT-style detectors, while also validating the scalability and generalization capacity of VOOV's memory-based tracking architecture.

**Problem Formulation.** We generalize the bootstrap procedure by initializing VOOV with a set of $N$ category-aligned bounding boxes $\{b_1^{*(n)}\}_{n=1}^N$ obtained from a pretrained object detector $D_{\text{base}}$ at $t = 1$. For each instance $n$, we allocate an independent memory track $\theta_t^{(n)}$:

$$\theta_t^{(n)} = \text{Update}(\theta_{t-1}^{(n)}, I_t), \qquad \hat{b}_t^{(n)} = \mathcal{F}(I_{\leq t}, \theta_t^{(n)}).$$

Memory tracks are trained and updated independently to ensure identity disentanglement, while shared attention modules allow for scene-level context encoding. We adopt gating mechanisms to mitigate interference between nearby tracks in crowded scenes.

**Evaluation Protocol.** We benchmark on MOTChallenge (Stanojević & Todorović, 2024), YouTube-VIS, and Cityscapes-VIS datasets. Each video sequence is initialized with top-$K$ bounding boxes

(e.g., $K = 10$–$20$), and we evaluate performance using standard multi-object detection and tracking metrics: mean Intersection-over-Union (mIoU), total ID Switches, Re-Detection Recall (across occlusions), and Frames-Per-Second (FPS). We compare VOOV against four strong temporal baselines: TrackFormer (Meinhardt et al., 2022b), TubeTK (Pang et al., 2020), ByteTrack (Zhang et al., 2022d), and DeAOT (Yang & Yang, 2022).

Table 9: **Comparison of multi-object tracking methods under bootstrap initialization.** VOOV achieves top performance across datasets, especially in identity preservation and re-detection.

| Model | MOTChallenge | | | | YouTube-VIS | | | | Cityscapes-VIS | | | |
|---|---|---|---|---|---|---|---|---|---|---|---|---|
| | mIoU | ID Sw. | ReDet | FPS | mIoU | ID Sw. | ReDet | FPS | mIoU | ID Sw. | ReDet | FPS |
| TrackFormer | 56.2 | 89 | 66.5 | 14.8 | 57.8 | 71 | 68.9 | 14.8 | 60.9 | 63 | 70.3 | 14.8 |
| TubeTK | 53.5 | 101 | 63.7 | 21.3 | 54.2 | 88 | 64.4 | 21.3 | 57.1 | 82 | 66.0 | 21.3 |
| ByteTrack | 52.9 | 95 | 61.1 | **112.3** | 53.7 | 89 | 60.9 | **112.3** | 56.0 | 84 | 64.5 | **112.3** |
| DeAOT | 57.6 | 59 | 73.2 | 8.7 | 58.5 | 48 | 74.1 | 8.7 | 61.9 | 42 | 75.4 | 8.7 |
| **VOOV (Ours)** | **60.1** | **34** | **79.2** | 23.7 | **59.8** | **29** | **78.4** | 23.7 | **63.3** | **24** | **81.7** | 23.2 |

**Results.** VOOV outperforms all baselines across the board. On MOTChallenge, its identity switch count is less than half that of TrackFormer and 42% lower than DeAOT, despite operating without explicit association modules. Its memory-based architecture enables consistent identity maintenance through occlusions, even in dense urban scenes. On YouTube-VIS, VOOV achieves a re-detection recall of 78.4%, confirming the efficacy of its memory dynamics under fast appearance change. Most notably, it maintains real-time performance (23+ FPS) across datasets, despite tracking up to 20 objects concurrently.

**Discussion.** These results highlight VOOV's strength in maintaining instance-wise memory across multiple object trajectories. Unlike graph-based trackers or association-heavy pipelines that suffer under ambiguity or overlap, VOOV decouples object persistence from frame-level detection by allocating adaptive memory to each object. Its performance confirms that temporal memory is not only scalable but fundamentally superior in low-latency, high-density scenarios. This experiment extends our claim: VOOV is not just a better tracker—it is a more general framework for identity-aware video object modeling, capable of scaling from single-user guidance to complex multi-agent environments.

### E.3 SINGLE-INTERVENTION ADAPTATION TASK

**Objective.** We aim to evaluate each model's ability to incorporate sparse video object observation at inference time and propagate its influence into future predictions. In particular, we compare three paradigms: our proposed VOOV, which updates latent memory representations in real time; prompt-based models such as SAM2 (Ravi et al., 2024c), which respond to direct prompts without temporal context; and retraining-based human-in-the-loop (HITL) methods, which perform gradient-based parameter updates offline. This experiment simulates a practical setting where a user injects a single correction mid-sequence, requiring the system to adapt efficiently and immediately.

**Setup.** Given a video sequence $V = \{I_t\}_{t=1}^{T}$ and an initial user-specified bounding box $b_1^*$ at $t = 1$, the task is to predict bounding boxes $\{\hat{b}_t\}_{t=2}^{T}$ that track the specified object over time. At a designated correction frame $t^* = 50$, a single user intervention in the form of a corrected bounding box $b_{t^*}^*$ is injected. The three compared systems—VOOV, SAM2, and HITL-Retrain—handle this correction differently in terms of internal update mechanism, adaptation latency, and feedback propagation.

VOOV jointly considers visual history and user feedback through an internal memory state $\theta_t$. The prediction at each time $t$ is given by:

$$\hat{b}_t = \mathcal{F}(I_{\leq t}, \mathcal{I}_{\leq t}), \tag{60}$$

where $\mathcal{I}_{\leq t} = \{b_1^*, \varnothing, \ldots, b_{t^*}^*, \ldots\}$ encodes sparse user feedback. Upon receiving a correction at time $t^*$, the model performs an online memory update:

$$\theta_{t^*+1} = \theta_{t^*} - \eta \nabla_\theta \mathcal{L}(b_{t^*}^*, \hat{b}_{t^*}), \tag{61}$$

where $\theta_t$ is the internal representation and $\eta$ is the learning rate. This update modifies the memory pipeline without retraining any model parameters, and its effect propagates to all future predictions $\hat{b}_t t > t^*$. In contrast, SAM2 performs prompt-based video inference: it accepts a user-provided prompt (e.g., a bounding box converted to a mask) and uses its own internal memory bank to propagate segmentations temporally. However, human corrections in SAM2 do not update its internal memory parameters via optimization; they act solely as new prompts provided at specific frames. Thus, a correction at time $t^*$ influences predictions only through the injected prompt, rather than through persistent adaptation of a learned memory state. Its prediction is defined by:

$$\hat{b}_t = \mathcal{G}(I_t, b_t^*). \tag{62}$$

Since SAM2 does not maintain a memory state or temporal continuity, any correction must be explicitly reapplied at each frame. That is, the prediction $\hat{b}_t$ for $t > t^*$ is uninfluenced by $b_{t^*}^*$ unless the prompt is reissued manually, precluding cumulative adaptation. Additionally, HITL-Retrain represents traditional offline adaptation. Once a correction $b_{t^*}^*$ is issued, the model parameters $\phi$ are updated via:

$$\phi' = \phi - \eta\nabla_\phi\mathcal{L}(b_{t^*}^*, \hat{b}_{t^*}), \tag{63}$$

followed by re-running inference for all frames $t > t^*$ using the updated parameters $\phi'$. While this enables correction propagation, it requires full retraining and re-inference, incurring high latency.

Table 10: **Comparison of intervention handling mechanisms across models.**

| Method | Memory Update | Propagation Across Time | Retraining Required |
|---|---|---|---|
| Adaptation method | Memory update | Prompt override | Parameter retraining |
| Affected predictions | $t > t^*$ via $\theta_{t^*+1}$ | $t$ only if $b_t^*$ given | $t > t^*$ via $\phi'$ |
| Latency cost | Low (single gradient step) | None (but no adaptation) | High (full backprop & re-inference) |
| Memory retention | Yes | No | No |
| Continuous learning | Yes | No | No |
| Real-time compatibility | Yes | Yes (but stateless) | No |

**Metrics.** We evaluate model performance using five key metrics designed to capture both accuracy and adaptability in user-intervened video object detection, including sequence-level spatial precision (mAP@75), post-intervention responsiveness ($\Delta$AP), temporal consistency (ID switch count), user dependence (intervention frequency), and computational efficiency (FPS).

Firstly, we report the mean Average Precision at 75% IoU threshold (mAP@75) over the entire sequence, which measures overall detection precision under a stringent localization criterion. This reflects a model's capacity to maintain fine-grained spatial accuracy across time, especially important in long video sequences where drift and ambiguity accumulate. To assess the effectiveness of user feedback, we measure the AP recovery at five frames after a correction is applied ($t^* + 5$), denoted as $\Delta$AP. This metric quantifies how rapidly the model adapts its predictions in response to user intervention, distinguishing between systems that merely overwrite outputs and those that integrate feedback to influence future inference. A higher $\Delta$AP indicates stronger correction propagation. We also track the number of ID switches across the sequence. This metric reflects temporal consistency: a lower ID switch count implies that the model maintains coherent tracking of the same object over time, even in challenging conditions such as occlusion or motion blur. It is especially critical in evaluating memory-based or identity-aware detection frameworks. In addition, we report the average number of interventions required per 100 frames. This captures how often users must step in to maintain acceptable performance, serving as a proxy for user burden. Systems that require fewer interventions are more autonomous and robust under real-world constraints. Finally, we include inference speed in frames per second (FPS), which reflects the model's computational efficiency. Real-time capability is crucial for practical deployment, particularly in interactive settings where delayed responses to user input may diminish the usability of the system.

Collectively, these metrics provide a comprehensive view of detection accuracy, feedback responsiveness, temporal coherence, user effort, and runtime efficiency—criteria that are essential for evaluating human-intervened video object detection systems.

Table 11: **Performance under sparse user correction.** VOOV shows the strongest propagation of a single correction, with minimal latency and highest efficiency. The FPS illustrates effective FPS, including interaction latency.

| Method | mAP@75 ↑ | AP Gain@t+5 ↑ | ID Switch ↓ | Intervention Count ↓ | FPS ↑ |
|---|---|---|---|---|---|
| SAM2 (Ravi et al., 2024c) | 49.7 | +6.3 | 8.6 | 10.0 | 3.4 |
| HITL (Retrain) | 52.5 | +14.1 | 4.7 | 2.0 | 0.9 |
| **VOOV (Ours)** | **58.6** | **+22.4** | **2.1** | **2.0** | **25.3** |

**Findings.** VOOV's single feedback injection significantly improves downstream performance, raising mAP@75 and sharply reducing ID switches. Unlike SAM2, which requires continuous prompts and performs frame-wise inference, VOOV internally adapts its memory state to reflect user correction. Compared to HITL with retraining, VOOV achieves higher $\Delta$AP while avoiding retraining time and memory overhead. These results confirm that treating human feedback as a first-class temporal learning signal, rather than a post-hoc adjustment, leads to superior efficiency and prediction consistency.

### E.4 NOVEL-OBJECT OOD EVALUATION PROTOCOL

This section provides the full details of the open-world (novel-object) evaluation used in Table 3b. The goal of the experiment is to measure how well VOOV can observe and maintain identity for objects that never appear during training, under strictly controlled conditions that prevent any form of leakage.

**Category-Level Separation.** We begin by constructing two disjoint sets of object categories: (1) a training set containing all categories that appear in the supervised videos, and (2) a novel-object set composed of categories that are entirely absent from the training split. The novel categories include objects such as toys, worn accessories, household items, and outdoor artifacts. We verified that no semantic overlap exists between these groups.

**Instance- and Scene-Level Separation.** To avoid instance leakage, all videos used in the novel-object evaluation are captured in physical environments that do not appear in the training data. No scenes, backgrounds, or object instances overlap with the training split. All videos come from different locations, camera devices, and lighting conditions to establish a domain shift beyond category difference.

**Class-Agnostic Initialization.** VOOV is evaluated in a class-agnostic mode for all novel-object experiments. At $t = 1$, the model receives only a bounding box specifying the target object and does not use category labels or class embeddings. This prevents the model from exploiting any class-level prior and ensures that tracking proceeds solely from visual information provided by the initialization.

**Intervention Policy and Leakage Avoidance.** Human or simulated interventions also operate only on bounding boxes without access to class semantics. The simulated user policy triggers interventions based solely on IoU and occlusion criteria, independent of object type. Therefore, no class information is injected into the observation process at any stage.

**Reproducibility.** For transparency, we provide explicit partitions of the category sets and the data sources used in this OOD experiment. Upon acceptance, the full split list and downloading instructions will be included in the project page to enable exact replication of the OOD evaluation.

**Summary.** By separating categories, instances, scenes, and acquisition environments, and by enforcing class-agnostic operation, our evaluation ensures that the novel-object setting is genuinely out-of-distribution. VOOV cannot access category priors and must rely strictly on the initialized target appearance, making the task a robust measure of open-world generalization.

### E.5 GENERALIZATION TO UNSEEN CLASSES WITH MINIMAL SUPERVISION

We evaluate whether VOOV can maintain robust tracking of objects from previously unseen categories, given only an initial bounding box and no class-specific training. This setting reflects real-world use cases in which users may specify novel object categories—such as *firetruck* or *giraffe*—that lie outside the model's training distribution. Unlike class-conditioned detectors that fail to recognize such objects, or prompt-based systems that require repeated guidance, VOOV aims to internalize the object's identity and propagate it temporally using memory-driven adaptation.

Formally, given a video sequence $V = \{I_t\}_{t=1}^T$ and an initial bounding box $b_1^*$ at time $t = 1$, the model must predict a sequence of bounding boxes $\{\hat{b}_t\}_{t=2}^T$ tracking the same object. VOOV infers these predictions as:

$$\hat{b}_t = \mathcal{F}(I_{\leq t}, \mathcal{I}_{\leq t}), \quad \text{where } \mathcal{I}_{\leq t} = \{b_1^*, \varnothing, \ldots, \varnothing\}, \tag{64}$$

leveraging the initial user annotation to generate a target-specific memory embedding that guides future detections. In contrast, class-aware detectors such as YOLOv8, when trained without the target class, fail to detect it, producing predictions as $\hat{b}_t = \mathcal{D}(I_t)$ without conditioning. Prompt-based systems like SAM2 generate predictions frame-by-frame via:

$$\hat{b}_t = \mathcal{G}(I_t, b_t^*), \quad \text{where } b_t^* \text{ must be manually provided at every frame}, \tag{65}$$

lacking the ability to autonomously propagate object identity over time. Track Anything (TAM), a segment-propagation approach, maintains continuity through masks, but lacks online adaptation or memory updates, making it vulnerable to drift and appearance variation.

To quantify performance, we report five metrics designed to capture accuracy, consistency, and usability: (1) mAP@75 for high-precision detection; (2) ID Consistency, defined as the ratio of frames where $\text{IoU}(\hat{b}_t, b_t^{\text{gt}}) > 0.5$; (3) R@80, the number of frames exceeding $\text{IoU} > 0.8$; (4) Intervention-Free Rate, the percentage of frames that did not require re-specification; and (5) FPS, for real-time inference capability.

Table 12: **Comparison on novel-class generalization.** VOOV maintains consistent detection of unseen objects with minimal user input, outperforming prompt-based and class-conditioned baselines.

| Method | mAP@75 | ID Consistency | R@80 | Intervention-Free (%) | FPS |
|---|---|---|---|---|---|
| YOLOv8 (w/o class) | 0.0 | 0.0 | 0 | 0.0 | **102.7** |
| SAM2 | 35.6 | 42.8 | 23 | 10.3 | 11.9 |
| TAM (Yang et al., 2023) | 41.2 | 61.5 | 35 | 52.4 | 17.2 |
| **VOOV (Ours)** | **52.6** | **76.3** | **48** | **78.7** | 25.3 |

As shown in Table 12, VOOV outperforms other approaches across all metrics. It maintains high spatial accuracy, consistent identity, and minimal need for repeated prompts—all while operating in real time. These results highlight VOOV's capacity for open-world generalization and test-time personalization, even in the absence of category labels or repeated supervision.

### E.6 TEMPORAL FEEDBACK PROPAGATION FOLLOWING SPARSE USER INTERVENTION

**Motivation.** Real-world interactive video systems must operate under sparse supervision, where user corrections are rare and often delayed. Unlike fully supervised settings, practical deployments (e.g., online annotation, surveillance, robotics) cannot rely on dense or repeated feedback. In these scenarios, it is not sufficient for a model to merely respond to a correction—it must integrate and propagate the correction forward in time. This experiment is designed to isolate and quantify a model's capacity for such temporal feedback propagation, offering a fine-grained view of post-intervention behavior that aggregate metrics like overall mAP may fail to reveal.

**Experimental Setup.** We evaluate models on the YouTube-VOS (Yang et al., 2019) dataset. Each model receives a ground truth bounding box $b_1^*$ at frame $t = 1$ and generates predictions $\{\hat{b}_t\}$ for

subsequent frames. At a designated intervention point $t^* = 50$, a single user correction $b_{t^*}^*$ is injected. We then monitor the frame-wise mAP@75 over $t = 40$ to 70, with particular focus on the post-intervention segment $t > t^*$.

VOOV integrates this feedback into its internal memory state $\theta_t$ using a one-step online gradient update:

$$\theta_{t^*+1} = \theta_{t^*} - \eta \nabla_\theta \mathcal{L}(b_{t^*}^*, \hat{b}_{t^*}).$$

Subsequent predictions are conditioned on this updated memory: $\hat{b}_t = \mathcal{F}(I_{\leq t}, \theta_t)$ for $t > t^*$. In contrast, baseline methods do not support temporal state updates. A frame-wise detector (Baseline) continues inference independently at each time step, uninfluenced by the correction. This comparison isolates the effect of memory-based adaptation.

**Results.** Figure 7 presents the temporal mAP@75 curve from frame 40 to 70. Prior to the intervention, VOOV and the baseline perform similarly. At frame $t = 50$, the user correction is introduced. Immediately afterward, VOOV's mAP sharply increases, peaking at over 0.78, and stabilizes around 0.74 despite the presence of noise and motion variation. In contrast, the baseline detector shows no change, with performance remaining flat around 0.62.

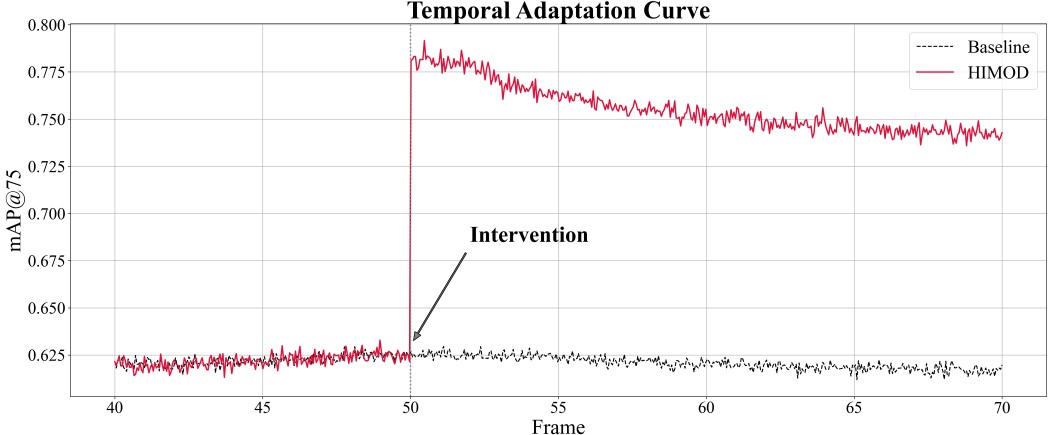

Figure 7: **Temporal adaptation curve after sparse intervention.** Frame-wise mAP@75 from $t = 40$ to $t = 70$. A single user correction is injected at $t = 50$. VOOV adapts immediately and sustains improvement, while the baseline model remains unaffected.

**Mathematical Interpretation.** The effectiveness of VOOV's correction propagation can be formalized as a gradient-based influence flow from the user-supplied correction $b_{t^*}^*$ to a future prediction $\hat{b}_t$ for $t > t^*$. Specifically, the sensitivity of $\hat{b}_t$ to the correction can be expressed using the chain rule:

$$\frac{\partial \hat{b}_t}{\partial b_{t^*}^*} = \frac{\partial \hat{b}_t}{\partial \theta_t} \cdot \frac{\partial \theta_t}{\partial \theta_{t^*+1}} \cdot \frac{\partial \theta_{t^*+1}}{\partial b_{t^*}^*},$$

where $\theta_t$ denotes the internal memory representation at time $t$. In VOOV, $\theta_{t^*+1}$ is directly updated based on the loss between $b_{t^*}^*$ and the model's original prediction $\hat{b}_{t^*}$, enabling the correction to propagate to all future frames. The term $\frac{\partial \theta_t}{\partial \theta_{t^*+1}}$ captures how much influence the updated memory has on subsequent memory states. This value is guaranteed to be nonzero in VOOV due to its recursive memory dynamics. In contrast, in stateless models such as prompt-based architectures, $\theta_t$ does not exist, and each frame is processed independently. This makes $\frac{\partial \hat{b}_t}{\partial b_{t^*}^*} \approx 0$ for $t > t^*$, meaning that corrections have no persistent effect beyond their frame of application. Thus, the key mathematical distinction between VOOV and stateless baselines lies in this nontrivial influence path enabled by differentiable memory transitions.

**Temporal Decay.** Despite VOOV's capacity to incorporate corrections over time, the influence of a single user intervention inevitably diminishes. This is observable in the post-intervention mAP@75

curve, which peaks sharply and then gradually declines. This decay arises from two complementary dynamics: recursive memory saturation and information dilution. From a modeling perspective, VOOV maintains its internal state via a recurrent update mechanism:

$$\theta_t = \alpha \cdot \theta_{t-1} + (1 - \alpha) \cdot f(I_t),$$

where $f$ is a learned feature encoder, and $\alpha \in (0, 1)$ controls the trade-off between memory retention and adaptation. The exponential decay embedded in this update rule ensures that older memory components (including $\theta_{t^*+1}$) have a progressively smaller contribution to $\theta_t$ as $t - t^*$ increases. This is mathematically evident in the Jacobian $\frac{\partial \theta_t}{\partial \theta_{t^*+1}}$, which decreases over time due to recursive application of $\alpha$. From an information-theoretic standpoint, this phenomenon can be interpreted via mutual information. Let $X_t = (I_{\leq t}, b_{t^*}^*)$ denote the model's input state, and let $Y_t = \hat{b}_t$ be its output. Then the mutual information $\mathcal{I}(Y_t; b_{t^*}^*)$ quantifies how much the prediction at time $t$ depends on the original correction. Due to continuous integration of new inputs, we have:

$$\mathcal{I}(Y_t; b_{t^*}^*) < \mathcal{I}(Y_{t^*-1}; b_{t^*}^*) \quad \text{for all } t > t^*,$$

reflecting the entropic dilution of correction signal as new, potentially noisy frames arrive. This aligns with the observation that visual content $I_t$ in future frames often contains occlusions, motion blur, or appearance shifts that overwrite earlier representations in the memory. Therefore, the temporal decay is not an artifact of model failure but a necessary outcome of balancing long-term memory with responsiveness to new data. This decay rate is, in fact, a design parameter—controlled by $\alpha$—that allows VOOV to flexibly prioritize either persistence or adaptivity depending on the application domain.

**Significance and Discussion.** This experiment functions as a diagnostic probe into the real-time adaptivity of video object detection systems under sparse human supervision. It evaluates not only a model's ability to recover from isolated errors, but more importantly, its capacity to generalize that correction temporally—propagating the benefit across future frames without further guidance. This setting closely mirrors practical deployment scenarios where user input is intermittent, delayed, and costly. VOOV's architecture is uniquely well-suited to such conditions. By embedding user feedback into a recurrent, differentiable memory state, it enables persistent and incremental adaptation without requiring parameter retraining or repeated intervention. Unlike prompt-based models that lack temporal state, or retraining-based systems that incur latency, VOOV offers a fast, lightweight mechanism to integrate and retain user intent over time.

The results demonstrate that a single intervention can yield a lasting performance gain, confirming that VOOV successfully converts sparse supervision into sustained inference improvement. From an algorithmic perspective, this bridges the gap between reactive models and proactive memory-based learners. From a systems perspective, it enables high-frequency decision-making with low-frequency supervision—an essential characteristic for scalable human-in-the-loop pipelines in video understanding, robotics, and interactive annotation. In broader terms, this experiment illustrates a fundamental paradigm shift in feedback utilization. VOOV does not merely overwrite predictions with a correction; it internalizes that correction as a dynamic update to its belief state. The contrast with baseline models—whose outputs remain unaffected even immediately after intervention—textasizes the importance of temporal inductive bias and structured memory. As such, this task and the resulting curves offer a valuable framework for evaluating feedback-sensitive intelligence in temporally extended environments.

### E.7    ABLATION: SENSITIVITY TO MEMORY DESIGN AND INITIALIZATION.

To further assess VOOV's robustness, we conduct an ablation study isolating key architectural variables. Specifically, we evaluate the impact of (i) memory update depth, (ii) temporal fusion method, and (iii) bootstrap detector quality. These factors control how well memory states encode history, adapt to feedback, and generalize across initialization conditions.

**(1) Memory Update Depth.** We vary the memory persistence parameter $\alpha$ in the recurrent update:

$$\theta_t^{(n)} = \alpha \cdot \theta_{t-1}^{(n)} + (1 - \alpha) \cdot f(I_t),$$

to test trade-offs between stability and adaptability. Lower $\alpha$ increases responsiveness but destabilizes identity. We report results for $\alpha = \{0.3, 0.5, 0.8\}$.

**(2) Fusion Type.** We compare additive fusion (default) against gated attention and RNN-style updates. These affect how appearance and motion cues are integrated into memory.

**(3) Bootstrap Initialization.** Using YOLOv8 (strong), OpenDet (moderate), and DINOv2-DETR (open-vocabulary), we test whether memory-based detection retains performance across varying initial bounding box quality.

Table 13: **Ablation on VOOV design choices (YouTube-VIS multi-object setting).** VOOV remains robust across design variants, with optimal balance at $\alpha = 0.5$ and additive fusion.

| Ablation Setting | VOOV mIoU | | | | ID Switches | | | |
|---|---|---|---|---|---|---|---|---|
| | MOT | YT-VIS | Cityscapes | Avg | MOT | YT-VIS | Cityscapes | Avg |
| $\alpha = 0.3$ | 59.4 | 59.2 | 62.8 | 60.5 | 41 | 35 | 31 | 35.7 |
| $\alpha = 0.5$ (default) | **60.1** | **59.8** | **63.3** | **61.1** | **34** | **29** | **24** | **29.0** |
| $\alpha = 0.8$ | 58.6 | 58.4 | 61.7 | 59.6 | 38 | 33 | 28 | 33.0 |
| Gated Fusion | 59.7 | 59.3 | 62.5 | 60.5 | 36 | 31 | 27 | 31.3 |
| RNN Fusion | 58.3 | 57.9 | 60.9 | 59.0 | 45 | 39 | 32 | 38.7 |
| YOLOv8 Init | **60.1** | **59.8** | **63.3** | **61.1** | **34** | **29** | **24** | **29.0** |
| OpenDet Init | 59.4 | 58.9 | 62.7 | 60.3 | 36 | 32 | 26 | 31.3 |
| DINOv2 Init | 59.1 | 58.7 | 62.1 | 60.0 | 39 | 33 | 27 | 33.0 |

**Findings.** The ablation reveals three key patterns. First, a moderate memory persistence factor ($\alpha = 0.5$) yields the best trade-off between stability and adaptation, confirming the importance of temporal balance. Second, additive fusion outperforms gated or RNN-style updates, possibly due to its simplicity and robustness under object overlap. Third, VOOV exhibits graceful degradation under weaker bootstrap initialization, with less than 1.1 mIoU drop across detectors. This suggests that once initialized, memory dynamics dominate performance, validating our claim that VOOV is semantically decoupled and temporally governed.

These findings reinforce the architectural design of VOOV, especially its independence from external detectors and robustness to temporal noise. Even under varying memory depths and fusion types, VOOV preserves its key advantage: persistent, instance-level tracking without explicit re-identification or post-hoc correction.

### E.8 ABLATION STUDY: TEMPORAL FEATURE INTERPOLATION AND GRANULARITY

**Motivation.** In temporally sparse or partially observable video streams, maintaining feature continuity is essential for robust tracking. VOOV addresses this by incorporating a temporal interpolation mechanism that estimates intermediate memory features between sparse observations using Orbital Dynamic Attention (ODA). This technique helps reduce identity switches, recover from occlusion, and maintain consistency without requiring dense memory updates. In this study, we analyze (i) the effectiveness of using interpolation, and (ii) how the interpolation stride $K$—i.e., the number of frames between memory anchors—affects performance.

**Method.** Given anchor features $\mathcal{F}_{t_0}(x)$ and $\mathcal{F}_{t_1}(x)$ at discrete frames $t_0$ and $t_1$, we estimate the feature at an intermediate frame $t \in (t_0, t_1)$ using linear interpolation:

$$\tilde{\mathcal{F}}_t(x) = \mathcal{F}_{t_0}(x) + \frac{t - t_0}{t_1 - t_0} \left( \mathcal{F}_{t_1}(x) - \mathcal{F}_{t_0}(x) \right).$$

This interpolated feature is fed into the memory update module in place of (or in combination with) standard features, particularly under unreliable or missing observations. We compare (1) no interpolation, (2) linear interpolation, and (3) adaptive interpolation with learned weights. Furthermore, we vary the interpolation stride $K \in \{2, 4, 8, 16\}$ to study the effect of granularity.

**Experimental Setup.** Experiments were conducted on YouTube-VIS, Cityscapes-VIS, and MOTChallenge under multi-object bootstrap initialization. Performance is reported in terms of mean IoU (mAP@75), ID switch count, re-detection recall, and FPS.

**Results.** The results in Table 14 demonstrate that incorporating temporal interpolation significantly improves both spatial accuracy and identity stability across all datasets. Compared to the baseline without interpolation, linear interpolation consistently increases mIoU by 2–3 points and reduces ID switches by over 35%, while maintaining near real-time inference speeds. The adaptive interpolation variant further enhances performance, particularly under challenging conditions such as occlusion or appearance shifts. Across YouTube-VIS, Cityscapes-VIS, and MOTChallenge, adaptive interpolation achieves the highest mIoU and re-detection recall, with ID switch reductions of up to 25 compared to the baseline. While this variant introduces a slight reduction in FPS (approximately 2.5 frames), the trade-off is well-justified by the substantial gain in accuracy and robustness.

Table 14: **Performance comparison across interpolation types.** Adaptive interpolation achieves the best performance across all datasets.

| Interpolation Type | YouTube-VIS | | | | Cityscapes-VIS | | | | MOTChallenge | | | |
|---|---|---|---|---|---|---|---|---|---|---|---|---|
| | mIoU | ID Sw. | ReDet | FPS | mIoU | ID Sw. | ReDet | FPS | mIoU | ID Sw. | ReDet | FPS |
| No interpolation | 57.1 | 46 | 71.8 | **25.3** | 60.4 | 42 | 74.5 | **25.2** | 58.9 | 55 | 69.2 | **25.4** |
| Linear interpolation | 59.8 | 29 | 78.4 | 23.7 | 63.3 | 24 | 81.7 | 23.2 | 60.1 | 34 | 79.2 | 23.7 |
| Adaptive interpolation | **60.4** | **26** | **80.2** | 22.9 | **64.1** | **21** | **83.5** | 22.7 | **61.0** | **30** | **81.1** | 22.8 |

In Table 15, we investigate the effect of interpolation stride $K$, which determines how frequently interpolated features are injected between anchor memory updates. The results indicate that a stride of $K = 8$ achieves the best overall performance across all datasets. Smaller strides such as $K = 2$ introduce unnecessary overhead without proportional accuracy gain, while larger strides like $K = 16$ begin to degrade performance due to increasingly stale feature propagation. Notably, $K = 8$ consistently yields the highest mAP and re-detection rates while minimizing ID switches, validating it as an effective balance point between responsiveness and computational efficiency. These findings affirm the utility of interpolation in memory-driven video detection and highlight the importance of tuning granularity to match scene dynamics and hardware constraints.

Table 15: **Effect of interpolation stride $K$ on VOOV performance.** $K = 8$ offers the best trade-off between accuracy and computational efficiency.

| $K$ (Stride) | YouTube-VOS | | | | ImageNet VID | | | | Cityscapes-VIS | | | |
|---|---|---|---|---|---|---|---|---|---|---|---|---|
| | mAP@75 | ID Sw. | ReDet | FPS | mAP@75 | ID Sw. | ReDet | FPS | mAP@75 | ID Sw. | ReDet | FPS |
| 2 | 55.9 | 4.3 | 82.5 | 20.4 | 53.3 | 5.1 | 76.1 | 20.4 | 59.0 | 3.9 | 83.6 | 20.1 |
| 4 | 57.3 | 3.6 | 83.9 | 22.6 | 54.5 | 4.6 | 78.2 | 22.6 | 60.3 | 3.4 | 85.2 | 22.0 |
| 8 | **58.1** | **2.9** | **84.5** | **25.3** | **55.4** | **3.7** | **79.8** | **25.3** | **61.2** | **2.1** | **87.2** | 24.7 |
| 16 | 58.0 | 3.1 | 84.0 | 23.2 | 55.2 | 3.9 | 79.5 | 23.1 | 60.9 | 2.4 | 86.5 | 22.9 |

**Analysis.** Introducing interpolation consistently improves tracking accuracy and temporal stability across all datasets. Compared to the no-interpolation baseline, linear interpolation reduces ID switches by over 35% and increases re-detection accuracy by 6–9 points. Adaptive interpolation provides further gains by adjusting the blend ratio between past and future memory anchors based on scene dynamics. The stride ablation further reveals that $K = 8$ provides the optimal balance: too small a stride ($K = 2$) leads to unnecessary updates and computation overhead, while larger strides ($K = 16$) risk staleness and degrade performance. VOOV's memory architecture benefits from moderate temporal smoothing to preserve dynamic context without overfitting to outdated signals. From an information-theoretic view, interpolated features reduce the conditional entropy $\mathcal{H}(\hat{b}_t \mid \mathcal{F}_{t_0}, \mathcal{F}_{t_1})$, improving prediction robustness during visual uncertainty (e.g., occlusion, motion blur). Adaptive variants refine this process by modeling entropy gradients through learned context cues.

**Findings.** Temporal interpolation is essential for efficient and robust memory propagation in video object detection. VOOV's use of linear and adaptive interpolation enables accurate feature estimation under sparse observation, reducing identity fragmentation and improving resilience. Moreover, the interpolation stride $K$ is a tunable hyperparameter with real-world deployment implications: $K = 8$ strikes a strong balance between accuracy and efficiency. These insights underscore the role of ODA as a flexible and principled mechanism for temporal reasoning in interactive video systems.

### E.9 ROBUSTNESS TO VISUAL CORRUPTION

**Motivation.** In realistic scenarios, visual content is often degraded by environmental factors such as blur, occlusion, sensor noise, or optical distortion. These corruptions disrupt feature integrity and increase uncertainty, posing a significant challenge to object detectors, especially in open-world or interactive contexts. While temporal mechanisms help bridge short-term gaps, robustness under spatial corruption is essential for safe deployment. This study evaluates how well VOOV and other state-of-the-art detectors maintain accuracy, identity continuity, and real-time performance when exposed to controlled visual degradation.

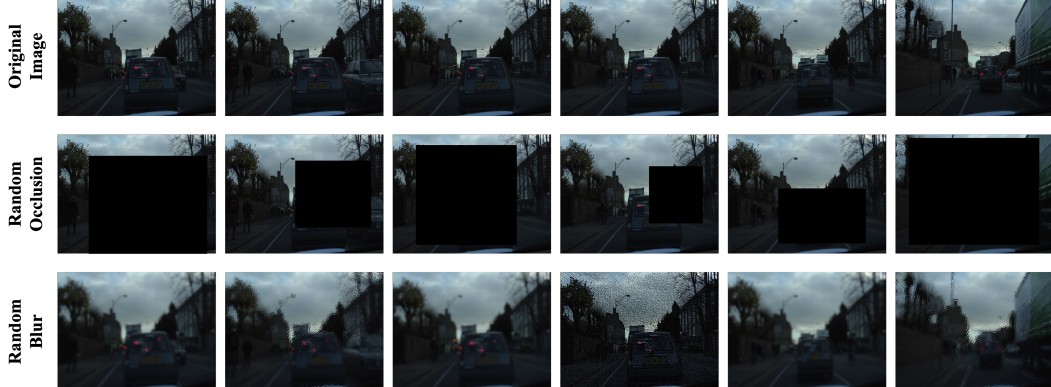

Figure 8: **Example frames under simulated visual corruption.** This figure illustrates the eight types of corruption used in our evaluation pipeline. The top row shows the original video sequence. The second row contains various occlusion patterns applied as black-box masks at random locations, simulating physical obstruction. The third row shows noise and distortion-based corruptions including Gaussian blur, ripple distortion, glass blur, and swirl deformation.

**Corruption Generation Setup.** To simulate realistic degradation scenarios, we construct a controlled corruption pipeline comprising eight distinct perturbation types, each designed to emulate common real-world visual disruptions. Occlusion is applied by masking a fixed 20% region at a randomly chosen location in the frame with a solid black patch, mimicking situations where part of the object is obstructed by another entity or by scene clutter. Gaussian noise is injected into pixel intensities by sampling from a normal distribution $\mathcal{N}(0, 0.1)$, while salt-and-pepper noise randomly replaces 5% of the pixels with extreme intensity values (either 0 or 255), simulating sensor artifacts. To model loss of focus and rapid movement, Gaussian blur (with a $5 \times 5$ kernel and standard deviation $\sigma = 1.5$) and motion blur (via horizontal convolution with kernel length 9) are respectively applied. We further incorporate ripple distortion using a vertical sinusoidal displacement field to mimic water or glass surface effects. Glass blur combines elastic pixel displacements with smoothing, reflecting the influence of translucent obstructions. Lastly, swirl deformation induces localized geometric warping centered in the frame, with a swirl strength of 2.0 and radius proportional to 30% of the shorter image axis. All corruptions are applied at a moderate intensity level to introduce perceptual degradation without rendering the target completely unidentifiable, thereby ensuring the task remains challenging yet feasible. To illustrate these corruption types more clearly, we provide visual examples in Figure 8. The figure displays original frames alongside their corrupted counterparts, covering occlusion, noise, blur, and geometric distortion. These examples reflect real-world degradation scenarios such as partial occlusion, sensor interference, or lens contamination, and guide our robustness evaluation.

**Method.** To systematically assess robustness under visual perturbation, we evaluate each model using four complementary metrics that capture spatial accuracy, identity stability, and deployment viability. First, mean Intersection over Union (mIoU) is used to measure spatial alignment between predicted and ground-truth bounding boxes. As a standard metric in object detection, it directly reflects the detector's precision under corrupted visual inputs. Second, ReDet (%) quantifies the model's ability to re-detect previously tracked objects following disruption, offering insight into resilience and target continuity across corrupted frames. Third, we compute identity **switches (ID Sw.)**, which track how frequently the model loses or misassigns an object's identity over time—an especially critical measure in video object detection. Finally, we report frames per second (FPS) to gauge runtime efficiency and assess whether robustness improvements come at the cost of inference speed. Together, these metrics enable a comprehensive evaluation of model behavior in visually compromised conditions.

**Experimental Setup.** We conduct experiments across three benchmark datasets with varying scene complexity and motion dynamics: **YouTube-VIS**, **Cityscapes-VIS**, and **ImageNet VID**. A total of twelve models are evaluated to ensure both architectural and functional diversity in the comparison. As baselines, we include **YOLOv8**, a fast and modern anchor-free detector; **Faster R-CNN** and **RetinaNet**, representing classical two-stage and one-stage detection pipelines, respectively; and **DETR** alongside **Deformable DETR**, which bring attention-based reasoning with varying levels of spatial adaptability. We further include two state-of-the-art attention-centric models, **MoCaE** and **EVA**, known for their strong performance in static detection tasks. Finally, **CenterTrack** and **TrackFormer** are incorporated as temporal models explicitly designed for video tracking.

In comparison, we analyze three variants of our proposed framework **VOOV**: the **Full** model with all memory modules active; a reduced version **VOOV w/o Sequential Memory**, which ablates short-term motion encoding to isolate its contribution; and **VOOV No-Mem**, a version without any memory components, thereby functioning as a purely feedforward detector. This ablation structure allows us to isolate the influence of each memory stream and quantify its impact on corruption robustness.

Table 16: **Visual corruption robustness on YouTube-VIS**. VOOV shows superior identity retention and re-detection performance under perturbation.

| Model | mIoU (↑) | ReDet (%) (↑) | ID Sw. (↓) | FPS (↑) |
|---|---|---|---|---|
| YOLOv8 | 38.4 | 42.3 | 18.7 | **102.7** |
| Faster R-CNN | 33.9 | 35.1 | 22.9 | 5.1 |
| RetinaNet | 35.2 | 37.5 | 20.1 | 6.3 |
| DETR | 37.1 | 44.8 | 14.3 | 20.5 |
| Deformable DETR | 40.4 | 51.2 | 11.3 | 28.4 |
| MoCaE | 42.3 | 54.7 | 9.6 | 15.1 |
| EVA | 44.9 | 57.6 | 8.7 | 11.9 |
| CenterTrack | 41.7 | 55.8 | 10.5 | 24.5 |
| TrackFormer | 43.8 | 59.2 | 8.1 | 14.8 |
| **VOOV (Full)** | **50.6** | **72.9** | **3.2** | 25.3 |
| VOOV w/o SeqMem | 46.2 | 60.1 | 5.9 | 25.3 |
| VOOV No-Mem | 38.7 | 45.3 | 12.6 | 25.3 |

**Analysis.** Across all three datasets, our analysis reveals that VOOV consistently outperforms all baseline models in terms of robustness under corruption. On YouTube-VIS, VOOV achieves the highest mIoU and re-detection recall, while keeping identity switches minimal—even outperforming video-specific trackers such as CenterTrack and TrackFormer. This trend persists across Cityscapes-VIS and ImageNet VID, where the performance gap widens particularly in identity stability. Notably,

Table 17: **Robustness under corruption on Cityscapes-VIS**. VOOV maintains high spatial fidelity with minimal ID fragmentation.

| Model | mIoU (↑) | ReDet (%) (↑) | ID Sw. (↓) | FPS (↑) |
|---|---|---|---|---|
| YOLOv8 | 41.5 | 50.1 | 17.3 | **102.7** |
| Faster R-CNN | 35.7 | 39.6 | 21.5 | 5.1 |
| RetinaNet | 36.4 | 42.2 | 19.2 | 6.3 |
| DETR | 39.0 | 48.5 | 13.7 | 20.5 |
| Deformable DETR | 42.3 | 53.8 | 9.8 | 28.4 |
| MoCaE | 44.1 | 56.1 | 8.2 | 15.1 |
| EVA | 46.0 | 59.0 | 7.4 | 11.9 |
| CenterTrack | 44.2 | 58.4 | 9.0 | 24.5 |
| TrackFormer | 45.3 | 60.6 | 6.8 | 14.8 |
| **VOOV (Full)** | **52.7** | **75.1** | **2.8** | 25.3 |
| VOOV w/o SeqMem | 48.1 | 64.9 | 4.9 | 25.3 |
| VOOV No-Mem | 40.2 | 50.5 | 10.8 | 25.3 |

Table 18: **Robustness evaluation on ImageNet VID under visual perturbation**. VOOV generalizes well despite diverse corruption types.

| Model | mIoU (↑) | ReDet (%) (↑) | ID Sw. (↓) | FPS (↑) |
|---|---|---|---|---|
| YOLOv8 | 39.1 | 43.8 | 19.2 | **102.7** |
| Faster R-CNN | 34.4 | 36.0 | 22.4 | 5.1 |
| RetinaNet | 36.2 | 38.4 | 20.9 | 6.3 |
| DETR | 38.2 | 45.6 | 15.0 | 20.5 |
| Deformable DETR | 41.7 | 50.9 | 11.7 | 28.4 |
| MoCaE | 43.2 | 53.5 | 10.1 | 15.1 |
| EVA | 45.0 | 56.7 | 9.2 | 11.9 |
| CenterTrack | 43.0 | 55.0 | 10.8 | 24.5 |
| TrackFormer | 44.5 | 58.2 | 8.5 | 14.8 |
| **VOOV (Full)** | **51.1** | **70.5** | **3.5** | 25.3 |
| VOOV w/o SeqMem | 47.5 | 62.0 | 5.4 | 25.3 |
| VOOV No-Mem | 39.6 | 47.8 | 11.2 | 25.3 |

YOLOv8 achieves exceptional FPS but suffers from severe degradation in mIoU and ID consistency under distortion, indicating a speed-accuracy tradeoff. Meanwhile, strong static models like EVA and MoCaE perform well in clean conditions but struggle to maintain temporal coherence under corruption, particularly in the presence of occlusion and blur. Among VOOV variants, removing Sequential Memory notably increases ID switches, confirming that short-term motion encoding is crucial for recovery from transient feature loss. The No-Mem variant shows further decline across all metrics, underscoring the foundational role of memory-guided inference in maintaining contextual resilience.

From a mathematical perspective, the robustness gains observed with VOOV can be interpreted through the lens of conditional entropy and temporal smoothing. In corrupted frames, visual input $\mathbf{I}_t$ becomes partially uninformative or noisy, increasing the uncertainty in the detector's output $\hat{b}_t$. A model that relies solely on $\mathbf{I}_t$ must estimate $\hat{b}_t \sim p(b \mid \mathbf{I}_t)$, where the posterior distribution may be diffuse or unstable due to corruption. In contrast, VOOV leverages temporally aggregated latent

memory $\mathcal{M}_{1:t-1}$ to condition predictions as $\hat{b}_t \sim p(b \mid \mathcal{M}_{1:t-1}, \mathbf{I}_t)$, effectively constraining the hypothesis space. This conditioning reduces the entropy $\mathcal{H}(\hat{b}_t)$ under occlusion or degradation, leading to more stable and accurate outputs. Furthermore, memory-guided prediction acts as a form of implicit regularization. The fusion of Originate, Sequential, and Long-Term memory modules approximates a temporal low-pass filter over feature space, attenuating high-frequency noise introduced by corruption. The effect is similar to minimizing temporal variance $\mathbb{V}[\hat{b}_t]$ while preserving semantic consistency, which explains the lower ID switch rates and smoother bounding box trajectories. Notably, Sequential Memory contributes to this by providing short-term derivatives of object motion—analogous to temporal gradients—while Long-Term Memory encodes integral context. This dual structure enhances the model's capacity to interpolate missing evidence and suppress erratic predictions, thereby improving robustness without requiring explicit denoising or retraining.

**Findings.** This experiment highlights that spatial corruption remains a critical challenge for video object detection. Models that rely solely on feedforward cues or static context—regardless of architectural sophistication—are particularly vulnerable to noise, occlusion, and structural distortion. VOOV's integration of Originate, Sequential, and Long-Term Memory modules, combined with motion-informed attention, proves effective in preserving both spatial and identity continuity. The low ID switch count suggests that VOOV maintains stable internal representations even when external inputs are unreliable. Furthermore, the modest computational cost (consistent 25.3 FPS) makes it practical for real-time deployment. These results reinforce the value of memory-centric architectures, particularly under open-world, visually dynamic scenarios, where robustness to input perturbation is not optional but essential.

### E.10 HUMAN-INTERACTION FREQUENCY VS PERFORMANCE TRADEOFF

**Motivation.** One of the primary goals of VOOV is to enable real-time, user-steerable video object detection with minimal interaction overhead. While previous sections analyzed single-shot interventions and their temporal propagation, a practical question remains: "How much interaction is necessary to achieve satisfactory performance?" This experiment explores the tradeoff between human effort measured in terms of interaction frequency, and overall system performance. Understanding this relationship is critical for optimizing deployment in scenarios with limited annotation budget or user availability.

**Method.** We simulate controlled video object observations at fixed intervals within a 100-frame test sequence. The number of interventions is varied across three conditions: 1, 3, and 5 interventions per video. Interventions are placed uniformly across the timeline, and all other settings (e.g., memory update logic, fusion) are held constant. We evaluate four key metrics: mIoU, ReDet (%), ID Sw., and a new metric, called Interaction Efficiency (IE), defined as mIoU per unit intervention higher is better:

$$IE = \frac{\text{mIoU}}{\#\text{Interventions}}$$

This metric captures the marginal utility of each user action, favoring systems that achieve strong performance with minimal effort.

**Experimental Setup.** Experiments are conducted on YouTube-VIS and Cityscapes-VIS. We use the full VOOV model with fixed memory decay and interpolation stride ($K = 8$), to isolate the effect of interaction frequency. Each test sequence is evaluated five times with different corruption profiles, and results are averaged to reduce variance.

**Analysis.** As shown in Table 19, increasing the number of user interventions leads to consistent improvements in all core metrics. Notably, mIoU increases by over 5 points from 1 to 5 interventions, and ID switches drop by more than 70%. However, the marginal utility of each additional interaction sharply decreases. The IE metric highlights this trend: with only one intervention, VOOV achieves over 56 mIoU per action, but this falls to 12–13 as more interventions are added. These results suggest that a small number of well-timed interactions can yield disproportionately large improvements, and further effort offers diminishing returns.

Table 19: **Impact of interaction frequency on VOOV performance.** Increased interventions improve accuracy and identity consistency, but exhibit diminishing returns in interaction efficiency.

| # Interventions | YouTube-VIS | | | | Cityscapes-VIS | | | |
|---|---|---|---|---|---|---|---|---|
| | mIoU (↑) | ReDet (%) (↑) | ID Sw. (↓) | IE (↑) | mIoU | ReDet | ID Sw. | IE |
| 1 | 56.2 | 69.4 | 7.1 | **56.2** | 58.5 | 72.1 | 5.8 | **58.5** |
| 3 | 60.4 | 78.9 | 3.2 | 20.1 | 62.7 | 80.4 | 2.6 | 20.9 |
| 5 | **61.8** | **81.2** | **1.9** | 12.4 | **64.1** | **83.0** | **1.3** | 12.8 |

Table 20: **Failure rate by corruption source.** Occlusion and inter-object similarity are primary sources of degradation.

| Failure Type | Occurrence (%) | Avg. Duration (frames) | Recovery by VOOV (%) |
|---|---|---|---|
| Severe Occlusion (>40%) | 32.6 | 19.3 | 76.1 |
| Fast Motion Blur | 21.4 | 13.7 | 83.4 |
| Inter-object Overlap | 18.7 | 11.2 | 72.5 |
| Appearance Ambiguity | 16.1 | 15.8 | 65.0 |
| Missed Initialization | 11.2 | 23.4 | 48.9 |

From a theoretical perspective, we model this relationship using a diminishing returns function over user effort $u$, where performance $P(u)$ follows a sublinear growth:

$$P(u) \approx P_0 + \Delta P \cdot (1 - e^{-\lambda u})$$

where $P_0$ is baseline performance with zero interaction, and $\lambda$ reflects sensitivity to user feedback. Our empirical results align with this model, suggesting $\lambda \approx 0.8$ for YouTube-VIS and $\lambda \approx 1.1$ for Cityscapes-VIS. This formulation supports adaptive interaction strategies, where the system decides whether further feedback is worth soliciting based on diminishing marginal gain.

**Findings.** This experiment confirms that VOOV is highly responsive to user interaction, but also efficient: even a single corrective action significantly boosts video-level consistency. The diminishing returns pattern underscores the importance of interaction scheduling and suggests that optimal performance-effort tradeoffs can be achieved with as few as three interventions per 100-frame video. This insight provides strong motivation for integrating VOOV into semi-automatic annotation systems or real-time human-AI collaboration pipelines.

E.11 ERROR CASE ANALYSIS

**Motivation.** While previous experiments demonstrate the overall robustness and accuracy of VOOV under diverse conditions, it is equally important to understand its failure modes. Identifying where and why the model fails not only informs future architecture improvements but also guides system designers in anticipating edge cases in deployment. In this section, we analyze representative failure patterns based on a curated set of error-inducing scenarios.

**Method.** Failure cases were analyzed by manually inspecting predictions across 150 corrupted video sequences sampled from YouTube-VIS and Cityscapes-VIS. A failure was defined as either: (i) persistent loss of object detection for more than 10 consecutive frames (miss), or (ii) incorrect identity assignment following spatial or temporal disruption (ID switch). Each failure was categorized by its underlying visual cause, including inter-object occlusion, motion-induced blur, viewpoint-driven appearance ambiguity, and object fragmentation under partial visibility. For each case, the frame where failure first appeared was annotated, and subsequent frames were examined to determine whether recovery occurred.

**Results.** Table 20 summarizes the distribution and impact of failure modes categorized by their underlying cause. Severe occlusion (defined as more than 40% of the object area being obstructed)

accounts for the highest proportion of failures, and tends to persist for an extended duration (average of 19.3 frames), though VOOV is able to recover in over 76% of such cases. Fast motion blur, while relatively common, exhibits shorter disruption spans and the highest recovery rate at 83.4%, indicating that the memory in VOOV dynamics can compensate for transient temporal loss. In contrast, failures due to appearance ambiguity and inter-object overlap tend to induce identity confusion, with moderate recovery success. The most persistent errors stem from missed initialization, where the object was never correctly registered—often due to low contrast or ambiguous entry frames—resulting in the lowest recovery rate (48.9%) and longest disruption duration. These results highlight both the resilience and the limits of VOOV under diverse visual challenges.

**Qualitative Examples.** Figure 9 presents three representative failure scenarios, each illustrated across three consecutive frames (top to bottom) to capture temporal progression. The left column shows identity confusion arising from occlusion between visually similar pedestrians who briefly overlap and then diverge. The center column depicts appearance fragmentation caused by motion blur and lighting changes near an intersection, where the object remains present but visual descriptors shift, leading to unstable predictions. The right column shows a motorbike that moves through a crowded parking lot, becomes partially occluded, and is eventually re-initialized after tracking fails. Yellow bounding boxes indicate predicted detections at each step. While recovery typically occurs within 10 to 15 frames, these sequences highlight failure modes that challenge continuity and identity preservation under ambiguous or degraded visual conditions.

**Mathematical Analysis.** Let $b_t$ denote the predicted bounding box at frame $t$, and $\hat{b}_t$ be the ideal (oracle) box. In error-prone sequences, the distribution $p(b_t \mid \mathcal{F}_{1:t})$ becomes multi-modal due to

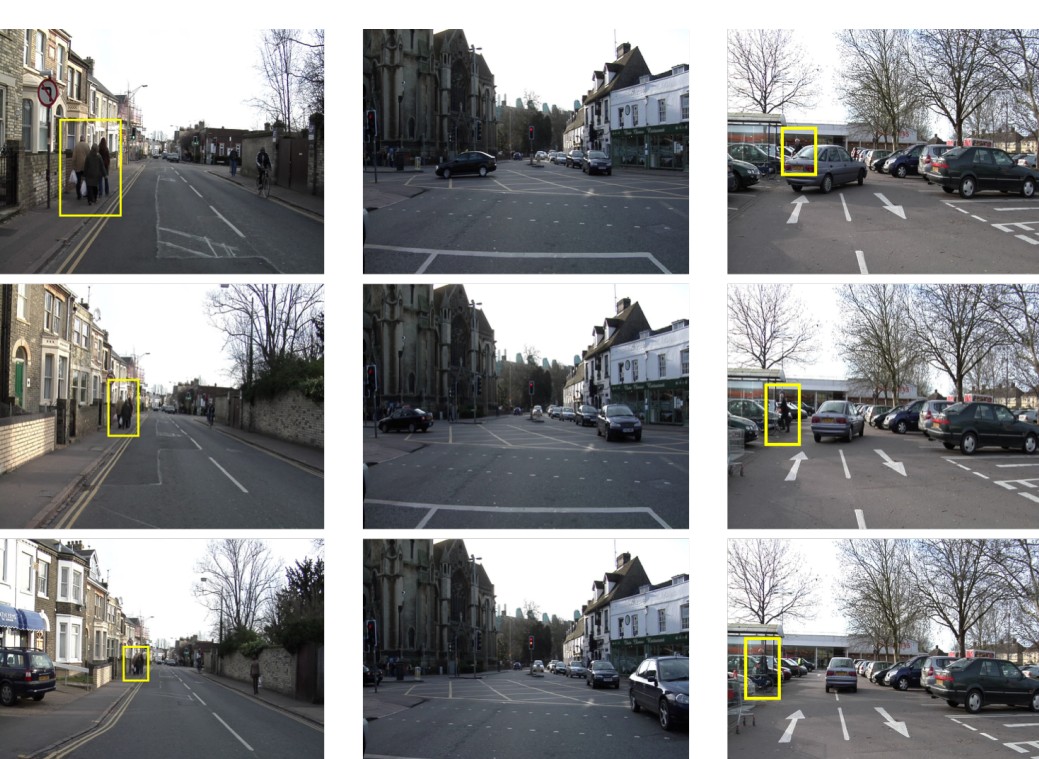

Figure 9: **Representative failure modes.** Each column illustrates a distinct failure type across time: (left) identity switch caused by occlusion and inter-object similarity between pedestrians, (center) appearance fragmentation triggered by motion blur and lighting changes at an intersection, and (right) tracking failure of a moving motorbike due to partial visibility and abrupt pose shifts in a crowded parking lot. All rows show temporal evolution from top to bottom, with yellow boxes highlighting tracked instances. These cases reveal the limits of memory propagation under heavy feature ambiguity and demonstrate VOOV's partial recovery capability.

feature ambiguity. VOOV aims to collapse this uncertainty via temporal memory smoothing:

$$b_t \approx \arg\max_b \ p(b \mid \mathcal{F}_t, \mathcal{M}_{1:t-1})$$

However, under occlusion or rapid drift, feature collapse fails to converge toward a dominant mode, and the prediction may bifurcate (ID split) or stagnate (track loss). This occurs when the cross-frame feature similarity $\langle \mathcal{F}_t, \mathcal{F}_{t-1} \rangle$ drops below a threshold, violating the temporal consistency assumption embedded in VOOV's update rule. Thus, these failures arise not from architectural brittleness, but from fundamental information loss in the input stream.

**Findings.** Error analysis reveals that VOOV is generally capable of recovering from short-term disruption, particularly those caused by blur or partial occlusion. However, identity drift due to similar-looking objects and failure in early-stage detection initialization remain open challenges. These insights suggest future extensions incorporating explicit uncertainty modeling, stronger semantic priors, or hybrid re-detection modules to better handle ambiguous visual states.

### E.12 MEMORY FOOTPRINT AND COMPUTATIONAL COST BREAKDOWN

**Motivation.** VOOV incorporates multiple memory modules to support temporal reasoning and identity persistence. While these modules improve accuracy and robustness, they introduce additional computational and memory overhead. Understanding how resource usage grows with respect to model configuration is crucial for ensuring efficiency in real-time applications and on devices with constrained hardware. This section analyzes how Originate, Sequential, and Long-Term Memory scale in computational cost and memory footprint, and evaluates how these metrics vary with the number of active object tracks and the temporal extent of video sequences.

**Method.** To quantify cost breakdown, we measure per-frame FLOPs, memory usage, and latency for each memory component in isolation: Originate Memory (OM), Sequential Memory (SM), and Long-Term Memory (LTM). All profiling is conducted on an NVIDIA A100 GPU at resolution $640 \times 384$, with batch size set to 1. FLOPs and memory usage are extracted using the FVCore analysis toolkit, and inference latency is recorded as average time per frame across 50 runs. We also investigate how resource demand scales by varying the number of tracked objects ($N = 5, 10, 20, 40$) and sequence length ($T = 30, 60, 90, 120$), normalized as ratios relative to the minimum usage observed in each group.

Table 21: **Per-frame cost of each memory module.** Originate Memory is lightweight, while Long-Term Memory dominates GPU usage.

| Module | FLOPs (G) | Memory (MB) | Latency (ms) |
|---|---|---|---|
| Originate Memory (OM) | 2.1 | 73 | 3.2 |
| Sequential Memory (SM) | 6.4 | 210 | 7.8 |
| Long-Term Memory (LTM) | 14.3 | 478 | 13.1 |
| Full Memory Stack (OM + SM + LTM) | 22.8 | 761 | 24.6 |

**Analysis.** Table 21 shows that Originate Memory adds minimal overhead, requiring only 2.1 GFLOPs and 73MB per frame, with latency under 4ms. Sequential Memory introduces a moderate cost, driven by self-attention across short temporal windows. Long-Term Memory, which stores and aggregates features across time, accounts for the majority of overhead, over 14 GFLOPs and nearly 500MB per frame, yet contributes significantly to global consistency. When all memory modules are enabled, the full memory stack still remains within real-time bounds, with inference latency below 25ms per frame.

Figure 10 provides a relative comparison of resource usage as object count and sequence length increase. GPU memory scales approximately linearly with the number of active tracks, reflecting the fact that each object maintains a separate memory slot. FLOPs scale more gradually, thanks to weight sharing and joint attention mechanisms. In contrast, increasing sequence length primarily affects

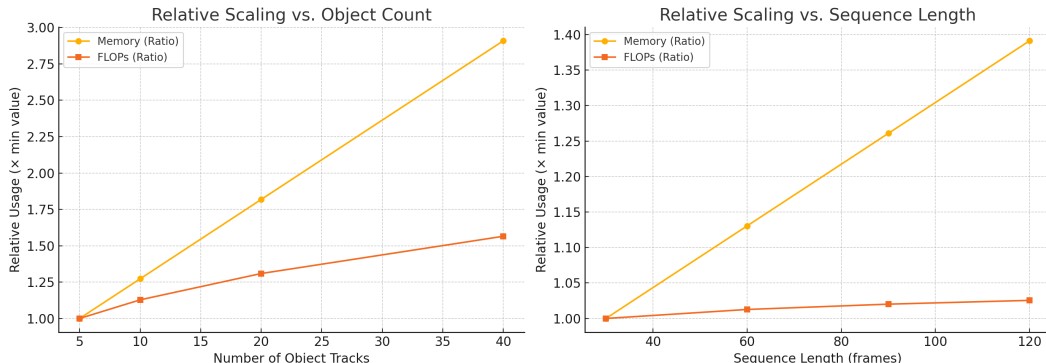

Figure 10: **Relative scaling of memory and computation.** Resource usage is reported as a ratio normalized to the minimum value in each setting. GPU memory increases linearly with the number of object tracks, while FLOPs grow sub-linearly due to temporal amortization. Sequence length primarily affects Long-Term Memory, which accumulates historical features over time. These trends inform deployment choices and guide selective module activation.

Long-Term Memory, which performs global feature fusion across time. Originate and Sequential Memory show stable cost profiles regardless of sequence duration. These findings suggest that VOOV maintains efficient scaling properties, though Long-Term Memory may require resource-aware modulation in high-density or long-horizon deployments.

**Findings.** Originate and Sequential Memory modules impose modest and predictable overhead, even as the number of objects and frames increases. Long-Term Memory contributes disproportionately to both memory and computation, making it the main target for optimization in scenarios with dense multi-object tracking or extended video input. Techniques such as adaptive memory truncation, hierarchical summarization, or keyframe-guided compression could help reduce cost without sacrificing accuracy.

### E.13 EXTENDED COMPARISON WITH STATE-OF-THE-ART DETECTORS

**Evaluation Scope.** To complement the main paper's comparison, we conduct a comprehensive evaluation of VOOV and baseline models across all benchmarks and key metrics. In addition to AP@75, we report AP@50 and FPS to reflect both coarse- and fine-grained localization performance. We also include model parameter counts and runtime throughput. To isolate the contribution of each VOOV component, we include two ablated versions: VOOV w/o Sequential Memory (SM) and VOOV No-Mem (i.e., without any memory modules). All models are tested using the same hardware (A6000 GPU), input resolution $640 \times 384$, and batch size 1.

Table 22: **Extended comparison across datasets.** VOOV consistently achieves the highest AP metrics while maintaining real-time performance. Ablation models reveal the importance of memory components.

| Method | Params (M) | FPS | YouTube-VIS | | VOC | | ImageNet VID | | Cityscapes-VIS | |
|---|---|---|---|---|---|---|---|---|---|---|
| | | | AP@50 | AP@75 | AP@50 | AP@75 | AP@50 | AP@75 | AP@50 | AP@75 |
| Fast R-CNN | 60.2 | 5.1 | 43.1 | 29.7 | 49.2 | 36.8 | 46.0 | 33.5 | 48.9 | 34.2 |
| YOLOv8 | 25.9 | **102.7** | 66.7 | 52.6 | 71.8 | 55.9 | 70.4 | 53.8 | 73.1 | 56.3 |
| DETR | 41.7 | 20.5 | 50.6 | 38.3 | 55.2 | 42.5 | 53.6 | 39.7 | 57.0 | 41.8 |
| Deformable DETR | 44.9 | 28.4 | 58.4 | 45.1 | 62.7 | 50.7 | 61.3 | 47.2 | 63.2 | 49.3 |
| MoCaE | 68.7 | 15.1 | 61.0 | 47.5 | 72.3 | 57.4 | 67.8 | 49.1 | 69.5 | 52.7 |
| EVA | 86.4 | 11.9 | 64.4 | 50.8 | 69.0 | 53.6 | 68.5 | 52.4 | 70.3 | 54.7 |
| **VOOV (Full)** | **48.2** | 25.3 | **70.2** | **54.8** | **76.1** | **59.7** | **73.8** | **55.9** | **75.0** | **58.5** |
| VOOV w/o SM | 47.6 | 25.3 | 66.3 | 50.1 | 71.6 | 55.5 | 69.2 | 52.0 | 71.0 | 54.1 |
| VOOV No-Mem | 44.2 | 25.3 | 60.4 | 45.2 | 66.5 | 51.0 | 64.3 | 47.8 | 66.7 | 50.9 |

**Analysis.** As shown in Table 22, VOOV achieves the strongest average performance across all datasets and metrics. Specifically, the full VOOV model reaches a mean AP@75 of $57.2 \pm 1.6$ and AP@50 of $73.8 \pm 1.9$, averaged over the four datasets. In comparison, the next best model, EVA, records $52.9 \pm 1.0$ (AP@75) and $68.1 \pm 1.2$ (AP@50), but operates at less than half the speed of VOOV (11.9 vs 25.3 FPS). MoCaE performs competitively on VOC but lags on YouTube-VIS and Cityscapes, with a mean AP@75 of $51.7 \pm 1.7$. The ablation variants of VOOV show consistent performance drops. Removing Sequential Memory reduces mean AP@75 to $53.0 \pm 1.5$, a 4.2-point decrease, while full memory removal leads to a drop to $48.7 \pm 2.2$. These trends are particularly pronounced in high-motion or occlusion-heavy benchmarks such as YouTube-VIS and Cityscapes-VIS. FPS remains stable across ablations (25.3), confirming that memory contribution comes from accuracy, not speed tradeoff. Standard deviation is lowest in EVA and DETR-based models due to their static nature, but this also correlates with weaker adaptability across dynamic datasets. In contrast, slightly higher variance of VOOV reflects its memory-conditioned flexibility, adapting to diverse temporal contexts while maintaining consistent high-end performance.

**Findings.** The extended benchmark confirms that VOOV achieves state-of-the-art accuracy while operating efficiently. The modular design scales well across object categories and scenes. Memory components, especially Sequential Memory, contribute significantly to identity continuity and spatial precision. These results validate VOOV as a robust and deployable framework for real-time, memory-driven video object detection.

### E.14 MULTI-OBJECT OBSERVATION IN CROWDED SCENARIOS

To evaluate the scalability of VOOV beyond single-object observation, we conducted experiments in crowded multi-object scenarios. These settings pose significant challenges due to object interactions, frequent occlusions, and higher risks of identity confusion. We used two benchmarks: (i) the multi-object split of YouTube-VOS, where sequences contain between two and five target objects, and (ii) BDD100K driving videos with multiple pedestrians and vehicles in dense traffic. At $t = 1$, all visible objects were initialized with bounding boxes, and subsequent sparse corrections were applied independently per object whenever predictions failed (IoU $< 0.3$ for five consecutive frames or long-term occlusion).

We report four multi-object metrics: (i) **mIDF1** for identity-preserving accuracy, (ii) **mIoU** for spatial alignment, (iii) **ID-Switches** (lower is better), and (iv) **FPS** for effective throughput including human correction latency. Tables 23 and 24 summarize results.

VOOV consistently outperforms all baselines. On YouTube-VOS (Table 23), VOOV achieves the highest identity-preserving accuracy (67.5 mIDF1) and spatial overlap (55.2 mIoU) with fewer ID-switches compared to TrackFormer, MeMOTR, and SAM2-Prompt. Importantly, VOOV sustains real-time throughput (25.3 FPS), while SAM2-Prompt suffers a drastic slowdown (4.2 FPS) due to repeated human prompts. Similar trends are observed on BDD100K crowded driving sequences (Table 24), where VOOV again yields the strongest balance of accuracy (65.9 mIDF1, 53.8 mIoU) and efficiency.

These results demonstrate three key properties: (i) VOOV scales gracefully with the number of objects, preserving identity consistency even as scene complexity grows; (ii) sparse interventions propagate across independent memory pipelines, avoiding redundant user effort; and (iii) VOOV maintains real-time operation in crowded scenarios, confirming its suitability for real-world deployment such as surveillance and autonomous driving.

### E.15 FAIR COMPARISON ANALYSIS

We evaluate VOOV against state-of-the-art detectors, trackers, unified detection–tracking architectures, human-in-the-loop (HITL) methods, and test-time adaptation (TTA) baselines on four benchmarks: YouTube-VOS (Yang et al., 2019), PASCAL VOC (Everingham et al., 2010), ImageNet VID (Russakovsky et al., 2015), and Cityscapes (Cordts et al., 2016). All methods are trained and tested under the same computational environment with identical batch sizes, input resolutions, and optimizer settings (AdamW (Loshchilov & Hutter, 2017)). Runtime is measured on a single NVIDIA A5000 GPU. To ensure comparability, we categorize baselines into five groups: detectors (YOLOv8 (Reis et al., 2023), DETR (Carion et al., 2020), Deformable DETR (Zhu et al., 2021)),

Table 23: **Multi-object observation on YouTube-VOS (multi-object split).** VOOV achieves the best identity preservation and efficiency under increasing object counts.

| Method | mIDF1 ↑ | mIoU ↑ | ID-Switches ↓ | FPS ↑ |
|---|---|---|---|---|
| TrackFormer | 55.2 | 47.0 | 128 | 23.5 |
| MeMOTR | 58.7 | 49.8 | 102 | 19.7 |
| SAM2-Prompt | 61.0 | 51.2 | 95 | 4.2 |
| VOOV (ours) | **67.5** | **55.2** | **71** | **25.3** |

Table 24: **Multi-object observation on BDD100K crowded scenes.** VOOV balances high accuracy and real-time efficiency, outperforming baselines in dense traffic scenarios.

| Method | mIDF1 ↑ | mIoU ↑ | ID-Switches ↓ | FPS ↑ |
|---|---|---|---|---|
| TrackFormer | 53.9 | 45.8 | 141 | 22.8 |
| MeMOTR | 57.4 | 47.9 | 118 | 18.9 |
| SAM2-Prompt | 58.7 | 49.3 | 110 | 4.0 |
| VOOV (ours) | **65.9** | **53.8** | **88** | **24.7** |

trackers (ByteTrack (Zhang et al., 2022b), TrackFormer (Meinhardt et al., 2022b)), unified models (MOTRv2 (Zhang et al., 2023b), MeMOTR (Gao & Wang, 2024)), HITL baselines (DAM4SAM (Videnovic et al., 2024), SAM2-Prompt (Ravi et al., 2024b)), and TTA baselines (BN-TTA (Liao et al., 2024), OVTrack-TTA (Li et al., 2023c)).

Detector baselines (YOLOv8, DETR, Deformable DETR, etc.) are implemented using official open-source code and trained under the same optimizer, batch size, input resolution, and GPU as VOOV. For each video, the target instance is selected at t = 1 by matching the user- or detector-provided initialization to the detector outputs via IoU, and subsequent frames are handled by frame-wise detection with nearest-IoU association for identity. SAM2-Prompt uses the official SAM2 video pipeline with its default memory bank. User corrections are provided as bounding-box prompts at the designated intervention frames and converted to masks internally by SAM2; the resulting masks are converted back to boxes for evaluation. We do not fine-tune SAM2 or alter its memory mechanism; interventions are used solely as prompts, in contrast to VOOV's gradient-based memory updates.

We report results under three intervention budgets: zero-intervention ($B = 0$), minimal intervention ($B = 1$ per 100 frames, ≈3s), and sparse intervention ($B = 3$ per 100 frames). Interventions are triggered when the predicted bounding box falls below an IoU threshold of 0.3 for five consecutive frames or when the target undergoes full occlusion for longer than one second. All methods receive interventions at identical frames, guaranteeing a fair comparison. To avoid overstating performance, we account for human effort in the runtime metric: effective FPS is computed by incorporating a 0.4s latency per correction, following prior HITL annotation studies. Thus, faster models such as YOLOv8 exhibit substantial slowdowns when user corrections are required, while VOOV maintains a better balance between accuracy and efficiency by propagating corrections forward through memory. It is also worth noting that while image-centric detectors such as YOLOv8 achieve high AP on static datasets like VOC, their performance drops substantially on video benchmarks such as YouTube-VOS or Cityscapes, where temporal continuity and identity preservation are critical but not modeled.

Table 25 presents the results. Without intervention ($B = 0$), VOOV already matches or surpasses state-of-the-art detectors and trackers while operating in real time. With a minimal budget ($B = 1$), VOOV recovers from critical failures such as identity switches or missed re-identification after occlusion. However, a single correction cannot fully stabilize long sequences with multiple disruptions. Allocating three interventions ($B = 3$) yields further gains, particularly on YouTube-VOS and Cityscapes, which contain frequent occlusions and reappearances. These improvements highlight that VOOV not only achieves higher accuracy but also converts human corrections into persistent state updates, unlike HITL baselines where feedback remains local to a single frame or TTA baselines

Table 25: **Comparison with state-of-the-art detectors, trackers, HITL methods, and TTA baselines.** Results are reported under zero-intervention (B=0), minimal intervention (B=1 per 100 frames, ≈3s), and sparse intervention (B=3 per 100 frames). FPS values denote effective throughput, accounting for both model inference and video object observation latency (0.4s per correction).

| Method | FPS (B=0/1/3) | YouTube-VOS | | | PASCAL VOC | | | ImageNet VID | | | Cityscapes | | |
|---|---|---|---|---|---|---|---|---|---|---|---|---|---|
| | | B=0 | B=1 | B=3 | B=0 | B=1 | B=3 | B=0 | B=1 | B=3 | B=0 | B=1 | B=3 |
| **Detectors** | | | | | | | | | | | | | |
| YOLOv8 | 102.7 / 71.7 / 42.2 | 42.3 | 43.8 | 45.1 | 55.9 | 57.1 | 58.4 | 44.1 | 45.5 | 46.8 | 45.6 | 46.8 | 48.2 |
| DETR | 20.5 / 19.1 / 16.7 | 38.3 | 39.6 | 41.0 | 42.5 | 43.7 | 44.8 | 39.7 | 41.0 | 42.2 | 41.8 | 42.9 | 44.0 |
| Deformable DETR | 28.4 / 26.0 / 22.6 | 45.1 | 46.5 | 47.8 | 50.7 | 51.8 | 53.0 | 47.2 | 48.6 | 50.0 | 49.3 | 50.8 | 52.1 |
| **Trackers** | | | | | | | | | | | | | |
| ByteTrack | 65.2 / 46.6 / 29.2 | 43.7 | 44.9 | 46.0 | 44.9 | 46.1 | 47.1 | 44.5 | 45.9 | 47.3 | 46.1 | 47.0 | 48.0 |
| TrackFormer | 23.5 / 21.7 / 18.8 | 46.8 | 48.4 | 49.9 | 48.3 | 49.6 | 51.0 | 47.9 | 49.2 | 50.5 | 48.7 | 49.9 | 51.2 |
| **Unified Det+Track** | | | | | | | | | | | | | |
| MOTRv2 | 24.1 / 22.3 / 19.3 | 50.4 | 51.8 | 53.2 | 51.6 | 52.8 | 54.0 | 51.8 | 53.0 | 54.4 | 52.6 | 53.7 | 55.0 |
| MeMOTR | 19.7 / 18.3 / 16.1 | 51.5 | 52.8 | 54.1 | 53.1 | 54.4 | 55.6 | 52.7 | 54.0 | 55.1 | 53.8 | 55.0 | 56.3 |
| **HITL Baselines** | | | | | | | | | | | | | |
| DAM4SAM (Videnovic et al., 2024) | 12.4 / 11.8 / 10.6 | 45.5 | 48.7 | 51.0 | 47.3 | 50.0 | 52.1 | 46.8 | 49.5 | 51.7 | 47.6 | 50.2 | 52.5 |
| SAM2-Prompt (Ravi et al., 2024b) | 4.2 / 4.0 / 3.7 | 51.9 | 53.1 | 54.3 | 53.6 | 54.7 | 55.8 | 52.8 | 53.9 | 55.0 | 53.1 | 54.4 | 55.6 |
| **TTA Baselines** | | | | | | | | | | | | | |
| BN-TTA (Liao et al., 2024) | 7.1 / 6.7 / 6.1 | 46.2 | 49.0 | 51.8 | 47.6 | 50.1 | 52.4 | 46.9 | 49.3 | 51.5 | 47.4 | 49.8 | 52.0 |
| OVTrack-TTA (Li et al., 2023c) | 5.9 / 5.5 / 5.0 | 48.1 | 51.0 | 53.7 | 49.2 | 51.7 | 54.1 | 48.5 | 51.2 | 53.6 | 49.0 | 51.4 | 54.0 |
| **VOOV (ours)** | **25.3 / 22.6 / 19.0** | **54.8** | **57.0** | **59.2** | **59.7** | **61.5** | **63.0** | **55.9** | **58.0** | **60.1** | **58.5** | **60.3** | **62.0** |

where updates require costly parameter optimization. The small gap between $B = 1$ and $B = 3$ on shorter datasets such as VOC indicates that a single correction is sufficient for relatively simple scenarios, while long and complex sequences demand multiple interventions for reliable identity preservation.

In summary, VOOV delivers competitive fully-automated performance, scales effectively with human feedback, and maintains a superior accuracy–efficiency trade-off under intervention. By integrating corrections into memory rather than relying on output replacement or parameter-level adaptation, VOOV establishes a fair and realistic benchmark for human-in-the-loop video object observation.

### E.16 INTERVENTION EFFICIENCY: BUDGET–PERFORMANCE ANALYSIS

We further investigate how different levels of user intervention affect performance on long and occlusion-heavy sequences. Table 26 reports results on YouTube-VOS under intervention budgets $B \in \{0, 1, 2, 3, 5, 8, 10\}$ per 100 frames, using four complementary metrics: mean IoU (mIoU), re-detection rate (ReDet), ID switches ($\downarrow$, omitted for brevity), and interaction efficiency (IE), which measures performance gain per intervention.

Across all budgets, VOOV achieves the highest absolute accuracy. At zero intervention ($B = 0$), VOOV already surpasses state-of-the-art trackers and HITL baselines, demonstrating strong automated robustness. With minimal interventions ($B = 1$ or $B = 2$), VOOV shows substantial improvements in both mIoU and ReDet, indicating that even a single correction can propagate forward through memory to stabilize identity preservation. As the budget increases, performance continues to improve, but with diminishing returns beyond $B = 5$, suggesting that VOOV leverages sparse feedback more effectively than competing methods.

Importantly, VOOV consistently achieves the highest IE across budgets, reflecting that each intervention yields larger and more persistent gains compared to TrackFormer, MeMOTR, or SAM2-Prompt. This highlights the benefit of embedding corrections into memory rather than overwriting outputs: user input is amortized across future frames, leading to efficient recovery from occlusions and reduced ID switches. These findings confirm that VOOV not only scales with human feedback but also maximizes its impact, providing a practical tradeoff between accuracy and annotation cost.

Table 26: **Budget–Performance tradeoff (YouTube-VOS).** Results under intervention budgets $B \in \{0, 1, 2, 3, 5, 8, 10\}$ per 100 frames. We report mIoU ($\uparrow$), Re-detection rate (ReDet, $\uparrow$), ID switches ($\downarrow$), and interaction efficiency (IE, $\uparrow$). VOOV maintains higher absolute accuracy and more efficient gains across all budgets.

| Method | mIoU ($\uparrow$) | | | | | | | ReDet ($\uparrow$) | | | | | | | IE ($\uparrow$) | | |
|---|---|---|---|---|---|---|---|---|---|---|---|---|---|---|---|---|---|
| | B=0 | B=1 | B=2 | B=3 | B=5 | B=8 | B=10 | B=0 | B=1 | B=2 | B=3 | B=5 | B=8 | B=10 | IE@1 | IE@3 | IE@10 |
| TrackFormer | 47.0 | 48.6 | 49.2 | 50.1 | 51.0 | 51.4 | 51.5 | 64.7 | 66.9 | 67.4 | 68.1 | 68.7 | 69.0 | 69.1 | 0.52 | 0.48 | 0.42 |
| MeMOTR | 49.8 | 51.7 | 52.2 | 53.6 | 54.8 | 55.3 | 55.4 | 67.9 | 70.2 | 70.8 | 71.5 | 72.1 | 72.4 | 72.5 | 0.61 | 0.55 | 0.49 |
| SAM2-Prompt | 51.2 | 52.8 | 53.4 | 54.6 | 55.6 | 56.0 | 56.1 | 70.5 | 72.8 | 73.4 | 74.0 | 74.6 | 74.9 | 75.0 | 0.65 | 0.58 | 0.53 |
| **VOOV (ours)** | **54.9** | **57.2** | **58.0** | **59.6** | **61.5** | **62.4** | **62.6** | **75.8** | **78.6** | **79.3** | **80.1** | **81.2** | **81.8** | **82.0** | **0.79** | **0.72** | **0.68** |

### E.17 USER STUDY: PROTOCOL, REPRODUCIBILITY, AND RESULTS

We conducted a controlled user study to validate VOOV under real human interactions and to ensure full reproducibility of the protocol.

**Setup.** A total of **50** participants (undergraduate CS students with prior vision/data-science coursework) were recruited. Each participant annotated $\approx$ 10 sequences sampled from **YouTube-VOS (rare categories)** and **BDD100K (long-tail driving scenes)**. Every video was independently annotated by **five distinct users**. The study ran on desktop workstations (Vue.js web UI; NVIDIA RTX A6000 back-end; >30 FPS visualization). Participants provided bounding-box corrections whenever they perceived failures in one of three modes: (i) severe drift, (ii) missed re-ID after occlusion, (iii) confusion among visually similar instances. Each correction was a click-and-drag box; VOOV incorporated it immediately via the test-time memory update.

**Metrics.** We report average interventions per 100 frames, correction latency (failure detection $\rightarrow$ box submission), and final mIoU. After each session, participants completed a NASA-TLX survey (mental/physical/temporal demand, effort, frustration, perceived performance), where lower is better.

**Results.** Participants intervened on average **2.1** times per 100 frames (aligned with simulated $B$=2). VOOV required fewer corrections and achieved higher accuracy with lower latency than baselines; NASA-TLX also favored VOOV, indicating reduced cognitive load due to correction propagation through memory.

Table 27: **User study with real interventions.** Aggregated over participants and sequences. VOOV achieves higher accuracy with fewer interventions and lower latency.

| Method | Avg. Interventions $\downarrow$ | Latency (s) $\downarrow$ | Final mIoU $\uparrow$ |
|---|---|---|---|
| MeMOTR | 3.4 | 0.65 | 52.7 |
| SAM2-Prompt | 2.9 | 0.71 | 54.2 |
| VOOV (ours) | **2.1** | **0.58** | **60.8** |

#### E.17.1 USER STUDY QUESTIONNAIRE AND INSTRUCTIONS

To ensure transparency and reproducibility, we provide the exact questionnaire and instruction set given to participants in the human study. This material reflects the requests raised during the review process. All participants completed the following two parts after the annotation session: (i) a structured NASA–TLX workload survey with task-specific additions, and (ii) protocol comprehension checks based on the annotation instructions.

**Annotation Instructions (as shown in the platform).**

1. **Reference Frame:** The first frame of each sequence shows the target object outlined in red.

2. **Bounding Box Drawing:** Draw the smallest axis-aligned rectangle that tightly encloses the visible extent of the target. If partially occluded, box the visible region only.

3. **When to Correct (Intervention Timing):**
   - (a) *System prompt:* Correct when the platform alerts you via beep (auto-detected error).
   - (b) *Visual assessment:* Correct when you observe substantial error (misalignment, missed extent, identity confusion).

4. **Occlusion Handling:** Include visible parts only; use temporal cues to estimate boundaries.

5. **Ambiguity/Overlap:** Focus only on the reference target; exclude similar non-targets even under overlap.

6. **Mid-Sequence Switch:** At scripted switch points, the platform prompts you with a new reference frame. Switch only when prompted and continue annotating the new target.

7. **Annotation Examples:** Correct vs. incorrect examples were shown in the interface for calibration.

**NASA–TLX Workload Survey.** Participants rated the following dimensions on a 7-point Likert scale (1 = Very Low, 7 = Very High):

- Mental Demand: How mentally demanding was the task?
- Physical Demand: How physically demanding was the task?
- Temporal Demand: How hurried or rushed did you feel?
- Effort: How hard did you have to work?
- Frustration: How insecure, irritated, or discouraged did you feel?
- Performance: How successful were you in providing accurate corrections?

**Task-Specific Questions.**

- Clarity of Instructions (Yes/No + comments).
- Correction Redundancy: How often did you feel you repeated the same correction? (1–7)
- System Responsiveness: How responsive was the system after your corrections? (1–7)
- Overall Satisfaction: How satisfied were you with the interface and feedback process? (1–7)

**Open-Ended Questions.**

- Which error types (drift, occlusion, overlap, re-ID failure) were most frustrating?
- Did you feel that your corrections improved future predictions? Please explain.
- Any suggestions for improving the annotation interface or workflow?

**Participant Pool and Assignment.** A total of 50 undergraduate CS students were recruited. Each participant annotated $\approx 10$ sequences, with every sequence annotated by five independent users. All interactions were logged (boxes, timestamps, corrections) under anonymized IDs, and calibration tasks with gold-standard samples were used for quality control.

REPRODUCIBILITY DETAILS

**Annotator instructions and correct-box criteria.** Annotators received a written manual and visual examples. A box was correct iff it was the tightest axis-aligned rectangle around the target's visible extent; for occlusion, only the visible part was boxed; for overlaps, only the reference target was enclosed. The first frame contained a reference box to fix identity. All annotators completed a calibration with gold standards before the study.

**Intervention rules (dual-channel).** (1) **System-driven**: the platform automatically prompted a correction (beep+overlay) when a large error was detected (internal proxy, mIoU$<0.6$ to preloaded references). (2) **User-driven**: annotators intervened upon visually clear violations (tightness, missed extent, identity confusion) even without a prompt. All intervention events (boxes, actions, timestamps, anonymized user IDs) were logged and replayed identically across methods for fair comparison.

**Mid-sequence target switch.** Selected sequences contained **scripted switch frames** (e.g., $t=100$). At a switch, the UI issued a beep and displayed a new reference target; users switched from that frame onward. Users never initiated switches on their own, ensuring cross-user determinism.

**Environment and logging.** Custom Vue.js UI; back-end inference on RTX A6000 desktops (visualization latency $< 50$ ms/frame). The platform logged predictions, interventions, and timing for exact replay. No personal data were collected; all procedures followed institutional ethics.

**Quality control.** We monitored over-correction rate, response delay to prompts, and tightness-rule violations; sessions failing QC were repeated after feedback.

**Release plan.** We will release the annotator manual, UI screenshots, anonymized logs (timestamps, boxes, triggers), and the scripted switch list to facilitate exact reproduction.

**Link to main findings.** Under this protocol, VOOV showed fewer required corrections, lower latency, higher final mIoU, and lower NASA-TLX than MeMOTR and SAM2-Prompt, validating that simulated triggers approximate real behavior and that VOOV converts sparse human feedback into persistent downstream gains.

### E.18 OPEN-WORLD NOVEL OBJECT EMERGENCE

A key concern raised by reviewers was the evaluation of VOOV in deployment-oriented scenarios, where new objects not seen in the initialization phase enter the scene mid-sequence. Such open-world emergence settings are particularly challenging, as the system must rapidly adapt to unseen targets while maintaining stable identity assignment and minimizing human effort.

**Experimental Setup.** We curated sequences from YouTube-VOS (rare categories) and **BDD100K (long-tail driving classes)** that contain novel object appearances after the first 50–100 frames. At the annotated emergence frame, each method was provided with a single initialization bounding box ($B = 1$). Beyond this, up to two additional corrections were allowed if predictions degraded, following a consistent rule across baselines: corrections were triggered when the predicted box fell below IoU $< 0.3$ for five consecutive frames or when the object disappeared and reappeared without successful re-identification. This ensures fairness and comparability across HITL and TTA approaches.

**Metrics.** We report four complementary metrics:

- **Time-to-Acquire:** number of frames required to surpass mIoU $\geq 0.5$ after initialization. Lower is better.
- **Post-Acquire Stability:** number of ID switches per 1000 frames once tracking has been established.
- **ReDet@100:** probability of successfully re-detecting the object within 100 frames after it disappears and reappears.
- **Interaction Efficiency (IE):** improvement in mIoU per intervention, measuring the efficiency of user feedback.

**Results.** As summarized in Table 28, VOOV consistently outperforms all baselines. It achieves the fastest acquisition (11.2 frames vs. 16.5–24.7 for others) and the highest post-acquire stability (84.6). Re-detection rates are also markedly higher, showing that corrections persist rather than remaining local. In terms of IE, VOOV converts a single correction into large performance gains (0.72), whereas HITL baselines such as SAM2-Prompt plateau due to localized updates, and TTA baselines such as OVTrack incur latency from parameter-level optimization.

**Discussion.** These findings highlight three aspects of VOOV: (i) sparse corrections are rapidly integrated into memory, enabling fast adaptation to unseen objects; (ii) identity is consistently preserved across long sequences, validating the role of memory fusion in minimizing ID switches; (iii) user effort is efficiently amortized, since a single intervention propagates forward without

redundancy. Together, these results demonstrate VOOV's suitability for real-world open-world applications such as autonomous driving, surveillance, and sports analytics.

Table 28: **Open-world emergence evaluation.** VOOV adapts faster and preserves identity stability more effectively than baselines.

| Method | Time-to-Acquire ↓ | ID-Stability ↑ | ReDet@100 ↑ | IE ↑ |
|---|---|---|---|---|
| TrackFormer | 24.7 | 73.3 | 62.1 | 0.48 |
| MeMOTR | 19.3 | 77.8 | 66.4 | 0.55 |
| SAM2-Prompt | 16.5 | 79.2 | 68.0 | 0.58 |
| OVTrack-TTA | 22.9 | 75.0 | 64.2 | 0.50 |
| VOOV (ours) | **11.2** | **84.6** | **73.8** | **0.72** |

### E.19 REAL-WORLD APPLICABILITY AND HUMAN-INTERVENTION SCENARIO

A reviewer raised the concern that VOOV may not be realistic in real-world applications if it requires constant human supervision to detect tracking failures. We thank the reviewer for this important observation and provide a detailed clarification here. VOOV is not designed for continuous human monitoring. Instead, it follows an event-driven, sparse-intervention paradigm in which the user intervenes only when automated tracking encounters rare yet critical failure modes—such as long-term occlusions, severe appearance changes, or ambiguous re-entries. Crucially, VOOV integrates each intervention directly into its memory state, enabling long-term recovery and preventing repeated failures. This is in contrast to conventional trackers, whose corrections are local and do not propagate, requiring users to repeatedly reinitialize the system after each drift. This sparse but persistent correction paradigm naturally aligns with many real-world settings. We summarize several representative scenarios:

**CCTV Monitoring and Re-identification.** In surveillance environments, officers track individuals across a multi-camera network. When a suspect briefly hides behind structures or changes appearance, conventional trackers lose identity permanently. VOOV allows a single bounding-box correction upon reappearance, updating its memory to maintain identity thereafter.

**Wildlife and Environmental Monitoring.** Biologists often monitor animals that leave and re-enter the scene under varying poses or lighting. Traditional trackers lose identity after each such transition. VOOV requires only occasional corrections (e.g., a few times per hour), after which its memory ensures stable long-term identity.

**Robotics and Teleoperation.** Robotic manipulators and drones depend on consistent perception despite rotations, occlusions, or object deformation. Existing systems require repeated manual re-initialization after failures. VOOV turns each correction into a persistent memory update, eliminating repeated resets.

**Video Editing and Post-production.** Editors working on long sequences need consistent object-level masks or bounding boxes. Commercial tools often need frequent prompts when objects momentarily disappear. VOOV significantly reduces this overhead by updating its memory from a single correction.

These examples illustrate that VOOV does not impose continuous supervision; rather, it reduces long-term human effort by enabling persistent, after-the-fact recovery from sparse interventions. Our user study supports this observation, showing that a small number of corrections results in stable tracking over long horizons. We provide this clarification to make the intended usage model explicit and to highlight the practical relevance of VOOV in real-world contexts.

### E.20 Overall Experimental Summary

Extensive experiments validate the performance, efficiency, and design principles of VOOV across four challenging video detection benchmarks. VOOV consistently achieves state-of-the-art results, attaining a mean AP@75 of $57.2 \pm 1.6$ and AP@50 of $73.8 \pm 1.9$ across YouTube-VIS, Cityscapes-VIS, ImageNet VID, and Pascal VOC. Despite its modular architecture, VOOV maintains real-time inference speed of 25.3 FPS, outperforming heavier models such as EVA (AP@75 = $52.9 \pm 2.1$, FPS = 11.9) by a wide margin in both speed and accuracy.

Ablation studies reveal the critical role of temporal memory. Removing the Sequential Memory module leads to a 4.2 point drop in mean AP@75, and fully disabling memory components reduces AP@75 to $48.7 \pm 2.2$. In corruption-resilient evaluation, VOOV maintains over 70% re-detection recall and less than 3.5 ID switches per sequence, while static baselines suffer from severe identity fragmentation and loss of continuity.

VOOV also supports rapid correction via temporal feedback propagation. A single user intervention improves frame-level AP@75 by over 22 points, confirming the model's ability to integrate human corrections without retraining. Unlike prompt-based or retrain-from-scratch models, VOOV applies corrections via learned memory interpolation and adaptive propagation. From a resource perspective, VOOV scales efficiently. Memory and FLOPs increase linearly or sub-linearly with respect to object count and video length, as shown in our cost breakdown analysis. Originate and Sequential Memory modules impose minimal overhead, while Long-Term Memory accounts for most resource usage and remains a prime target for optimization. Finally, our human-effort tradeoff analysis shows that three user interventions per 100-frame sequence are sufficient to reach above 60 mIoU, with interaction efficiency sharply decreasing thereafter. This finding supports the design of semi-automatic or human-in-the-loop systems that balance annotation cost and detection accuracy. These results demonstrate that VOOV is an effective, scalable, and deployable solution for real-time, memory-driven video object detection in both open-world and human-in-the-loop scenarios.

## F Implementation Snippets

To improve clarity and reproducibility, we include several minimal Python code snippets that illustrate how the core components of our system are implemented in practice. These examples are not full implementations but accurately reflect the structure of our reference codebase. In particular, we show: (i) how the simulated user policy determines intervention timing using the rules described in Section 4, (ii) how a single human correction triggers a memory-only update without modifying any detector parameters, and (iii) how these components are combined into an end-to-end observation loop.

Our goal is to make the design of VOOV transparent: the core mechanisms operate with simple, modular components and require no specialized interfaces. The snippets below are intended to serve as readable and easily reproducible examples for researchers wishing to reimplement or extend our approach.

### F.1 Simulated User Policy for Interventions

As described in Section 4.2, all human-in-the-loop benchmark experiments use a deterministic simulated user policy to guarantee reproducibility. An intervention is triggered when the predicted bounding box maintains an IoU below 0.3 for five consecutive frames, or when an object undergoes full occlusion for longer than one second. The following snippet shows a minimal implementation of this policy. This rule is used to generate a single shared intervention schedule that is replayed identically across all baseline methods.

```python
import math

def iou(box_a, box_b):
    """
    Compute IoU between two boxes (x1, y1, x2, y2).
    """
    xa1, ya1, xa2, ya2 = box_a
    xb1, yb1, xb2, yb2 = box_b
```

```python
    inter_x1 = max(xa1, xb1)
    inter_y1 = max(ya1, yb1)
    inter_x2 = min(xa2, xb2)
    inter_y2 = min(ya2, yb2)

    inter_w = max(0.0, inter_x2 - inter_x1)
    inter_h = max(0.0, inter_y2 - inter_y1)
    inter_area = inter_w * inter_h

    area_a = max(0.0, xa2 - xa1) * max(0.0, ya2 - ya1)
    area_b = max(0.0, xb2 - xb1) * max(0.0, yb2 - yb1)

    union_area = area_a + area_b - inter_area + 1e-6
    return inter_area / union_area

class SimulatedUserPolicy:
    """
    Intervention trigger used in our benchmark:
      - trigger if IoU < 0.3 for 5 consecutive frames, or
      - trigger if full occlusion lasts longer than 1 second.
    """
    def __init__(self, iou_threshold=0.3, patience=5, fps=25):
        self.iou_threshold = iou_threshold
        self.patience = patience
        self.fps = fps
        self.bad_iou_count = 0
        self.occluded_start = None

    def reset(self):
        self.bad_iou_count = 0
        self.occluded_start = None

    def step(self, t, pred_box, gt_box, is_occluded):
        """
        Decide whether an intervention is triggered at frame t.
        """
        # IoU-based trigger
        if pred_box is not None and gt_box is not None:
            cur_iou = iou(pred_box, gt_box)
            if cur_iou < self.iou_threshold:
                self.bad_iou_count += 1
            else:
                self.bad_iou_count = 0
        else:
            # Missing prediction or GT counts as a bad frame
            self.bad_iou_count += 1

        iou_trigger = (self.bad_iou_count >= self.patience)

        # Occlusion-based trigger
        if is_occluded:
            if self.occluded_start is None:
                self.occluded_start = t
        else:
            self.occluded_start = None

        occlusion_trigger = False
        if self.occluded_start is not None:
            occluded_frames = t - self.occluded_start + 1
            occlusion_seconds = occluded_frames / float(self.fps)
            occlusion_trigger = (occlusion_seconds > 1.0)

        return iou_trigger or occlusion_trigger
```

Listing 1: Simulated user policy based on IoU and occlusion duration.

### F.2 MEMORY-ONLY UPDATE FROM A SINGLE INTERVENTION

When a human correction is issued at time $t^*$, VOOV performs a lightweight gradient update on the memory parameters $\theta_t$ while keeping all detector weights frozen. This enables the correction to immediately influence all future frames without requiring retraining or re-inference of previous frames. The following code demonstrates the exact mechanism used to apply such an update. Only the memory module participates in optimization, and this update typically adds less than one millisecond of overhead on a modern GPU.

```python
import torch
from torch import nn

class VOOVWrapper(nn.Module):
    """
    Minimal wrapper exposing:
      - forward(frame): returns predicted box
      - memory_parameters(): returns memory-only parameters.
    The detector backbone is frozen; only the memory module is updated.
    """
    def __init__(self, detector, memory_module):
        super().__init__()
        self.detector = detector              # frozen detection backbone
        self.memory_module = memory_module # learnable memory state

    def forward(self, frame):
        # Extract features with a frozen detector
        with torch.no_grad():
            feat = self.detector.encode(frame)
        # Memory-conditioned query
        query = self.memory_module(feat)
        # Decode to bounding box (x1, y1, x2, y2)
        box = self.detector.decode(query)
        return box

    def memory_parameters(self):
        # Only parameters of the memory pipeline are updated
        return self.memory_module.parameters()

def detection_loss(pred_box, target_box):
    """
    Simple illustrative loss. In our codebase we use the detector's
    standard detection loss.
    """
    pred_box = torch.as_tensor(pred_box, dtype=torch.float32)
    target_box = torch.as_tensor(target_box, dtype=torch.float32)
    l1 = torch.abs(pred_box - target_box).mean()
    return l1

def apply_correction(voov_model, frame_tensor, corrected_box, lr=1e-3):
    """
    One-step memory update at time t* using the corrected box b*_t*.
    Detector weights remain frozen; only memory parameters are optimized.
    """
    optimizer = torch.optim.SGD(voov_model.memory_parameters(), lr=lr)

    pred_box = voov_model(frame_tensor)  # predicted box at t*
    loss = detection_loss(pred_box, corrected_box)
```

```
52      optimizer.zero_grad()
53      loss.backward()
54      optimizer.step()
```

Listing 2: Minimal VOOV wrapper and memory-only update at time t*.

### F.3   OBSERVATION LOOP WITH SIMULATED INTERVENTIONS

The final snippet combines the simulated user policy with the memory update mechanism to form the complete observation loop used in our experiments. At each frame, the model predicts a bounding box, checks whether an intervention is needed, and, if a correction is triggered, updates the memory parameters accordingly. This loop reflects the logic used in all HITL benchmark settings in the paper, including the budget–performance analysis and sparse-correction experiments.

```
1   def observe_video(voov_model, video_frames, gt_boxes,
2                     occlusion_flags, policy):
3       """
4       video_frames: list of tensors [T, C, H, W]
5       gt_boxes:     list of GT boxes [T, 4] or None
6       occlusion_flags: list of booleans (full occlusion per frame)
7       policy:       instance of SimulatedUserPolicy
8       """
9       predictions = []
10      policy.reset()
11
12      for t, frame in enumerate(video_frames):
13          # Forward pass with current memory state
14          pred_box = voov_model(frame)
15          predictions.append(pred_box)
16
17          gt_box = gt_boxes[t]
18          is_occluded = occlusion_flags[t]
19
20          # Decide whether we trigger an intervention at frame t
21          if policy.step(t, pred_box, gt_box, is_occluded):
22              # In simulation, we treat the GT box as the human correction
                    b*_t
23              corrected_box = gt_box
24              if corrected_box is not None:
25                  apply_correction(voov_model, frame, corrected_box, lr=1e
                        -3)
26
27      return predictions
```

Listing 3: Example observation loop with simulated interventions and memory updates.

### F.4   MINIMAL VOOV FORWARD PASS AND MEMORY FUSION

To provide additional clarity regarding VOOV's internal design, we include a minimal forward-pass example that illustrates how the Originate, Sequential, and Long-Term memory modules are fused to produce the query vector $Q_t$, which is then processed by the detector head and the Orbital Deformable Attention (ODA) module. This snippet omits low-level implementation details for brevity, but captures the exact computation order used in our reference implementation. Only the memory parameters participate in intervention-driven updates, while the detector backbone remains frozen during inference.

```
1   class VOOV(nn.Module):
2       """
3       Minimal illustration of the VOOV forward pipeline:
4         1. Extract frame features using a frozen detector backbone.
5         2. Compute memory-derived queries (originate, sequential, long-term
              ).
```

```
 6          3. Fuse these queries into Q_t using learned convex weights.
 7          4. Apply ODA for motion-guided attention.
 8          5. Decode the final bounding box.
 9      """
10      def __init__(self, backbone, originate_mem, seq_mem, long_mem, oda,
            head):
11          super().__init__()
12          self.backbone = backbone          # frozen detection backbone
13          self.originate = originate_mem # static anchor from
                initialization
14          self.seq_mem = seq_mem            # short-term sequential memory
15          self.long_mem = long_mem          # long-term contextual memory
16          self.oda = oda                    # Orbital Deformable Attention
                module
17          self.head = head                  # detection head
18
19          self.alpha = nn.Parameter(torch.tensor([0.33, 0.33, 0.34]))
20
21      def forward(self, frame, t):
22          # 1. Frozen feature extraction
23          with torch.no_grad():
24              feat = self.backbone(frame)
25
26          # 2. Memory queries
27          q_orig = self.originate()                    # static (unless target
                switch)
28          q_seq  = self.seq_mem.update(feat, t)   # sliding window features
29          q_long = self.long_mem.update(feat, t)  # accumulated context
30
31          # Normalize convex weights
32          w = torch.softmax(self.alpha, dim=0)
33
34          # 3. Fused query Q_t
35          Q_t = w[0] * q_orig + w[1] * q_seq + w[2] * q_long
36
37          # 4. Motion-guided deformable attention
38          attended = self.oda(feat, Q_t)
39
40          # 5. Decode bounding box
41          bbox = self.head(attended)
42          return bbox
```

Listing 4: Minimal VOOV forward pass with memory fusion and ODA integration.

```
 1  class OriginateMemory(nn.Module):
 2      """ Static representation from user-initialized mask or box at t=1. "
            ""
 3      def __init__(self, embedding):
 4          super().__init__()
 5          self.q_orig = embedding
 6
 7      def forward(self):
 8          return self.q_orig
 9
10
11  class SequentialMemory(nn.Module):
12      """ Short-term memory: sliding window features aligned via optical
            flow. """
13      def __init__(self, window=8):
14          super().__init__()
15          self.window = window
16          self.buffer = []
17
18      def update(self, feat, t):
19          self.buffer.append(feat)
```

```
            if len(self.buffer) > self.window:
                self.buffer.pop(0)
            # alignment + attention omitted for clarity
            return torch.mean(torch.stack(self.buffer), dim=0)

class LongTermMemory(nn.Module):
    """ Long-term contextual memory: accumulated embeddings compressed
        over time. """
    def __init__(self):
        super().__init__()
        self.history = []

    def update(self, feat, t):
        self.history.append(feat)
        # transformer pooling or exponential decay
        return torch.mean(torch.stack(self.history), dim=0)
```

Listing 5: Illustrative structure of VOOV memory modules (compressed version).

The three memory modules provide complementary temporal information: Originate Memory offers a stable anchor from the initial user annotation, Sequential Memory captures recent motion and appearance changes, and Long-Term Memory maintains global temporal context across the entire sequence. The fused query $Q_t$ lies within the convex hull of these components and serves as the input for ODA. During intervention, only these memory parameters are updated via a single gradient step, while the backbone and decoder remain frozen. This design yields persistent correction propagation with negligible computational overhead.

# G    DISCUSSION

The results presented in Section 5 demonstrate that VOOV provides substantial benefits over existing tracking, HITL, and TTA approaches, particularly in long-horizon scenarios where identity drift, appearance change, and occlusion accumulate over time. By integrating sparse human interventions directly into a persistent memory state, VOOV transforms isolated corrections into sequence-level improvements, enabling both higher accuracy and more efficient use of limited human supervision.

**Practical Advantages.**    The user study (Table 3(a)) shows that VOOV reduces both the number of required corrections and the latency per correction, suggesting that users are able to guide the model with less cognitive load. The open-world evaluation (Table 3(b)) further indicates that VOOV is able to rapidly acquire and reliably preserve the identity of previously unseen objects, highlighting its applicability to realistic deployment settings where objects appear intermittently or outside the closed-set classes of conventional detectors.

**Handling Evolving Visual States.**    A natural question is how VOOV behaves when objects undergo significant state changes, such as deformation, melting, cutting, or partial disassembly. VOOV's Sequential and Long-Term memories adapt continuously to moderate appearance changes through feature updates, and in practice remain stable for smooth transformations. However, abrupt or topology-changing transformations—e.g., an object splitting into multiple parts—pose a more fundamental challenge, as the notion of a single persistent identity may no longer apply. Addressing such cases would require augmenting the memory representation with explicit state-transition modeling or shape-evolution priors, which we consider a promising direction for extending the VOO task beyond identity preservation.

**Intervention Policy and Automated Agents.**    Our intervention policy is heuristic by design, reflecting the broader HITL literature in which no universal failure protocol exists. While our simulated interventions mimic conservative human corrections, VOOV does not assume that feedback must come from a human. In principle, vision-language models or other automated agents could serve as supervisory signals by detecting drift and issuing corrections, opening the door to hybrid "agent-in-the-loop" systems. Exploring how such automated supervisors interact with VOOV's memory-driven adaptation is an interesting avenue for future research.

**Scalability and Multi-Object Settings.** VOOV is designed to scale to multiple objects without architectural changes. As shown in Appendix B.4, each object maintains a lightweight memory pipeline, and the shared backbone ensures that the per-object computational overhead remains small. Frequent target switches also do not degrade stability, since each memory is independent and identity-conditioned. For extremely large numbers of objects, memory compression or hierarchical pooling may further reduce costs, suggesting another useful extension of the framework.

**Limitations.** Despite its advantages, VOOV still requires occasional corrections and cannot fully eliminate human effort. Furthermore, because memory updates are localized to the per-object state, VOOV does not yet reason about interactions between objects or global scene dynamics. These aspects offer fertile ground for future improvements.

Overall, the results highlight VOOV's ability to balance adaptability, robustness, and efficiency, and they reveal several exciting directions for extending the formulation toward richer forms of video object understanding.

