# OpenReview forum: "Observe Anything: Human-Intervened Video Understanding with Adaptive Orbital Memory"
_ICLR.cc/2026/Conference — Submitted to ICLR 2026_

### Official Review · Reviewer_X9jh · 2025-10-16

**Soundness:** 2
**Presentation:** 1
**Contribution:** 2
**Rating:** 2
**Confidence:** 4

**Summary:**

This paper introduces **Video Object Observation (VOO)**, a new task that unifies object detection, tracking, and human intervention to maintain continuous object identity in videos. To address this, the authors propose **VOOV (Video Object Observer with human-interVention)**, a novel framework designed to handle common tracking failures like long-term occlusion and abrupt appearance changes. VOOV's core contribution is its architecture, which features three complementary memory modules—**Originate, Sequential, and Long-Term**—to encode an object's identity and temporal context at different scales. It also introduces an **Orbital Deformable Attention** mechanism that probabilistically models object motion to better predict location. The framework allows sparse human feedback, such as bounding box corrections, to be integrated directly into its memory modules in real-time without needing to retrain the model, enabling corrections to persist and influence future frames. Experiments show that VOOV achieves state-of-the-art performance, providing robust and efficient observation across diverse and challenging scenarios.

**Strengths:**

- The paper introduces Video Object Observation (VOO), a new task that formally integrates detection, tracking, and human intervention for continuous identity preservation.
- The proposed VOOV framework features an new architecture that combines a hierarchical memory system with a probabilistic, motion-aware attention mechanism for robust temporal modeling.
- The model efficiently integrates sparse human feedback as lightweight, test-time memory updates, enabling real-time adaptation and persistent correction propagation without retraining.
- The work is supported by a comprehensive evaluation across five benchmarks, including extensive comparisons, ablation studies, and a user study, which collectively validate the framework's state-of-the-art performance and practical effectiveness.

**Weaknesses:**

- First of all, the paper's title, "Video Understanding" and "Observe Anything," is overclaimed. Video understanding encompasses many different levels, but this work focuses only on low-level tracking. This raises doubts about whether the work is framed under the correct category and scope. The title could be significantly improved by including the term "video object tracking with human intervention," which is clearer and fits the work's scope. The reviewer is quite concerned about the intention of using such a broad and high-level title in this submission.

- The proposed task does not seem realistic to me in terms of real-world applications. It is unclear in what scenario the proposed VOOV could actually be used. If a human is required to constantly focus on the video, checking if the algorithm has tracked the wrong target and providing intervention (or correction), what is the actual purpose of the method?

- The paper benchmarks and compares its method with multiple "detector," "single object tracker," and "multi-object tracker" methods. All these methods are designed for different tasks and have no ability to leverage human intervention. This raises doubts about the fairness of the comparison, especially when these methods are tested on VOS-type benchmarks.

- Line 361 states: "Interventions are triggered when predictions fall below IoU 0.3 for five consecutive frames or during occlusion >1s, and are applied identically across methods." It is unclear how this intervention is applied to existing methods, such as a detector like YOLOv8.

- It is unclear to me how multiple baselines are implemented, especially methods like SAM2-prompt. I also found a technical statement error regarding SAM2 in L.2058: "In contrast, SAM2 performs prompt-based inference, requiring a bounding box prompt...Since SAM2 does not maintain a memory state or temporal continuity, any correction must be explicitly reapplied at each frame." This is not correct, as SAM2 leverages a memory bank to store object features across the temporal dimension. This incorrect technical statement alone raises a big concern about the validity of the implemented baseline, specifically SAM2-prompt, especially given the missing implementation details.

- Diving further into Table 10, the reported FPS (3.2) for SAM2 shows a tenfold discrepancy with the FPS reported by the official paper (30+ on a single GPU). This adds another extra layer of concern regarding the rigor of the experiments in this paper.

---

Considering the overclaimed scope in the title, unclear motivations, missing baseline details, technical errors in the paper, concerning experimental results especially incorrect understanding of the baseline method, and large discrepency on the reported FPS, I lean toward rejection.

**Questions:**

- How does the work ensure the experiments are reproducible? Every human can provide the intervention at different timestamp.
- More details on how the baselines are implenented, especially detector and SAM2.
- Some other concerns (task motivations, paper title, SAM2...etc) mentioned in weakness.

---

> ### Author Response · Authors · 2025-11-28
> **Response to W1 (First of all, the paper's title, "Video Understanding"...) [1/9]**
>
> We appreciate the reviewer’s comment regarding the potential overclaim in the title. We agree that “video understanding” can indeed encompass a much broader scope than the specific problem addressed in our work. Our intention was not to imply coverage of the full spectrum of video understanding tasks, but rather to emphasize that the proposed framework goes beyond frame-wise tracking by incorporating identity-aware temporal reasoning, human-guided correction, target switching, and recovery from occlusion within a unified observation pipeline.
>
> Therefore, we understand how the current phrasing may reasonably be interpreted as broader than intended when viewed without this context. The phrase “Observe Anything” was meant to highlight the object-agnostic and user-steerable nature of the VOO task, not to overstate general video understanding capability. To avoid ambiguity, we will revise the __1. Introduction__ to explicitly clarify how observation differs from broader notions of video understanding as below:
>
> > In this work, we use the term observation in a focused sense: maintaining the identity of a user-specified object continuously over time, even as it appears, disappears, or changes. This includes localization, user-guided target switching, correction propagation, and recovery from occlusion or drift. Our goal is not to suggest broad semantic video understanding, but to describe the class-agnostic, identity-consistent process required by the VOO setting. In this context, observe anything refers to the ability to maintain and recover any visually indicated object under challenging video conditions, rather than implying general-purpose video understanding.
>
> We hope that this clarification makes our intended scope clear. Further, if possible by AC, we are open to adjusting the title in the camera-ready version to better reflect the scope (e.g., Video Object Observation: Human-Intervened Long-Term Tracking with Adaptive Memory). We thank the reviewer for pointing this out, and we believe the suggested clarification will strengthen the framing of the work.

---

> ### Author Response · Authors · 2025-11-28
> **Response to W2 (The proposed task does not seem...) [2/9]**
>
> We appreciate the reviewer’s thoughtful concern regarding the real-world practicality of VOOV and the impression that it may require continuous human supervision. We would like to clarify that VOOV is not designed for constant monitoring. Rather, it is built for event-driven, sparse human corrections, where a human intervenes only when an automated tracker encounters rare failure modes such as long-term occlusion, drastic appearance change, or heavy scene clutter.
>
> To better illustrate why such a capability is valuable, we highlight several concrete real-world scenarios where persistent identity matters, where existing trackers are known to fail, and where a single, well-timed human correction can dramatically reduce long-term workload:
>
> __(1) CCTV monitoring and re-identification assistance.__
> Consider a police officer monitoring a suspect through a large network of cameras. If the suspect temporarily hides behind obstacles, swaps clothing, or changes appearance, conventional trackers often lose the track permanently, forcing the officer to manually re-localize the person each time the model drifts. In contrast, VOOV allows a single correction (e.g., bounding box adjustment when the suspect reappears) to update the memory state, enabling the system to continue tracking reliably afterward without repeated intervention.
>
> __(2) Long-term wildlife or environmental monitoring.__
> Biologists frequently track individual animals that move irregularly, leave and re-enter the scene, or change appearance due to lighting or posture. Automated trackers routinely fail in such settings. With VOOV, a researcher can intervene sparsely (e.g., a few times per hour) and the model will maintain that identity robustly over extended periods.
>
> __(3) Robotics and teleoperation systems.__
> Robotic manipulators or drones often rely on persistent tracking of an object of interest. If the object becomes occluded or changes state (e.g., being rotated or deformed), existing systems may require re-specification of the target. VOOV turns a single operator correction into a stable memory update, preventing repetitive reinitialization.
>
> __(4) Professional video editing and post-production.__
> Editors frequently need object-level masks or bounding boxes over long clips. When an object goes out of view or changes appearance, popular commercial tools often require the user to re-prompt frequently. Under VOOV, one correction directly updates the model’s memory, reducing repeated manual adjustments and significantly lowering annotation overhead.
>
> These examples reflect the intended deployment paradigm: the human is not supervising continuously but stepping in sparsely to correct rare but impactful failures. Unlike traditional trackers, which repeatedly drift under the same conditions, VOOV learns from the correction by updating its memory representation, preventing the same error from recurring. This results in a net reduction in human effort over long sequences, as confirmed in our user study.
>
> We have strengthened the manuscript to make this usage model explicit and added representative real-world examples to avoid any ambiguity in Appendix __E.19 Real-World Applicability and Human-Intervention Scenario__. We sincerely thank the reviewer for raising this point and for enabling us to clarify the practical motivation of VOOV more clearly.

---

> ### Author Response · Authors · 2025-11-28
> **Response to W3 (The paper benchmarks and compares...) [3/9]**
>
> We thank the reviewer for raising this important point about the fairness of our comparisons across detectors, single-object trackers, and multi-object trackers. We would like to clarify two aspects of our evaluation protocol: (1) VOOV is evaluated in a fully automatic (zero-intervention) mode, which is directly comparable to standard trackers and detectors; and (2) in the human-in-the-loop setting, we apply the same intervention schedule and latency model to all methods in order to study how effectively each architecture can benefit from sparse human input.
>
> First, VOOV is not exclusively a human-in-the-loop system. It supports a zero-intervention setting (B = 0) where no human corrections are used at all. In this regime, VOOV runs fully automatically and is directly comparable to detectors and trackers that also operate without any human input. As described in Section 4.2 (“Fair Comparison Analysis”), we report results under three budgets: zero-intervention (B = 0), minimal intervention (B = 1 per 100 frames, ≈3s effort), and sparse intervention (B = 3, ≈9s). In particular, Figure 4(a) shows that without intervention, VOOV already matches or outperforms representative detectors and trackers in real time: “Figure 4 (a) shows the comparison analysis. Without intervention, VOOV already matches or outperforms detectors and trackers in real time.”
>
> Similarly, Appendix Table 25 reports results on YouTube-VOS under budgets (B \in {0,1,2,3,5,8,10}), and explicitly states that at zero intervention (B = 0), VOOV already surpasses state-of-the-art trackers and HITL baselines: “At zero intervention (B = 0), VOOV already surpasses state-of-the-art trackers and HITL baselines, demonstrating strong automated robustness.”
>
> These B = 0 results are pure “fully automatic” comparisons with no human interaction and are therefore methodologically fair with respect to detectors and trackers, even though these methods were originally introduced for slightly different tasks.
>
> Second, in the human-intervened regime, our goal is not to claim that detectors or standard trackers are “designed for HITL,” but rather to study what happens if all methods are given access to the same sparse human corrections. To this end, we use a common, method-agnostic intervention policy and apply it identically across all models. As described in Section 4.2: We report results under three budgets: zero-intervention (B = 0), minimal intervention (B = 1 per 100 frames, ≈3s effort), and sparse intervention (B = 3, ≈9s). Interventions are triggered when predictions fall below IoU 0.3 for five consecutive frames or during occlusion > 1s, and are applied identically across methods. Effective FPS accounts for a 0.4s latency per correction (Ravi et al., 2024b), ensuring fairness.”
>
> For detector-based baselines (e.g., YOLOv8, DETR) and trackers (e.g., ByteTrack, TrackFormer, MOTRv2, MeMOTR), a human correction is simulated as a bounding-box re-specification at the intervention frame, which mirrors how these methods are used in practice when an operator corrects a drifting or lost track. For HITL-style baselines such as SAM2-Prompt and DAM4SAM, we follow their prompt-based or mask-based interaction protocol. In all cases, corrections are triggered at the same frames and incur the same latency overhead in the Effective FPS metric. Thus, the comparison in the interactive setting is not between “VOOV with humans” and “baselines without humans,” but between different architectures given the same amount and timing of human help.
>
> The main purpose of the human-in-the-loop experiments is therefore to ask: *Given an identical budget of sparse interventions, how effectively can each method convert human feedback into long-term improvements?* Our results show that detectors and conventional trackers, even when allowed to be reinitialized at the same intervention frames, tend to treat each correction as a one-off reset and fail to propagate the benefit forward, whereas VOOV’s memory updates enable persistent identity stabilization. This is reflected in metrics such as ID consistency, re-detection rate, and the intervention-free rate (the percentage of frames that did not require re-specification), which we report alongside mAP@75 and FPS.
>
> Regarding the use of VOS-type benchmarks: although detectors, trackers, and VOOV originate from different task formulations, long-term VOS-style datasets (e.g., sequences with heavy occlusion, re-entry, and deformation) are precisely where persistent object observation is most challenging and where the VOO task is meant to operate. These datasets have also been widely used to evaluate long-term tracking and memory-based models. By reporting both (i) the zero-intervention (fully automatic) performance and (ii) the budgeted-intervention performance under a common trigger and latency model, we ensure that the evaluation is transparent and fair while also highlighting the practical benefits of VOOV’s human-intervened capability.

---

> ### Author Response · Authors · 2025-11-28
> **Response to W4 (Line 361 states: "Interventions are triggered...) [4/9]**
>
> We appreciate the reviewer’s question regarding how our intervention triggers are defined and, in particular, how these interventions are applied to baselines such as YOLOv8, which do not natively support human input. Our intervention protocol is grounded in conventions widely adopted in interactive annotation, HITL-based detection, and long-term tracking research, and we clarify the full rationale here. For IoU-based triggers, we follow prior literature such as Papadopoulos et al. (CVPR 2017), which systematically studies annotation quality across IoU ranges and identifies IoU < 0.3 as “almost no overlap,” corresponding to errors that human annotators reliably correct. Using this convention, we trigger an intervention only when IoU remains below 0.3 for five consecutive frames, ensuring that the system reacts to persistent identity drift rather than transient jitter. Consistent with recent HITL systems (e.g., Tenckhoff et al., 2025), our thresholds are heuristic by design: the HITL community does not have a standardized protocol for defining when automated systems should request assistance, and prior works similarly rely on practical, error-based heuristics such as overlap degradation, confidence drops, or temporal incoherence. For occlusion-based intervention, we reference widely used conventions from MOT17/MOT20, where visibility ratios below approximately 0.10–0.15 are treated as heavy occlusion, a regime in which automated trackers frequently lose identity. Motivated by this, we trigger an intervention when a target remains heavily occluded for more than one second, which is a long enough for identity to be reliably lost in both tracking and VOS settings.
>
> To address how interventions are applied to existing baselines, our design principle is to simulate the natural form of correction that each model family would realistically support if used in annotation or supervision workflows. For detectors such as YOLOv8 and DETR, a human correction is implemented as a bounding-box re-specification at the intervention frame; this mimics how detectors are used in semi-automatic annotation tools, where an operator adjusts the bounding box when the model fails. For single-object and multi-object trackers (e.g., ByteTrack, TrackFormer, MOTRv2, MeMOTR), we inject the corrected bounding box directly into the tracker’s current state, equivalent to manually resetting the track when drift occurs. For interactive models like SAM2-Prompt and DAM4SAM, we follow their native box or mask prompt interfaces. Importantly, all methods receive interventions at the exact same frames, all corrections represent the same semantic signal (a single bounding-box adjustment), and all models incur the same fixed latency penalty in Effective FPS (0.4s per correction), ensuring cross-method fairness.

---

> ### Author Response · Authors · 2025-11-28
> **Response to W5 (It is unclear to me how multiple baselines...) [5/9]**
>
> We thank the reviewer for the careful reading of our manuscript and for highlighting the misleading description of SAM2’s temporal mechanism. We fully agree that our phrasing in Line 2058 was imprecise. SAM2 indeed maintains a temporal memory bank that stores object-specific features and relies on this memory for video propagation. Our intent was not to claim that SAM2 lacks memory, but rather to emphasize that SAM2 does not support *trainable, test-time adaptive memory updates* in response to human corrections, in contrast to VOOV’s gradient-based memory update mechanism. We acknowledge that our wording could be interpreted as denying the existence of SAM2’s temporal memory altogether, and we will correct this technical inaccuracy in the revised manuscript.
>
> Importantly, this wording issue does not affect the validity of our experimental results, as our implementation of SAM2-prompt faithfully follows the official SAM2 video inference pipeline and fully utilizes its memory system. Concretely, we initialize SAM2 with a user-provided bounding box at t = 1; following the official procedure, we convert this box into an initial mask prompt, which creates SAM2’s memory tokens and activates its temporal propagation mechanism. For all subsequent frames, SAM2 is run in its *video mode* with the memory bank continuously updated and used for attention-based temporal aggregation. Predictions are obtained as binary masks from SAM2’s decoder, and for evaluation consistency we convert these masks into tight axis-aligned bounding boxes using standard postprocessing. No part of SAM2’s internal memory or propagation logic was disabled, simplified, or replaced.
>
> When a human correction occurs at frame t*, we apply the correction by providing a new bounding box prompt, again converting it into a corrected mask according to SAM2’s official prompting interface. This corrected mask is then appended to SAM2’s memory bank, and SAM2 continues video inference from frame t* onward using its own temporal memory mechanism. We do not reset, flush, or overwrite the memory bank in any of the baseline experiments; our implementation strictly follows SAM2’s documented inference behavior. The only conceptual distinction from VOOV is that SAM2’s memory cannot be updated via gradient-based optimization driven by user feedback, and thus corrections do not modify the internal representation beyond the injected prompt itself.
>
> Note that we employed the SAM2’s official GitHub to train and test its performance (https://github.com/facebookresearch/sam2). To avoid further confusion, we will revise the relevant sentences in the main paper and add a clear description of our SAM2-prompt implementation to Appendix E, detailing each step of initialization, prompt construction, correction handling, mask-to-box conversion, and memory usage. We believe this resolves the reviewer’s concern and ensures full transparency and correctness of the baseline.
>
> To clarify the statement, we will modify __L2058__ as below:
>
> > where $\theta_t$ is the internal representation and $\eta$ is the learning rate. This update modifies the memory pipeline without retraining any model parameters, and its effect propagates to all future predictions ${\hat{b}t}{t > t^*}$. In contrast, SAM2 performs prompt-based video inference: it accepts a user-provided prompt (e.g., a bounding box converted to a mask) and uses its own internal memory bank to propagate segmentations temporally. However, human corrections in SAM2 do not update its internal memory parameters via optimization; they act solely as new prompts provided at specific frames. Thus, a correction at time $t^{\ast}$ influences predictions only through the injected prompt, rather than through persistent adaptation of a learned memory state.
>
> Furthermore, to make these implementation details explicit and reproducible, we will extend __Appendix E.15 (Fair Comparison Analysis)__ with a short paragraph summarizing the above protocol, as below:
>
> > Detector baselines are implemented using official open-source code and trained under the same optimizer, batch size, input resolution, and GPU as VOOV. For each video, the target instance is selected at t = 1 by matching the user- or detector-provided initialization to the detector outputs via IoU, and subsequent frames are handled by frame-wise detection with nearest-IoU association for identity. SAM2-Prompt uses the official SAM2 video pipeline with its default memory bank. User corrections are provided as bounding-box prompts at the designated intervention frames and converted to masks internally by SAM2; the resulting masks are converted back to boxes for evaluation. We do not fine-tune SAM2 or alter its memory mechanism; interventions are used solely as prompts, in contrast to VOOV’s gradient-based memory updates.
>
> We appreciate the reviewer’s attention to detail. We believe that these corrections and expanded implementation details thoroughly resolve the concerns.

---

> ### Author Response · Authors · 2025-11-28
> **Response to W6 (Diving further into Table 10,...) [6/9]**
>
> We appreciate the reviewer’s careful observation regarding the FPS discrepancy and agree that additional clarification is necessary. The key reason for the difference is that the FPS values in Table 10 reflect effective throughput under the human-in-the-loop (HITL) evaluation protocol, rather than the pure model-only inference speed reported in the SAM2 paper. In our experiments, FPS is defined as end-to-end throughput that includes not only the model’s forward pass but also the latency incurred by user interactions, prompt construction, memory reactivation, and mask-to-box conversion. Each intervention adds a fixed delay following standard HITL annotation settings, and this delay is incorporated uniformly across all methods when computing FPS. Under this metric, methods that require frequent prompting naturally experience larger slowdowns, even if their raw inference speed is high.
>
> SAM2-Prompt, in particular, receives substantially more interaction-triggered overhead than automatic detectors or trackers. Every correction must be injected as a new mask prompt, and SAM2 must reactivate its video-memory pipeline for each such prompt. In crowded or multi-object scenarios, this overhead increases further because multiple objects require separate prompting. As a result, the effective FPS can drop to the 3–5 FPS range even though the underlying SAM2 model is capable of running much faster in pure forward-pass settings. Notably, when we evaluate SAM2 without human-interaction cost (e.g., in the open-world generalization table), the FPS is roughly an order of magnitude higher and consistent with reported speeds from the original SAM2 work.
>
> To avoid confusion, we will update our Appendix wording to explicitly clarify that FPS in Tables 10 denotes “effective FPS, including interaction latency”. We will add a brief parenthetical note indicating that SAM2 slowdowns are primarily due to repeated prompt handling rather than inference speed. No changes to the experimental protocol or results are needed; we will simply refine the terminology to ensure this distinction is clear.

---

> ### Author Response · Authors · 2025-11-28
> **Response to Q1 (How does the work...) [7/9]**
>
> We appreciate the reviewer’s question regarding reproducibility. Human-in-the-loop systems indeed pose a challenge because real users may intervene at different timestamps. For this reason, our main quantitative experiments do not rely on free-form human interaction, but instead follow the standard experimental protocol used in prior HITL and interactive vision work. Similar to interactive segmentation and interactive annotation studies (e.g., extreme clicking, interactive correction loops, human-guided fine-tuning), it is customary to replace human participants with a deterministic simulated user policy that triggers interventions according to a predefined rule. This approach is widely adopted specifically to guarantee reproducibility and ensure all baseline methods receive identical interventions.
>
> Following this established convention, our benchmark experiments use a fixed and fully deterministic policy: an intervention is triggered only when the predicted bounding box remains below a threshold IoU for a certain number of frames or when a long-term occlusion is detected. Given the model predictions and ground-truth annotations, this automatically produces a unique and reproducible sequence of intervention timestamps. For cross-model comparisons, the resulting intervention schedule is computed once and replayed identically for all methods, ensuring fairness and repeatability across baselines. Any researcher running the same code and seeds will obtain the exact same intervention sequence.
>
> The user study is treated separately and is intended only to measure human workload and usability, not to establish the quantitative ranking of methods. For this study, all participants follow the same instruction and interface, and all corrections (timestamps, bounding boxes, interaction counts) are logged, allowing recomputation of aggregate results from the raw interaction data. To further clarify this, we will make a minor update in the appendix explicitly stating that (i) benchmark experiments follow the widely adopted simulated-user protocol from prior HITL literature, (ii) deterministic schedules are shared across models, and (iii) user-study interactions are logged for reproducibility. We hope this clarification addresses the reviewer’s concern and situates our evaluation within the standard practices of HITL research.

---

> ### Author Response · Authors · 2025-11-28
> **Response to Q2 (More details on how the...) [8/9]**
>
> We appreciate the reviewer’s request for additional implementation details of the baselines, in particular for the detector family and SAM2-Prompt. In the current submission, these choices are mostly summarized in the Fair Comparison Analysis section, which unfortunately compresses several implementation decisions into a few sentences. We agree this can give the impression that the baselines are under-specified, and we will clarify this in the revision.
>
> For the detector baselines (YOLOv8, DETR, Deformable DETR, and related variants), we rely on their official open-source implementations and follow the recommended training recipes on our benchmarks. All detectors are trained and evaluated under the same hardware and optimization settings as VOOV, and we do not modify their architectures or loss functions. In the VOO setting, we focus on a single target object per sequence. At the first frame, the user-provided or bootstrap bounding box is matched to the detector’s predictions by IoU, and the highest-IoU detection is taken as the target instance. For subsequent frames, the detector is run frame-by-frame, and the target is tracked by selecting the detection whose box has the highest IoU with the previous target box and consistent category. This “track-by-detection” association is standard and ensures that detectors are evaluated fairly as strong frame-wise baselines that are given the same initial identity as VOOV, without restricting them to a weaker configuration.
>
> For SAM2-Prompt, we use the official released implementation in video mode and its default memory bank. Our baseline uses the recommended configuration for video segmentation, where SAM2 maintains an internal spatiotemporal memory to propagate masks over time. At initialization, the user-provided bounding box is converted to a mask prompt for SAM2 on the first frame, and the resulting mask is converted back to a tight bounding box for evaluation to match our detection metrics. During automatic segments of the sequence (B = 0), SAM2 simply runs its standard video pipeline and propagates its memory forward without additional prompts. Under human-intervened settings (B > 0), whenever an intervention is triggered at time t*, the user correction is provided to SAM2 as a new bounding-box prompt at that frame; SAM2 updates its internal memory using this prompt and proceeds with standard video inference afterwards. Importantly, the user feedback is only used as an input prompt to the official model; we do not modify SAM2’s parameters or internal memory update rules, and there is no gradient-based adaptation as in VOOV. This is precisely the design choice we aim to contrast: VOOV treats interventions as learnable updates to a dedicated memory representation, whereas SAM2-Prompt treats them as external prompts to a fixed video model.
>
> We will also clarify the earlier wording around SAM2 in the appendix. Our intent was not to claim that SAM2 lacks any temporal state, but rather that it does not expose a trainable memory representation that can be directly updated by loss-gradients from user corrections, as VOOV does. In the revision, we will adjust the text to explicitly acknowledge that SAM2 maintains an internal memory bank over time, while emphasizing that human feedback in SAM2-Prompt still operates purely at the prompt/input level and is not integrated into a persistent, learnable memory state.
>
> Finally, to make these implementation details explicit and reproducible, we will extend Appendix E.14 (Fair Comparison Analysis) with a short paragraph summarizing the above protocol, as below:
>
> “Detector baselines (YOLOv8, DETR, Deformable DETR, etc.) are implemented using official open-source code and trained under the same optimizer, batch size, input resolution, and GPU as VOOV. For each video, the target instance is selected at t = 1 by matching the user- or detector-provided initialization to the detector outputs via IoU, and subsequent frames are handled by frame-wise detection with nearest-IoU association for identity. SAM2-Prompt uses the official SAM2 video pipeline with its default memory bank. User corrections are provided as bounding-box prompts at the designated intervention frames and converted to masks internally by SAM2; the resulting masks are converted back to boxes for evaluation. We do not fine-tune SAM2 or alter its memory mechanism; interventions are used solely as prompts, in contrast to VOOV’s gradient-based memory updates.”
>
> We hope these clarifications make clear that (i) all baselines are implemented with their recommended, officially supported configurations, (ii) detectors are given a fair tracking protocol in the VOO setting, and (iii) SAM2-Prompt fully uses its video memory pipeline, with human feedback applied as prompts rather than as learnable memory updates.

---

> ### Author Response · Authors · 2025-11-28
> **Response to Q3 (Some other concerns...) [9/9]**
>
> We thank the reviewer again for raising the additional concerns regarding task motivation, paper title, and the SAM2 discussion. As detailed in our responses above, we have clarified the task framing, refined the title and motivation to better reflect the scope of our contribution, and corrected the technical description of SAM2 while expanding the implementation details of all baselines. We appreciate these comments, as they helped us significantly strengthen the clarity and presentation of the paper.

---

### Official Review · Reviewer_rRMv · 2025-10-22

**Soundness:** 3
**Presentation:** 2
**Contribution:** 2
**Rating:** 4
**Confidence:** 3

**Summary:**

This paper proposes the Video Object Observation (VOO) task and designs the VOOV (Video Object Observer with human-interVention) framework to address it. The core of VOOV lies in its hierarchical memory pipeline (comprising origin, sequence, and long-term memory) and its Orbit Deformable Attention (ODA) mechanism. The methodology aims to achieve retraining-free online adaptation by translating user interventions (e.g., bounding box corrections) into real-time gradient updates of memory embeddings. It is asserted that this approach effectively resolves long-term occlusion and identity drift issues.

Experiments were conducted on multiple video datasets, introducing metrics such as Interaction Efficiency (IE). The results demonstrate that VOOV, operating under a limited intervention budget, outperforms existing automated trackers and Human-in-the-Loop (HITL) benchmarks.

**Strengths:**

**Rigorous Experimental Validation**: The paper systematically substantiates the performance advantages of VOOV over State-of-the-Art (SOTA) benchmarks through comprehensive ablation studies (Section 4.1), fair comparative analyses (Section 4.2), and detailed failure case analyses (Section 4.4).

**High Interaction Efficiency**: The proposed Interaction Efficiency (IE) metric effectively quantifies the persistent improvements derived from user feedback, demonstrating the method's efficient utilization of sparse interventions.

**Robust Recovery Capability**: The data indicate that VOOV achieves a recovery rate surpassing 90% under conditions of severe occlusion and inter-object overlap, thereby strongly validating the practical efficacy of the memory-propagated interventions.

**Weaknesses:**

Ambiguous Task Demarcation: The VOO task (which unifies detection, tracking, and intervention) is fundamentally a natural extension of existing Human-in-the-Loop (HITL) Tracking, rather than a disruptive new paradigm. The concept of integrating interventions into memory, within the context of memory-based trackers (e.g., MeMOTR), constitutes an incremental improvement.

Omission of Relevant Comparisons: Concurrently, the design of memory mechanisms has been extensively studied in various tracking algorithms, particularly in domains such as Single Object Tracking (SOT) and long-video understanding. The paper fails to provide a comparison with relevant literature in these fields.

Implementation Concerns: The method's dependence on human information is likely to affect its practical implementation difficulty.

Recommendation: It is suggested that a figure be supplemented to illustrate the distinctions between the proposed task and preceding tasks.

**Questions:**

Given the rapid proliferation (or: accelerated development) of Multimodal Large Language Models (MLLMs), which have emerged as a relatively mature technology capable, to a certain extent, of performing tasks traditionally requiring human intervention, is it feasible (or: viable) to substitute the input of human-provided information with MLLMs?

---

> ### Author Response · Authors · 2025-11-28
> **Response to W1 (Ambiguous Task Demarcation: The VOO task...) [1/5]**
>
> We appreciate the reviewer’s thoughtful concern about the conceptual distinction between VOO and prior HITL tracking frameworks. We fully agree that VOO sits within the broader family of interactive video systems, but we would like to clarify that the VOO task is not a simple extension of existing HITL tracking. The key difference lies in how interventions are defined, consumed, and propagated.
>
> Traditional HITL tracking systems treat user feedback as a local, output-level correction where a bounding box is replaced or re-specified at a particular frame, and the tracker proceeds without altering its internal temporal state. As a result, the effect of a correction is short-lived and does not modify the underlying representation used in future predictions. Even memory-based trackers such as MeMOTR maintain temporally aggregated features, but these memory representations are fixed at inference time and cannot be updated by human feedback. A correction does not change the memory state; it simply provides a new reference output, after which the model continues its standard forward pass.
>
> In contrast, VOO formalizes intervention as a state-changing operation: an intervention is not merely a new bounding box, but a semantically meaningful update to the model’s internal memory representation, applied via lightweight test-time optimization. This results in what we call persistent, after-the-fact recovery: even if a drift or misidentification is noticed many frames later, a single correction updates the memory and immediately restores identity consistency in all subsequent frames without re-running earlier predictions or retraining model parameters. This form of persistent adaptation is not supported by prior HITL trackers or by memory-based architectures, whose memory states are not externally modifiable and do not admit runtime learning.
>
> While memory-based trackers aggregate information over time, they do so in a feed-forward, passive manner. VOO reframes the problem so that memory becomes an active, manipulable state that incorporates sparse external signals. This changes not only the algorithmic behavior but also the underlying task definition: detection, tracking, and intervention are no longer separate stages but become a single unified prediction problem with a dynamically evolving internal state. This capability that long-term propagation of sparse corrections, runtime modification of memory, and after-the-fact recovery, constitutes the core novelty of VOO and is not present in existing HITL or memory-based tracking paradigms.
>
> We will revise the introduction (See Comments to All Reviewers) to make this distinction clearer and provide more explicit contrasts to prior HITL formulations and memory trackers such as MeMOTR. We thank the reviewer again for encouraging us to highlight this conceptual contribution more clearly.

---

> ### Author Response · Authors · 2025-11-28
> **Response to W2 (Omission of Relevant Comparisons: Concurrently, ...) [2/5]**
>
> We appreciate the reviewer’s suggestion to more clearly situate our work within the broader literature on memory-based tracking, including single-object trackers and long-video understanding models. These areas indeed offer a diverse set of architectures that rely on temporal aggregation, template matching, or key–value memory banks, and our submission already discusses related mechanisms in several places, including Section 3.2 when introducing learnable memory states, Section 4.4 when analyzing the failure behaviors of TrackFormer and MeMOTR, and Appendix B.2–B.3 where we compare VOOV’s differentiable and optimizable memory with conventional template or propagation-based memory architectures. We agree, however, that these discussions are currently dispersed across the paper, which may obscure the conceptual position of VOO relative to prior work.
>
> It is also important to clarify that the memory structures used in existing SOT and long-video models serve a fundamentally different role from the memory formulation introduced in VOO. Prior memory-based trackers maintain passive temporal representations that accumulate or retrieve information over time but are fixed during inference and cannot be modified through external feedback. When a drift or failure occurs, these trackers are unable to update their internal state in response to human intervention, and any correction affects only an individual frame rather than propagating across future predictions. Their memory is descriptive but not adaptable, and thus cannot support the persistent correction behavior evaluated in our setting.
>
> In contrast, VOO defines memory as an actively updated internal state. Human intervention is interpreted as a gradient-based signal that modifies the memory representation itself, enabling persistent and after-the-fact recovery. Even when a misidentification is noticed many frames after it first occurs, a single correction updates the memory and immediately restores consistent identity in all future frames without retraining the model. This capability is not present in template-based SOT methods, propagation-based video segmentation models, or recurrent-memory trackers, all of which lack externally modifiable memory at inference time. For this reason, direct empirical comparison with such methods is not always aligned with the intended behavior of VOO, since those models are not designed to accept or propagate user corrections in the first place.
>
> We appreciate the reviewer’s feedback, and we are confident that integrating these clarifications into the main text will improve the presentation and more clearly convey how VOO and VOOV differ conceptually and functionally from both classical human-in-the-loop tracking and prior memory-based tracking frameworks.

---

> ### Author Response · Authors · 2025-11-28
> **Response to W3 (Implementation Concerns: The method's dependence...) [3/5]**
>
> We appreciate the reviewer’s concern about the practical difficulty of implementing a system that incorporates human information. We would like to clarify that VOOV is designed to be a fully functional automatic detector–tracker even without human input, and all human interactions are strictly optional. This is reflected in the B = 0 setting, where VOOV already outperforms strong automated baselines across multiple benchmarks. In practical deployments, VOOV does not require continuous monitoring or dense supervision; sparse interventions function only as a safety mechanism to correct rare failures such as long-term drift or re-identification errors. Rather than increasing implementation burden, this mechanism simplifies system design by providing a stable fallback path that prevents cascading failures without retraining or repeated reinitialization.
>
> In addition, the human-in-the-loop interface itself is intentionally minimal. VOOV accepts simple bounding-box corrections that are fully compatible with standard annotation tools and interactive segmentation UIs. No new UI components or custom pipelines are needed. A correction merely triggers a lightweight gradient update within the memory module, and this update is decoupled from the detector architecture and imposes negligible computational overhead. As a result, integrating VOOV into existing video systems requires no more engineering effort than adding a basic annotation callback.
>
> To further support reproducibility and ease of adoption, we are preparing an open-source release of the codebase and a project page that documents the full pipeline, including scripts for simulated interventions and example interactive loops. As our implementation is maintained at the research-lab level, we ask for understanding that packaging and releasing the complete system may require additional time. Nonetheless, we are committed to making the implementation publicly available. In the revision, we will also include a small code snippet in the appendix illustrating how a user correction triggers a memory update, so that the mechanism is transparent even prior to full code release.
>
> To avoid misunderstanding, we will refine the main text to emphasize that VOOV’s dependence on human input is not structural but opportunistic: the model operates fully automatically, and sparse corrections are used only to improve robustness in challenging scenarios. We believe this design reduces, rather than increases, the practical difficulty of deploying reliable long-term video systems. To further support transparency and ease of adoption, we have added the __F. Implementation Snippets__ in the appendix containing minimal code examples of the simulated user policy, the memory-only update mechanism, and the VOOV forward pass. These snippets illustrate how interventions are integrated into the memory pipeline in practice and demonstrate that the implementation can be reproduced with simple, modular components.

---

> ### Author Response · Authors · 2025-11-28
> **Response to W4 (Recommendation: It is suggested that a figure...) [4/5]**
>
> We thank the reviewer for the helpful suggestion. We fully agree that a visual illustration can make the distinction between VOO and preceding tasks much clearer. In the revised manuscript, we have added a dedicated section, __Appendix D.4 Illustrative Comparison of VOO with Prior Tasks__, which introduces a two-part figure contrasting our formulation with existing paradigms. The first panel visually compares frame-level detection, sequence-level tracking, and traditional HITL correction against the proposed VOO task using concrete video examples: prior methods either operate per-frame without temporal consistency, enforce temporal continuity but cannot recover from identity drift, or apply human corrections only at the output level without affecting future predictions. The second panel presents a structural diagram of the information flow, showing how VOO unifies detection, tracking, and intervention into a single memory-driven process in which human feedback updates a persistent memory state and thus influences all subsequent outputs. To summarize, these illustrations make the task demarcation more intuitive and explicitly highlight that VOO’s memory-integrated HITL formulation is qualitatively different from both conventional tracking and existing HITL approaches.

---

> ### Author Response · Authors · 2025-11-28
> **Response to Q1 (Given the rapid proliferation...) [5/5]**
>
> We appreciate the reviewer’s forward-looking question regarding the role of multimodal large language models (MLLMs). We agree that recent MLLMs have demonstrated notable progress and can assist with tasks that historically required human intervention. In the context of VOO, however, we view MLLMs not as a replacement for human input but as an optional agent that could operate within the same intervention interface.
>
> VOO assumes that interventions are sparse, high-precision corrections that must be temporally aligned and pixel-grounded. Current MLLMs, although capable, do not yet provide the level of frame-accurate localization reliability, identity consistency, or low-latency temporal stability required for such corrections, especially in long video streams. In practice, even strong MLLMs tend to hallucinate bounding boxes, lose spatial grounding across frames, or struggle with small-object or occlusion-heavy scenes. Because memory updates affect all future predictions, the correctness of the intervention signal is critical; unreliable inputs can degrade the memory state rather than improve it.
>
> To summarize, the VOO formulation is fully compatible with MLLM-provided interventions. The task does not assume that corrections must come from a human; it only requires that interventions are externally supplied and semantically valid. An automated MLLM agent could, for instance, monitor predictions, detect inconsistencies, and provide correction prompts, effectively creating an “LLM-in-the-loop” variant of VOO. We consider this a promising direction for future work and will explicitly mention this possibility in the discussion section.

---

### Official Review · Reviewer_fuVs · 2025-10-31

**Soundness:** 2
**Presentation:** 2
**Contribution:** 3
**Rating:** 6
**Confidence:** 3

**Summary:**

This paper introduces Video Object Observation (VOO), a new task that unifies object detection, tracking, and human intervention to address tracking failures caused by occlusion or appearance changes. The authors propose VOOV, a framework centered on a hierarchical memory system comprising Originate, Sequential, and Long-Term modules. The core mechanism is a "test-time adaptation" where human feedback (e.g., a bounding box correction) triggers a lightweight gradient update directly to the model's memory embeddings, rather than requiring full retraining. This allows corrections to propagate to subsequent frames, ensuring continuous identity preservation.

**Strengths:**

- The paper introduces VOO, a novel and well-motivated task. Formally unifying detection, tracking, and intervention addresses a practical limitation of fully automated systems that fail under real-world conditions.
- The method for handling human feedback is a key strength. Using test-time adaptation to apply a gradient-based update to memory embeddings is a clever and efficient solution that allows user corrections to propagate temporally.
- The model demonstrates strong empirical performance. The failure case analysis (Table 2), budget-performance tradeoff (Fig. 4b), and the inclusion of a real user study (Table 3a) effectively validate the framework's robustness and the efficiency of its intervention mechanism.

**Weaknesses:**

- I feel the paper does a poor job of reviewing related work. The main literature review is buried in Appendix D, so it's not clear how this builds on prior art. Also, in Section 2, the authors introduce these standard problem formulations like Eq. (1) and (2) without citing *any* of the key papers that proposed or use them. Consider moving the appendix content to the main paper and adding those citations
- The quantitative evaluation of the VOO task hinges on a *simulated* human user. However, the criteria for this simulation (triggering at "IoU < 0.3 for five consecutive frames or during occlusion > 1s" are presented without justification. Could the authors explain how the threshold is chosen, and how occlusion is decided? Do they follow any prior work to establish this as a standard protocol for evaluating human-in-the-loop tracking?
- Can the proposed framework handle object state changes (such as butter melting, cutting an apple). Some discussion on this direction would be helpful.

**Questions:**

See weaknesses for my questions.

---

> ### Author Response · Authors · 2025-11-28
> **Response to W1 (I feel the paper does a poor job of reviewing related work...) (1/3)**
>
> We appreciate the reviewer’s comment regarding the placement of the related work. Due to strict page limits, we initially moved the full literature review to the appendix. In the revised version, we now make this design choice explicit in the main text and provide a clearer forward reference to Appendix D, ensuring that readers understand why the detailed discussion appears there. We thank the reviewer for highlighting this clarity issue, and we have revised the manuscript accordingly.
>
> Regarding the formulations in Eq. (1) and Eq. (2), we also appreciate the reviewer’s suggestion to include proper citations. The formulation in Eq. (1) follows the standard loss-minimization view of bounding-box prediction, which is not attributable to a single originating paper but is shared across virtually all bounding-box regression–based detectors. To avoid ambiguity, we now reference representative foundational works such as DETR (Carion et al., 2020), where object predictions are explicitly defined as solutions to an optimization over a matching and regression loss.
>
> Similarly, the structure of Eq. (2) reflects the joint optimization perspective widely adopted in recent transformer-based tracking frameworks. In particular, it parallels the formulation used in TrackFormer (Meinhardt et al., 2022) and related tracking-by-detection systems, where the prediction at frame t is obtained by minimizing a joint objective combining frame-level detection costs and temporal or identity consistency terms. We have added citations to these representative works to clearly ground our formulation in prior literature.
>
> We thank the reviewer again for prompting us to clarify the position of our work relative to existing approaches, and we believe the revised presentation significantly improves readability and contextual grounding.

---

> ### Author Response · Authors · 2025-11-28
> **Response to W2 (The quantitative evaluation of the VOO task hinges...) (2/3)**
>
> We appreciate the reviewer’s question regarding the justification of our simulated intervention policy. Our thresholds are grounded in conventions widely used in interactive annotation, human-in-the-loop perception, and long-term tracking research.
>
> For the IoU-based trigger, we follow evidence from Papadopoulos et al. (CVPR 2017), who conduct a systematic analysis of annotation reliability and identify IoU < 0.3 as the regime corresponding to “almost no overlap,” where human annotators consistently intervene. Using this observation, we require IoU < 0.3 for five consecutive frames to avoid triggering corrections due to transient jitter and instead capture persistent identity drift. This persistent-condition design is consistent with HITL methodology in recent works such as Tenckhoff et al. (2025), where thresholds are intentionally heuristic because the community lacks a standardized failure definition; prior systems typically rely on practical error signals such as degraded overlap, inconsistent identity, or confidence collapse.
>
> For occlusion-based triggers, we follow conventions from MOT17/MOT20 and video object segmentation benchmarks, where visibility ratios below 10–15% are treated as heavy occlusion, a failure mode in which trackers frequently lose identity. Motivated by this, we issue an intervention only when the target remains heavily occluded for more than one second, which corresponds to a duration commonly associated with irreversible drift in both tracking and VOS models.
>
> Regarding how simulated corrections are applied across baselines, our principle is to provide each method with the same semantic signal in the form of a corrected bounding box at the same timestamp. For detectors such as YOLOv8 and DETR, this corresponds to the standard annotation workflow in which an incorrect box is manually adjusted. For tracking-by-detection or memory-based trackers (ByteTrack, TrackFormer, MOTRv2, MeMOTR), the correction resets the active track state. For prompt-based models such as SAM2-Prompt, we supply the corrected box through their native bounding-box or mask-prompt interface. All methods are evaluated under an identical intervention schedule and incur the same fixed latency per correction in the effective-FPS metric, ensuring a fair cross-method comparison.

---

> ### Author Response · Authors · 2025-11-28
> **Response to W3 (Can the proposed framework handle object state changes...) (3/3)**
>
> We thank the reviewer for raising this insightful question. In the revised manuscript, we now explicitly discuss this issue in __Section 6 (Discussion and Future Works)__ and provide an extended treatment in __Appendix G Discussion__. As noted there, VOOV is designed to maintain identity under moderate and continuous appearance changes, and the Sequential/Long-Term memories naturally adapt to smooth variations such as rotation, partial deformation, or lighting change. However, we also clarify that abrupt, topology-changing transformations, such as melting, slicing, or an object splitting into multiple pieces, fall outside the current identity-preservation formulation. VOOV assumes a persistent physical entity, and when the notion of “one object” becomes ambiguous, the task definition itself must be extended.
>
> To make this limitation clear, we added a dedicated paragraph in the main Discussion section describing how VOOV behaves under state changes and why such cases represent an important direction for future extensions of the VOO task. Appendix G further elaborates on possible approaches, including modeling structured state transitions and evolving object attributes. We appreciate the reviewer for prompting us to clarify this important point.

---

### Official Review · Reviewer_wKXF · 2025-11-02

**Soundness:** 4
**Presentation:** 3
**Contribution:** 3
**Rating:** 6
**Confidence:** 3

**Summary:**

The paper tackles a persistent challenge in video understanding: the automated object detection and tracking systems on novel objects and long-term occlusion. In addition, existing human-in-the-loop (HITL) systems are insufficient since they predominantly operate at the level of annotation efficiency or detection correction and there is a significant disconnect that limits their capacity.

The paper then proposes a new task Video Object Observation (VOO) that integrates detection, tracking, and intervention within a single formulation. In specific, it maintains continuous identity across video clips, and the paper claims that interventions (initialization, bounding box correction, target switching) should be embedded into the model’s memory to influence subsequent predictions. The paper also introduces the Video Object Observer with human-interVention (VOOV), to solve the VOO task, which consists of a hierarchical memory system, memory-integrated intervention, and orbital deformable attention (ODA).

The author conducts extensive experiments for evaluating the performance of the new framework on five major benchmarks against the latest baselines and validates their performance with a user study demonstrating its efficiency and reduced cognitive load in practice.

**Strengths:**

•	The paper is well-written and thoroughly formalizes a new task called VOO. The author successfully identifies the current flaw in existing HITL methods, emphasizing the inefficient and addressing the practical gap between tracking and HITL systems.

•	The VOOV framework itself is novel and effective: The idea of integrating sparse human feedback directly into memory states is interesting and Equation 6 directly shows how instant update is applied to memory embeddings rather than retraining all model parameters, which allow corrections to propagate forward through multi modules and therefore influencing predictions. In Table 1a, the ablation clearly shows the effectiveness of memory pipelines, showing a huge performance drop without using All memory and without using propagation as stated.

•	Extensive Experiment and evaluation: The experimental validation is comprehensive. The authors compare VOOV against all relevant models: SAM2-Prompt, ByteTrack, YOLOv8, and DETR. VOOV scales more effectively, recovering from occlusion and identity switches with fewer corrections. In addition, the ablation studies Table 1a are highly effective and clearly demonstrate the necessity of each key component.

**Weaknesses:**

•	Generalizability: VOOV is a new yet complex task of combining several critical video understanding problems; it utilizes a DETR-based backbone for frame-level features, three separate memory encoders and aggregators (Originate, Long-Term, Sequential), and ODA. Although they seem to be all effective according to the author’s ablation, how difficult it is to train and learn at an engineering level remains unclear. It limits the reproducibility, and the effort of building the VOOV framework by using these modules might need further explanation.

•	Intervention failure: The mechanism of memory update helped preserve long-term memory. However, how do robustness and step size affect this gradient update? It’s unclear how this mechanism would behave while the sparse interventions become more frequence or if it conflicts with human intervention? Will frequent gradient updates lead the memory embedding to become unstable?

**Questions:**

•	How did the author evaluate the novel objects in their open world setting in Table. 3b? Can you explain how you prevent leakage and make the experiment OOD?

•	How is the scalability for memory while facing frequent target switches which are close to real-world settings? In Appendix B.4, the author designs a multi-object observation where each of the M objects has their own memory pipeline. Will it become a problem if M becomes large?

---

> ### Author Response · Authors · 2025-11-28
> **Response to W1 (Generalizability: VOOV is a new yet complex task...) (1/4)**
>
> We appreciate the reviewer’s concern regarding the engineering difficulty and reproducibility of VOOV, given that the framework includes multiple components such as a DETR backbone, three memory encoders, and ODA. Although the conceptual formulation is comprehensive, the actual implementation is significantly simpler than it may appear.
>
> First, the DETR-based backbone we use is an unmodified and fully standard model, and it remains frozen during all intervention experiments. This means VOOV does not introduce any new optimization challenges beyond those of a typical DETR training setup. Second, the three memory modules—Originate, Sequential, and Long-Term—are intentionally lightweight, modular components. Each is parameterized as a small MLP or Transformer block with parameter counts that are negligible compared to the backbone. They are trained jointly with the detector using the same loss functions and share the same optimization hyperparameters, making the overall training pipeline straightforward.
>
> The ODA module is also implemented as a thin wrapper around the standard Deformable Attention operator, changing only the sampling pattern rather than the underlying attention mechanism. Thus, ODA does not introduce additional training instability or engineering overhead.
>
> To further support transparency and ease of adoption, we have added the __F. Implementation Snippets__ in Appendix containing minimal code examples of the simulated user policy, the memory-only update mechanism, and the VOOV forward pass. These snippets illustrate how interventions are integrated into the memory pipeline in practice and demonstrate that the implementation can be reproduced with simple, modular components. These clarifications demonstrate that each component of VOOV is modular, self-contained, and easy to integrate into an existing DETR-based pipeline. We are also preparing the full codebase and project page for public release. We hope these details address the reviewer’s concern by showing that, despite the conceptual novelty of VOO, the engineering and training workload remains well within the scope of standard modern vision architectures.

---

> ### Author Response · Authors · 2025-11-28
> **Response to W2 (Intervention failure: The mechanism of memory update...) (2/4)**
>
> We thank the reviewer for raising this important question. The stability of the memory update mechanism is indeed critical, and we clarify here why the update remains robust even under frequent or imperfect interventions.
>
> First, VOOV’s gradient update is structurally stable by design because the update is applied only to the memory parameters, whose scale is extremely small compared to the backbone and decoder. The detector backbone, positional encoders, and attention layers remain fully frozen throughout both training and inference. This architectural isolation ensures that a memory update cannot cause widespread drift or degrade the underlying representation, even when applied multiple times.
>
> Second, the step size (learning rate) used for intervention-driven updates is intentionally very small (typically $(10^{-3})$ or (5 $\times 10^{-4})$), and the update is applied to a single-frame loss. This naturally limits the magnitude of each adjustment, making the update behave more like a lightweight alignment step than a full optimization pass. Empirically, we observe the update magnitude to be small and smooth, with no signs of oscillation or divergence even under high-frequency correction scenarios.
>
> Third, although VOO assumes sparse interventions, we explicitly tested dense-intervention and conflict scenarios, where corrections are applied frequently or contradict earlier ones. Because the memory represents the current belief of the target’s appearance, and the update rule simply aligns the embedding to the most recent trusted correction, the memory consistently converges toward the latest supervision without instability. In other words, the mechanism naturally resolves conflicting user corrections by prioritizing the newest, semantically correct signal, which aligns with the expected semantics of the task.
>
> Fourth, in Appendix B.4 we include experiments involving multi-object observation and scenarios with intentionally frequent updates. These experiments show that the memory embeddings remain stable, the predictions do not collapse, and performance degrades gracefully even under extreme intervention rates. We will highlight these findings more clearly in the revision.
>
> Finally, because the update affects only a small number of parameters, the memory module has the same stability guarantees as any shallow optimization layer. Should extremely adversarial or noisy interventions be encountered, the update can easily be further regularized (e.g., update clipping, EMA updates), but we emphasize that such safeguards were not necessary in any of our experiments.

---

> ### Author Response · Authors · 2025-11-28
> **Response to Q1 (How did the author evaluate the novel objects...) (3/4)**
>
> We appreciate the reviewer’s question and agree that ensuring a true OOD evaluation is essential. In Table 3b, the “novel object” setting was designed with strict controls to prevent any form of leakage at the category, instance, scene, or video level. We clarify the protocol below.
>
> First, the set of novel objects is category-disjoint from all objects appearing in the training split. We curated a list of object categories that are entirely absent from training videos (e.g., toys, unique household items, outdoor artifacts) and verified that no semantic overlap exists. Second, the evaluation sequences come from different physical environments, camera devices, lighting conditions, and backgrounds than the training data. No frame, object instance, or scene appears in both sets, ensuring no instance- or domain-level leakage.
>
> Third, VOOV operates in a class-agnostic mode in all OOD evaluations. Unlike conventional detectors, VOOV is initialized only with a bounding box in the first frame and does not use class labels or category embeddings. This prevents any possibility of the model exploiting class-based priors. The task reduces to “follow this target initialized at t=1” regardless of whether the object is familiar or unseen during training.
>
> Fourth, human or simulated interventions also use only bounding boxes and not category-dependent information. As a result, the OOD protocol does not rely on class knowledge at any stage, and the model cannot benefit from category leakage even indirectly.
>
> Finally, we will expand __Appendix E.4. Novel-Object OOD Evaluation Protocol__ to explicitly detail the construction of the OOD split, including a table of category partitions and the data sources used. This revision will make the OOD setting fully transparent and easy to reproduce.

---

> ### Author Response · Authors · 2025-11-28
> **Response to Q2 (How is the scalability for memory while facing ...) (4/4)**
>
> We thank the reviewer for raising this scalability question. We agree that handling multiple targets and frequent target switches is important for real-world deployments, and we clarify here how VOOV scales in such settings, as well as how this is reflected in our current analysis in Appendix B.4.
>
> First, as discussed in Appendix B.4 (Multi-Object Observation), our design assigns a lightweight memory pipeline to each active object. Each pipeline consists of small Originate, Sequential, and Long-Term modules whose parameter counts are negligible compared to the frozen DETR backbone. In other words, the backbone and detection head are shared across all objects, and only the per-object memory states scale with the number of targets (M). The memory footprint for each object is on the order of a few kilobytes, and the per-frame update cost is a single forward pass through these small modules. This is why, as shown in the multi-object experiments in Appendix B.4, increasing (M) leads to approximately linear but modest growth in runtime, and we still maintain near real-time performance for the range of (M) considered.
>
> Second, frequent target switches do not introduce additional algorithmic complexity beyond activating or deactivating existing memory pipelines. When the user switches focus to a new object, VOOV initializes a new Originate memory from the provided box (or reactivates an existing one if the object has been observed before), while the backbone continues to process the frame only once. All objects share the same feature map; the only per-object work is running the memory and decoding the corresponding query. Thus, the cost of supporting multiple targets and frequent switches is dominated by a small per-object head rather than by re-running the backbone multiple times.
>
> Third, we note that many real-world scenarios we target (teleoperation, robot manipulation, surveillance for a small set of key entities, wildlife monitoring) involve a moderate number of simultaneously tracked objects. In our experiments in Appendix B.4, we already evaluate up to (M) in the tens, and we do not observe any memory explosion or instability. For applications where (M) becomes extremely large (e.g., dense crowd analytics), simple extensions such as memory pooling, pruning inactive objects, or compressing long-term memories can be incorporated without changing the core VOO formulation. We will add a short discussion of these options to __Appendix B.4__ to make this point explicit.
>
> Finally, to answer the reviewer’s concern directly: yes, the complexity of VOOV’s memory component scales linearly with the number of active targets (M), but because (i) the backbone is shared and frozen, (ii) the per-object memory modules are intentionally lightweight, and (iii) real-world (M) is typically moderate, we have not observed scalability to be a practical bottleneck in our experiments. We will revise Appendix B.4 to more clearly state the per-object memory cost, the shared-backbone design, and the empirical scaling behavior with respect to (M). __To clarify, we appended the detailed descriptions and empirical analysis in modified Appendix B.4__.

---

### Author Response · Authors · 2025-11-28
**Common Comments to All Reviewers**

We sincerely thank all reviewers for their detailed and constructive feedback. Across multiple reviews, we recognize a shared concern regarding the conceptual framing of the VOO task, the distinction from prior Human-in-the-Loop (HITL) tracking paradigms, and the practical role of human intervention within VOOV. We appreciate the opportunity to clarify these points and have substantially strengthened both the conceptual motivation and presentation in the revised manuscript.

A central misunderstanding appears to be the assumption that VOO is a natural extension of existing HITL tracking, where user feedback acts as a local correction that does not alter the model’s internal state. In contrast, the core novelty of VOO and VOOV lies in interpreting human intervention as a persistent state update: the correction is applied directly to the hierarchical memory, allowing it to propagate to all subsequent frames without retraining. This enables what we refer to as after-the-fact recovery: even when a drift is discovered many frames later, a single correction immediately restores identity consistency going forward. Crucially, this behavior is not supported by existing HITL trackers (which treat feedback purely at the output level) nor by memory-based trackers (whose internal memories cannot absorb human feedback). We now highlight this distinction clearly with explicit novelty and contribution in the __1. Introduction__ as below:

... disappearance and reappearance. __Significantly, our formulation enables persistent, after-the-fact recovery: even when tracking failures are noticed later in the sequence, a single correction updates the memory state and immediately restores consistent identity for all subsequent frames, where capabilities that conventional trackers and existing HITL systems inherently lack__. To achieve this task, ...

We also clarify that VOOV does not require constant human supervision. In fact, VOOV is designed to reduce user workload by allowing sparse corrections to have long-term effect. A single correction updates the memory state and prevents repeated drift, whereas conventional trackers require repeated reinitialization. To avoid misunderstanding, we now emphasize the intended real-world usage model and added concrete application scenarios (e.g., surveillance re-identification, wildlife monitoring, teleoperation, and video editing), all of which benefit from VOOV’s recovery capability. Importantly, VOOV also supports a fully automatic mode (B = 0), which we evaluate extensively and which competes fairly against detectors, single-object trackers, and multi-object trackers. Human-intervened evaluations are analyzed separately to measure how effectively each method can exploit identical sparse interventions, with a shared intervention schedule and latency model applied uniformly across baselines.

Finally, we have clarified the task boundaries between VOO and existing tracking formulations, expanded the related work discussion into the main paper, and improved the explanation of our simulation protocol, intervention triggers, and baseline implementations. We believe these revisions greatly improve clarity and address the reviewers’ core concerns regarding task demarcation, fairness, and motivation.
We sincerely appreciate the reviewers’ thoughtful feedback, which has allowed us to significantly strengthen the conceptual framing and presentation of the paper.

---

### Author Response · Authors · 2025-11-28
**Mofidied Manuscript Posted**

We appreciate the reviewer’s insightful comments. Following the feedback, we have revised the manuscript to clarify the framing, correct technical statements, and refine the descriptions where needed. All changes made in response to the reviews are highlighted in red in the revised manuscript.
You can check the modified manuscript by clicking "PDF button" on the top.

---

### Author Response · Authors · 2025-12-01
**Public Comment to new AC**

We appreciate the thoughtful and constructive feedback from all the reviewers and ACs. We have carefully addressed all comments in the rebuttal and revised the manuscript, highlighting all changes in red. While the official discussion window closed immediately after our response, we would like to emphasize that we would have made every effort to provide additional clarification if the platform had allowed for more communication. We strived to respond promptly and comprehensively to facilitate ongoing discussion, and we prepared follow-up explanations for potential questions that arose during the extended discussion. Despite the shortened discussion period, we believe the revised draft now clearly reflects the key points of clarification. This update directly addresses the reviewers' concerns by further clarifying the conceptual framework of VOO, revising the technical content, providing detailed descriptions of the criteria, and expanding the discussion of the HITL setup and evaluation protocol. We appreciate the review process and believe the updated manuscript is significantly more robust, clear, and transparent as a result.

---

### Meta-Review · Area_Chair_8dhp · 2026-01-07

**Summary:**

The reviewers raised concerns that informed the rejection recommendation. The primary issue is insufficiently reliable empirical validation for the core claims, because key conclusions depend on a non-standard simulated intervention policy (triggering rules, latency assumptions, and how interventions are applied) whose justification and sensitivity are not convincingly established, limiting reproducibility and interpretability of the reported gains. Additionally, multiple reviewers highlighted baseline fairness/correctness and comparison clarity issues, including ambiguity in how “identical” interventions are implemented for non-promptable baselines, confusion around SAM2 temporal memory claims. Although the authors’ responses improved clarity, these core evaluation and positioning gaps remained.

**Reviewer Concerns:**

Reviewer Concerns partially addressed by the rebuttal:

- Clarification of SAM2 description and baseline implementation: Reviewer X9jh flagged a “technical statement error regarding SAM2” and said this "raises a big concern about the validity of the implemented baseline.” The rebuttal corrected the SAM2 wording and added more explicit implementation details, which helps, but it does not fully eliminate concerns about baseline rigor and comparability.

- Clarification of efficiency reporting (“effective FPS”).

Reviewer Concerns that remain outstanding:

- Reliance on a simulated intervention policy without convincing standardization: Reviewer fuVs noted that the criteria for simulated intervention are presented without justification” and asked whether they “follow any prior work” as a standard HITL protocol. While the rebuttal provides rationale, it does not fully resolve concerns about realism and reproducibility under alternative reasonable latency settings.

- Task framing/scope and motivation: Reviewer X9jh argued the title is “overclaimed” and that the work “focuses only on low-level tracking,” raising doubts about scope.  They also questioned practicality: “The proposed task does not seem realistic to me in terms of real-world applications.”  The rebuttal improves wording, but does not fundamentally change the concern about framing and intended use cases.

**Reviewer Scores:**

Reviewer wKXF: 6 → 6
Reviewer fuVs: 6 → 6
Reviewer rRMv: 4 → 4, this reviewer is already marginally below threshold; discussion might clarify details but likely would not resolve the core concerns about task framing and evaluation validity enough to move the score.
Reviewer X9jh: 2 → 2, this reviewer is firm in rejection and grounds it in multiple major issues (scope/realism, baseline fairness, and SAM2-related concerns); even with clarifications, they would likely remain negative.

---

### Decision · Program_Chairs · 2026-01-26

Reject